# Transformers Handle Endogeneity in In-Context Linear Regression

**Haodong Liang**
UC Davis
hdliang@ucdavis.edu

**Krishnakumar Balasubramanian**
UC Davis
kbala@ucdavis.edu

**Lifeng Lai**
UC Davis
lflai@ucdavis.edu

## Abstract

We explore the capability of transformers to address endogeneity in in-context linear regression. Our main finding is that transformers inherently possess a mechanism to handle endogeneity effectively using instrumental variables (IV). First, we demonstrate that the transformer architecture can emulate a gradient-based bi-level optimization procedure that converges to the widely used two-stage least squares (2SLS) solution at an exponential rate. Next, we propose an in-context pretraining scheme and provide theoretical guarantees showing that the global minimizer of the pre-training loss achieves a small excess loss. Our extensive experiments validate these theoretical findings, showing that the trained transformer provides more robust and reliable in-context predictions and coefficient estimates than the 2SLS method, in the presence of endogeneity.

## 1 Introduction

The transformer architecture (Vaswani et al., 2017) has demonstrated remarkable in-context learning (ICL) capabilities across various domains, such as natural language processing (Devlin et al., 2019; Radford et al., 2019; Brown et al., 2020), computer vision (Dosovitskiy et al., 2021; Carion et al., 2020), and reinforcement learning (Lee et al., 2022; Parisotto et al., 2020). Self-attention mechanism, a core component of transformers, allows these models to capture long-range dependencies in data, which is critical for success in these tasks. Despite their impressive performance, the theoretical understanding of transformers remains limited, leaving important questions unanswered about their true capabilities and the underlying mechanisms driving their exceptional results.

Recent efforts to theoretically understand transformers' ICL capabilities have focused on their performance in fundamental statistical tasks. Focusing on simple function classes, Garg et al. (2022) highlighted that transformers, when trained on sufficiently large and diverse data from a specific function class, can generalize across most functions of that class without task-specific fine-tuning. Building on this, subsequent work by Bai et al. (2023) established that attention layers enable transformers to perform gradient descent, implementing algorithms like linear regression, logistic regression, and LASSO; see also Akyürek et al. (2023); Von Oswald et al. (2023); Li et al. (2023); Fu et al. (2023); Ahn et al. (2023); Jin et al. (2025). The learning dynamics of transformer was analyzed in Huang et al. (2024). Furthermore Zhang et al. (2024a;b) showed that *trained* transformers' ICL abilities for linear regression tasks are theoretically robust under certain distributional shifts.

Existing works on analyzing the ICL ability of transformers for linear regression tasks, however, ignore *endogeneity* and have mainly focused on the *exogenous* setup where the additive noise is uncorrelated with the explanatory variables. Ignoring *endogeneity* in linear regression leads to biased and inconsistent estimates, resulting from issues like omitted variable bias, simultaneity, and measurement error, which can distort causal inferences and lead to incorrect policy conclusions (Hausman, 2001; Wooldridge, 2015; Angrist & Pischke, 2009; Greene, 2018). Instrumental variable (IV) regression is a widely adopted method to handle endogeneity by utilizing instruments that are correlated with the endogenous variables but uncorrelated with the error term (Angrist & Krueger, 2001). A naturally intriguing question that therefore arises is:

*Can transformers leverage instrumental variables and provide reliable predictions
and coefficient estimates, in the presence of endogeneity?*

In this work, we aim to answer this question and offer new insights on in-context linear regression tasks. Our key contributions include:

- We demonstrate that looped transformers can address endogeneity in linear regression by leveraging instrumental variables. Specifically, we show that transformers can implement two-stage least squares (2SLS) regression through a bi-level gradient descent procedure, where each iteration is executed by a two-layer transformer block. Moreover, the convergence rate to the 2SLS estimator is exponential with respect to the number of blocks.
- We propose an ICL training scheme for transformers to efficiently handle endogeneity. Under this scheme, we show that the global minimizer of the in-context pre-training loss achieves a small excess loss compared to the global optimal expected loss.
- We evaluate the performance of the trained transformer model through extensive experiments, finding that it not only matches the performance of the 2SLS estimator on standard IV tasks but also generalizes effectively to more complex scenarios, including the challenging cases of weak instruments, non-linear IV, and underdetermined IV problems.
- As part of our analysis, we derive the first non-asymptotic bound for the 2SLS estimator under random design, providing valuable insights for future theoretical work.

## 1.1 RELATED WORKS

**In-context Learning.** Initial works by Garg et al. (2022) and Bai et al. (2023) adopted the standard multi-layer transformer architecture to conduct the experiments. Later, Giannou et al. (2023) and Yang et al. (2024) showed that a looped architecture reduces the required depth of transformers and exhibits better efficiency in learning algorithms. Gao et al. (2024) illustrated that the looped transformer architecture with extra pre-processing and post-processing layers can achieve higher expressive power than a standard transformer with the same number of parameters. Apart from works concerning the implementability of first-order gradient descent algorithms by transformers, other works have also examined higher-order and non-parametric optimization methods. Specifically, Giannou et al. (2024) showed that transformers can emulate Newton's method for logistic regression. Cheng et al. (2024) showed that transformers can implement functional gradient descent and hence enable them to learn non-linear functions in-context. Relationship between in-context learning and Bayesian inference is also studied in Ye et al. (2024); Falck et al. (2024).

Nichani et al. (2024) illustrated how the transformers can learn the causal structure by encoding the latent causal graph in the first attention layer. Goel & Bartlett (2024) explored the representational power of transformer for learning linear dynamical systems. Makkuva et al. (2024a;b); Rajaraman et al. (2024); Edelman et al. (2024) considered ICL Markov chains with transformers, including both landscape and training dynamics analyses. To the best of our knowledge, we are not aware of prior works on handling endogeniety with transformers.

**Instrumental Variable Regression.** IV regression has been widely studied in econometrics (Angrist & Krueger, 2001; Angrist & Pischke, 2009). Recent works in machine learning explored the optimization based approaches for the IV regression problem. Singh et al. (2019) proposed the kernel IV regression to model non-linear relationship between variables. Muandet et al. (2020) proposed that a non-linear IV regression problem can be formulated as a convex-concave saddle point problem. Della Vecchia & Basu (2023); Chen et al. (2024); Fonseca et al. (2024) proposed a stochastic optimization algorithm for IV regression.

*Notation:* Throughout this paper, unless otherwise specified, lower-case letters denote random variables or samples, while upper-case letters represent datasets (collections of samples). Bolded letters indicate vectors or matrices, whereas unbolded letters indicate scalars. The notation $\boldsymbol{X}_{:,i}$ refers to the $i$-th column, and $\boldsymbol{X}_{i,:}$ refers to the $i$-th row of matrix $\boldsymbol{X}$. $\lambda_{\min}(\cdot)$ denotes the minimum eigenvalue, and $\sigma_{\min}(\cdot)$ denotes the minimum singular value of a matrix. By default, $\|\cdot\|$ denotes the Euclidean norm for a vector, or the spectral norm for a matrix.

## 2 ENDOGENEITY AND INSTRUMENTAL VARIABLE REGRESSION

Suppose we are interested in estimating the relationship between response variable $y \in \mathbb{R}$ and predictor variable $\boldsymbol{x} \in \mathbb{R}^p$ with endogeneity. Given instruments $\boldsymbol{z} \in \mathbb{R}^q$, we consider the model

$$y = \boldsymbol{\beta}^\top \boldsymbol{x} + \epsilon_1, \quad \text{and} \quad \boldsymbol{x} = \boldsymbol{\Theta}^\top \boldsymbol{z} + \boldsymbol{\epsilon_2}, \tag{1}$$

where $\boldsymbol{\beta} \in \mathbb{R}^p$, and $\boldsymbol{\Theta} \in \mathbb{R}^{q \times p}$ are the true model parameters, $\epsilon_1 \in \mathbb{R}$ and $\boldsymbol{\epsilon_2} \in \mathbb{R}^p$ are (centered) random noise terms with variance $\sigma_1^2$ and covariance matrix $\boldsymbol{\Sigma}_2$, respectively. Further, $\boldsymbol{\epsilon}_2$ is an unobserved noise correlated with $\epsilon_1$, leading to the correlation between $\boldsymbol{x}$ and $\epsilon_1$, which introduces confounding in the model between $\boldsymbol{x}$ and $y$. Under this setting, the standard ordinary least squares (OLS) estimator is a biased and inconsistent estimator of $\boldsymbol{\beta}$ (see Wooldridge (2015), Chapter 9). To address this issue, instrumental variable (IV) regression is a widely used method to provide a consistent estimate for $\boldsymbol{\beta}$.

**Definition 2.1** (2SLS estimator). IV regression is a regression model to provide consistent estimate on the causal effect $\boldsymbol{\beta}$ for the endogeneity problem (1), by utilizing the instrument $\boldsymbol{z}$. Given observational values $(\boldsymbol{Z}, \boldsymbol{X}, \boldsymbol{Y}) = \{(\boldsymbol{z}_i, \boldsymbol{x}_i, y_i)\}_{i=1}^n$, the standard approach to estimate the IV regression model is 2SLS; see, for example, Wooldridge (2015), Chapter 15.

    i. *First stage*: Regress $\boldsymbol{X}$ on $\boldsymbol{Z}$ to obtain $\hat{\boldsymbol{\Theta}}$

$$\hat{\boldsymbol{\Theta}} = (\boldsymbol{Z}^\top \boldsymbol{Z})^{-1} \boldsymbol{Z}^\top \boldsymbol{X}.$$

    ii. *Second stage*: Regress $\boldsymbol{Y}$ on $\boldsymbol{Z}\hat{\boldsymbol{\Theta}}$ to obtain:

$$\hat{\boldsymbol{\beta}}_{\text{2SLS}} = (\hat{\boldsymbol{\Theta}}^\top \boldsymbol{Z}^\top \boldsymbol{Z} \hat{\boldsymbol{\Theta}})^{-1} \hat{\boldsymbol{\Theta}}^\top \boldsymbol{Z}^\top \boldsymbol{Y}. \tag{2}$$

We introduce the standard assumptions required to show the convergence rate of the above estimator.

**Assumption 1** (Instrumental variable). A random variable $\boldsymbol{z} \in \mathbb{R}^q$ is a valid IV, if it satisfies the following conditions:

    i. Fully identification: $q \geq p$ (without loss of generality, we assume data $\boldsymbol{Z}, \boldsymbol{X}$ are full rank).
    ii. Correlated to $\boldsymbol{x}$: $\text{Corr}(\boldsymbol{z}, \boldsymbol{x}) \neq \boldsymbol{0}$.
    iii. Conditional uncorrelated to $y$: $\text{Corr}(\boldsymbol{z}, \epsilon_1) = 0$.

In particular, condition (i) above ensures the existence of unique solution for $\hat{\boldsymbol{\beta}}_{\text{2SLS}}$. We refer to Stock & Watson (2011, Chapter 12) for additional elaborate discussions on the above conditions. To derive non-asymptotic convergence rates, we further assume the following regularity conditions.

**Assumption 2** (Regularity conditions). Suppose instrument $\boldsymbol{z}$ is a centered random variable. We assume the following conditions hold:

    i. Bounded parameters: $\|\boldsymbol{\beta}\| \leq B_\beta$, $\|\boldsymbol{\Theta}\| \leq B_\Theta$.

    ii. Bounded variables: $\|\boldsymbol{z}\| \leq B_z$, $\|\boldsymbol{x}\| \leq B_x$, $|\epsilon_1| \leq B_{\epsilon_1}$, $\|\boldsymbol{\epsilon}_2\| \leq B_{\epsilon_2}$.

    iii. Linear instrument: $\mathbb{E}\left[x_k | \boldsymbol{z}\right] = \langle \boldsymbol{\Theta}_k, \boldsymbol{z} \rangle$.

The boundedness condition in (ii) is required to invoke matrix Bernstein inequalities (Tropp, 2015) in the analysis. We anticipate that this condition may be relaxed to subgaussian or moment conditions by using more sophisticated matrix concentration results.

**Theorem 2.1** (MSE of 2SLS estimator). *Given Assumptions 1 and 2, consider clipping operation*

$$\textit{clip}_{B_\beta}(\hat{\boldsymbol{\beta}}) := \begin{cases} \hat{\boldsymbol{\beta}} & \textit{if } \|\hat{\boldsymbol{\beta}}\| \leq B_\beta \\ \frac{B_\beta}{\|\hat{\boldsymbol{\beta}}\|} \hat{\boldsymbol{\beta}} & \textit{if } \|\hat{\boldsymbol{\beta}}\| > B_\beta \end{cases}.$$

*When*

$$n \geq \max\left\{ 4c^2 B_z^4 \left( q + \log\left(\frac{4c^2 B_z^4 K}{q}\right) - \frac{3}{2} \right), \frac{q e^{\frac{3}{2}}}{K}, \frac{p^2(q+1)^2 K}{q K_0^2} \right\},$$

where $K := \frac{\lambda_{\min}(\mathbf{\Sigma}_z)}{6B_z^2}$ and $K_0 := \frac{\lambda_{\min}(\mathbf{\Sigma}_z)\sigma_{\min}^2(\mathbf{\Theta})}{2B_{\epsilon_2}^2}$, the mean squared error of the **2SLS** estimate is bounded by:

$$\mathbb{E}\left[\|\textbf{\textit{clip}}_{B_\beta}(\hat{\boldsymbol{\beta}}_{\textsf{2SLS}}) - \boldsymbol{\beta}\|^2\right] \leq \mathcal{O}\left(\frac{q}{n}\left(\frac{B_\beta^2}{K} + C^2(n)\sigma_1^2\right)\right), \tag{3}$$

where $C(n)$ is defined in Equation (42), $\mathbf{\Sigma}_z := \mathbb{E}[\boldsymbol{z}\boldsymbol{z}^\top]$, and $c$ is an absolute constant.

**Remark 2.1.** We keep the slightly complicated form (3) so that the $\mathcal{O}$ notation only hides some absolute constant multipliers that are independent of problem-related constants. Note that when $n$ is large enough, we have $C(n) \to \frac{B_\Theta B_z}{\lambda_{\min}(\mathbf{\Sigma}_z)\sigma_{\min}^2(\mathbf{\Theta})}$, so $C(n)$ is also bounded. Thus the error bound (3) decays with rate $\mathcal{O}(\frac{1}{n})$.

We note that although the consistency of the **2SLS** estimator is a standard result in econometrics, most existing works focus on the asymptotic properties of the estimator. Theorem 2.1 provides the first non-asymptotic bound for estimation error $\|\hat{\boldsymbol{\beta}}_{\textsf{2SLS}} - \boldsymbol{\beta}\|^2$, under random design. The detailed proof is provided in Appendix A.1.

## 3 TRANSFORMERS HANDLE ENDOGENIETY

### 3.1 TRANSFORMER ARCHITECTURE

Denote the input matrix as $\boldsymbol{H} = [\boldsymbol{h}_1, \ldots, \boldsymbol{h}_n] \in \mathbb{R}^{D \times n}$, where each column corresponds to one sample vector.

**Definition 3.1** (Attention layer)**.** A self-attention layer with $M$ heads is denoted as $\textsf{ATTN}_{\boldsymbol{\theta}}(\cdot)$, with parameters $\boldsymbol{\theta} = \{(\boldsymbol{Q}_m, \boldsymbol{K}_m, \boldsymbol{V}_m)\}_{m \in [M]} \subseteq \mathbb{R}^{D \times D}$. Given input $\boldsymbol{H}$,

$$\tilde{\boldsymbol{H}} = \textsf{ATTN}_{\boldsymbol{\theta}}(\boldsymbol{H}) := \boldsymbol{H} + \frac{1}{n}\sum_{m=1}^{M}(\boldsymbol{V}_m\boldsymbol{H}) \times \sigma((\boldsymbol{Q}_m\boldsymbol{H})^\top(\boldsymbol{K}_m\boldsymbol{H})) \in \mathbb{R}^{D \times n}, \tag{4}$$

or element-wise:

$$\tilde{\boldsymbol{h}}_i = [\textsf{ATTN}_{\boldsymbol{\theta}}(\boldsymbol{H})]_i := \boldsymbol{h}_i + \sum_{m=1}^{M}\frac{1}{n}\sum_{j=1}^{n}\sigma(\langle\boldsymbol{Q}_m\boldsymbol{h}_i, \boldsymbol{K}_m\boldsymbol{h}_j\rangle) \cdot \boldsymbol{V}_m\boldsymbol{h}_j \in \mathbb{R}^D, \tag{5}$$

where $\sigma(\cdot)$ is the ReLU function.

**Definition 3.2** (MLP layer)**.** An MLP layer is denoted as $\textsf{MLP}_{\boldsymbol{\theta}}(\cdot)$, with parameters $\boldsymbol{\theta} = (\boldsymbol{W}_1, \boldsymbol{W}_2) \in \mathbb{R}^{D' \times D \times D \times D'}$. Given input $\boldsymbol{H}$,

$$\tilde{\boldsymbol{H}} = \textsf{MLP}_{\boldsymbol{\theta}}(\boldsymbol{H}) := \boldsymbol{H} + \boldsymbol{W}_2\sigma(\boldsymbol{W}_1\boldsymbol{H}),$$

or element-wise:

$$\tilde{\boldsymbol{h}}_i = [\textsf{MLP}_{\boldsymbol{\theta}}(\boldsymbol{H})]_i := \boldsymbol{h}_i + \boldsymbol{W}_2\sigma(\boldsymbol{W}_1\boldsymbol{h}_i).$$

**Definition 3.3** (Transformer)**.** An L-layer transformer is denoted as $\textsf{TF}_{\boldsymbol{\theta}}(\cdot)$, with parameters $\boldsymbol{\theta} = (\boldsymbol{\theta}_{\textsf{ATTN}}^{(1:L)}, \boldsymbol{\theta}_{\textsf{MLP}}^{(1:L)})$. Given input $\boldsymbol{H} = \boldsymbol{H}^{(0)}$,

$$\boldsymbol{H}^{(l)} = \textsf{MLP}_{\boldsymbol{\theta}_{\textsf{MLP}}^{(l)}}(\textsf{ATTN}_{\boldsymbol{\theta}_{\textsf{ATTN}}^{(l)}}(\boldsymbol{H}^{(l-1)})), \quad l = 1, \ldots, L.$$

The output of this transformer is the final layer output: $\tilde{\boldsymbol{H}} := \boldsymbol{H}^{(L)} = \textsf{TF}_{\boldsymbol{\theta}}(\boldsymbol{H}^{(0)})$.

**Definition 3.4** (Looped transformer)**.** An $\bar{L}$-looped transformer is a special transformer architecture, denoted as $\textsf{LTF}_{\bar{\boldsymbol{\theta}}, \bar{L}}(\cdot)$, with parameters $\bar{\boldsymbol{\theta}} = (\bar{\boldsymbol{\theta}}_{\textsf{ATTN}}^{(1:L_0)}, \bar{\boldsymbol{\theta}}_{\textsf{MLP}}^{(1:L_0)})$. Given input $\boldsymbol{H} = \boldsymbol{H}^{(0)}$,

$$\boldsymbol{H}^{(l)} = \textsf{TF}_{\bar{\boldsymbol{\theta}}}(\boldsymbol{H}^{(l-1)}), \quad l = 1, \ldots, \bar{L}.$$

The output of this looped transformer is the final loop output: $\tilde{\boldsymbol{H}} := \boldsymbol{H}^{(\bar{L})} = \textsf{LTF}_{\bar{\boldsymbol{\theta}}, \bar{L}}(\boldsymbol{H}^{(0)})$.

Previous works (e.g., Bai et al. (2023), Zhang et al. (2024a)) have shown that transformers can perform in-context linear regression by emulating gradient descent (GD) with in-context pretraining. However, these studies have two key limitations. First, their analysis is based on single-level optimization algorithms, which is insufficient to demonstrate that transformers can efficiently learn more complex algorithms like 2SLS (Definition 2.1). Second, most ICL-related research focuses on the predictive performance of transformers, paying little attention to their ability to provide accurate coefficient estimates. We extend the current ICL framework by showing that transformers can implement a bi-level GD procedure (see Section 3.2) with looped transformer architecture (Definition 3.4), allowing them to efficiently emulate 2SLS and provide coefficient estimates that are at least as accurate as 2SLS in the presence of endogeneity (as in Equation (1)).

## 3.2 GRADIENT DESCENT BASED IV REGRESSION

We first introduce a gradient-based bi-level optimization procedure to obtain the 2SLS estimator in Equation (2). Given the dataset $(\boldsymbol{Z}, \boldsymbol{X}, \boldsymbol{Y}) = \{(\boldsymbol{z}_i, \boldsymbol{x}_i, y_i)\}_{i=1}^n$, the objective function of IV regression can be formulated as the following bi-level optimization problem:

$$\min_{\boldsymbol{\beta}} \quad \mathcal{L}(\boldsymbol{\beta}) = \frac{1}{n} \sum_{i=1}^n (y_i - \boldsymbol{z}_i^\top \hat{\boldsymbol{\Theta}} \boldsymbol{\beta})^2, \quad \text{where} \quad \hat{\boldsymbol{\Theta}} := \arg\min_{\boldsymbol{\Theta}} \quad \frac{1}{n} \sum_{j=1}^n (\boldsymbol{x}_j - \boldsymbol{z}_j^\top \boldsymbol{\Theta})^2. \quad (6)$$

Consider the following gradient updates with learning rates $\alpha, \eta$:

$$\boldsymbol{\Theta}^{(t+1)} = \boldsymbol{\Theta}^{(t)} - \eta \boldsymbol{Z}^\top (\boldsymbol{Z}\boldsymbol{\Theta}^{(t)} - \boldsymbol{X}), \quad (7a)$$

$$\boldsymbol{\beta}^{(t+1)} = \boldsymbol{\beta}^{(t)} - \alpha \boldsymbol{\Theta}^{(t)\top} \boldsymbol{Z}^\top (\boldsymbol{Z}\boldsymbol{\Theta}^{(t)} \boldsymbol{\beta}^{(t)} - \boldsymbol{Y}). \quad (7b)$$

Note that the GD-2SLS updates in Equation (7) are designed to solve Equation (6). We now show that regardless the convergence of $\boldsymbol{\Theta}^{(t)}$, the GD estimator $\boldsymbol{\beta}^{(t)}$ will always converge to the 2SLS estimator in Equation (2) with exponential rate.

**Theorem 3.1** (Implementing 2SLS with gradient-based method). *Given training data* $(\boldsymbol{Z}, \boldsymbol{X}, \boldsymbol{Y}) = \{(\boldsymbol{z}_i, \boldsymbol{x}_i, y_i)\}_{i=1}^n$. *Suppose the learning rates* $\alpha, \eta$ *satisfy the following conditions:*

$$0 < \alpha < \frac{2}{\sigma_{\max}^2(\boldsymbol{Z}\hat{\boldsymbol{\Theta}})} \quad \text{and} \quad 0 < \eta < \frac{2}{\sigma_{\max}^2(\boldsymbol{Z})},$$

*where* $\sigma_{\max}(\cdot)$ *denotes the largest singular value of a matrix. Then, the GD updates in Equation (7) converge to the 2SLS estimator at an exponential rate:*

$$\|\boldsymbol{\beta}^{(t)} - \hat{\boldsymbol{\beta}}_{\textit{2SLS}}\| \leq \mathcal{O}\left(\Lambda^t\right),$$

*where, with* $\rho(\cdot)$ *denoting the spectral radius of the matrix,*

$$\Lambda := \max\{\gamma(\alpha), \kappa(\eta)\}, \qquad \gamma(\alpha) := \rho(\boldsymbol{I} - \alpha \hat{\boldsymbol{\Theta}}^\top \boldsymbol{Z}^\top \boldsymbol{Z} \hat{\boldsymbol{\Theta}}), \quad \kappa(\eta) := \rho(\boldsymbol{I} - \eta \boldsymbol{Z}^\top \boldsymbol{Z}). \quad (8)$$

To the best of our knowledge, Theorem 3.1 provides the first theoretical result demonstrating that 2SLS can be efficiently implemented using a gradient-based method, with an exponential convergence rate. We provide the proof in Appendix B.1 and present simulation results in Appendix C.1 to examine the convergence behavior of the optimization process.

## 3.3 TRANSFORMERS CAN EFFICIENTLY IMPLEMENT GD-2SLS

The looped transformer architecture (Definition 3.4), as proposed by Giannou et al. (2023), introduces an efficient approach to learn iterative algorithms by cascading the same transformer block for multiple times. With the GD updates in Equation (7), we will show that there exists a looped transformer architecuture that can efficiently learn the 2SLS estimator. We emphasize here that although we can implement 2SLS by sequentially attaching two separate GD iterates (each handling OLS for one stage), the overall convergence depends heavily on the convergence of the first stage estimate $\hat{\boldsymbol{\Theta}}$. Hence, significantly more number of layers are needed to ensure convergence. In addition, the advantage of looped transformer architecture cannot be fully exploited with this approach.

**Theorem 3.2** (Implement a step of GD-2SLS with a transformer block). *Suppose the embedded input matrix takes the form:*

$$
\boldsymbol{H}^{(2l)} = \begin{bmatrix}
\boldsymbol{z}_1 & \cdots & \boldsymbol{z}_n & \boldsymbol{z}_{n+1} \\
\boldsymbol{x}_1 & \cdots & \boldsymbol{x}_n & \boldsymbol{x}_{n+1} \\
y_1 & \cdots & y_n & 0 \\
\boldsymbol{\Theta}^{(l)}_{:,1} & \cdots & \boldsymbol{\Theta}^{(l)}_{:,1} & \boldsymbol{\Theta}^{(l)}_{:,1} \\
\vdots & \vdots & \vdots & \vdots \\
\boldsymbol{\Theta}^{(l)}_{:,p} & \cdots & \boldsymbol{\Theta}^{(l)}_{:,p} & \boldsymbol{\Theta}^{(l)}_{:,p} \\
\boldsymbol{\beta}^{(l)} & \cdots & \boldsymbol{\beta}^{(l)} & \boldsymbol{\beta}^{(l)} \\
\hat{\boldsymbol{x}}^{(l)}_1 & \cdots & \hat{\boldsymbol{x}}^{(l)}_n & \hat{\boldsymbol{x}}^{(l)}_{n+1} \\
1 & \cdots & 1 & 1 \\
1 & \cdots & 1 & 0
\end{bmatrix} \in \mathbb{R}^{D\times(n+1)}. \tag{9}
$$

*Given $\boldsymbol{H}^{(2l)}$, there exists a double-layer attention-only transformer block with parameters $\boldsymbol{\theta} = \boldsymbol{\theta}^{(2l+1:2l+2)}_{ATTN} = \{(\boldsymbol{Q}^{(2l+1:2l+2)}_m, \boldsymbol{K}^{(2l+1:2l+2)}_m, \boldsymbol{V}^{(2l+1:2l+2)}_m)\}_{m\in[M^{(2l+1:2l+2)}]} \subset \mathbb{R}^{D\times D}$, where the number of heads $M^{(2l+1)} = 2p$, $M^{(2l+2)} = 2(p+1)$ and embedding dimension $D = qp+3p+q+3$, that implements a 2SLS gradient update in Equation (7) with any given learning rates $\alpha, \eta$:*

$$
\boldsymbol{H}^{2(l+1)} = TF_{\boldsymbol{\theta}^{(2l+1:2l+2)}_{ATTN}}(\boldsymbol{H}^{(2l)}) = \begin{bmatrix}
\boldsymbol{z}_1 & \cdots & \boldsymbol{z}_n & \boldsymbol{z}_{n+1} \\
\boldsymbol{x}_1 & \cdots & \boldsymbol{x}_n & \boldsymbol{x}_{n+1} \\
y_1 & \cdots & y_n & 0 \\
\boldsymbol{\Theta}^{(l+1)}_{:,1} & \cdots & \boldsymbol{\Theta}^{(l+1)}_{:,1} & \boldsymbol{\Theta}^{(l+1)}_{:,1} \\
\vdots & \vdots & \vdots & \vdots \\
\boldsymbol{\Theta}^{(l+1)}_{:,p} & \cdots & \boldsymbol{\Theta}^{(l+1)}_{:,p} & \boldsymbol{\Theta}^{(l+1)}_{:,p} \\
\boldsymbol{\beta}^{(l+1)} & \cdots & \boldsymbol{\beta}^{(l+1)} & \boldsymbol{\beta}^{(l+1)} \\
\hat{\boldsymbol{x}}^{(l+1)}_1 & \cdots & \hat{\boldsymbol{x}}^{(l+1)}_n & \hat{\boldsymbol{x}}^{(l+1)}_{n+1} \\
1 & \cdots & 1 & 1 \\
1 & \cdots & 1 & 0
\end{bmatrix} \in \mathbb{R}^{D\times(n+1)}.
$$

Our existence proof specifies an attention structure such that one layer updates only the first-stage estimate $\hat{\boldsymbol{x}}^{(l)}_i$ for all samples, followed by another layer to update the parameters $\boldsymbol{\Theta}^{(l)}$ and $\boldsymbol{\beta}^{(l)}$. Furthermore, as noted in the proof of Theorem 3.2 (ref. Appendix B.2), regardless of the initial values of $\boldsymbol{\Theta}^{(l)}, \boldsymbol{\beta}^{(l)}$ and $\hat{\boldsymbol{x}}^{(l)}$, the structures of the transformer blocks remain the same. This allows us to exploit the looped transformer architecture to significantly reduce the number of parameters and improve learning efficiency (Yang et al., 2024).

By cascading the transformer block $\bar{L}$ times, with Theorem 3.1, one can show that transformers are able to mimic the 2SLS estimator with exponential convergence rate, as described in the following corollary.

**Corollary 3.1** (Implementing GD-2SLS with looped transformer). *For any $0 < \varepsilon < 1$, given learning rates $\alpha, \eta$, and $\Lambda \in (0,1)$, as defined in Equation (8), there exists a transformer formulated as $TF_{\boldsymbol{\theta}}(\cdot) := TF_{\boldsymbol{\theta}'}(LTF_{\bar{\boldsymbol{\theta}}, \bar{L}}(\cdot))$, which consists of an $\bar{L}$-looped transformer $LTF_{\bar{\boldsymbol{\theta}}, \bar{L}}$ with $\bar{\boldsymbol{\theta}} = \bar{\boldsymbol{\theta}}^{(1:2)}_{ATTN} = \{(\bar{\boldsymbol{Q}}^{(1:2)}_m, \bar{\boldsymbol{K}}^{(1:2)}_m, \bar{\boldsymbol{V}}^{(1:2)}_m)\}_{m\in[\bar{M}^{(1:2)}]} \subset \mathbb{R}^{D\times D}$, $\bar{L} = \lceil \mathcal{O}(\log_\Lambda(\varepsilon)) \rceil$, and a final attention layer[1] $\boldsymbol{\theta}' = \boldsymbol{\theta}'_{ATTN} = \{(\boldsymbol{Q}'_m, \boldsymbol{K}'_m, \boldsymbol{V}'_m)\}_{m\in[M']} \subset \mathbb{R}^{D\times D}$, where $\bar{M}^{(1)} = 2p$, $\bar{M}^{(2)} = 2(p+1)$, $M' = 2$, such that given embedded input $\boldsymbol{H}^{(0)}$ taking the format in Equation (9), the model output satisfies:*

$$
|read_y(TF_{\boldsymbol{\theta}}(\boldsymbol{H}^{(0)})) - \hat{\boldsymbol{\beta}}^{\top}_{2SLS}\boldsymbol{x}_{n+1}| \leq B_x\varepsilon,
$$

*where $read_y(\cdot)$ is a function that reads the prediction $\hat{y}_{n+1}$ from the output of the transformer.*

We emphasize here that our construction differs from the implementation of Bai et al. (2023, Theorem 4) for OLS in the following aspects:

---

[1] This layer updates the prediction $\hat{y}_{n+1} := \boldsymbol{\beta}^{(\bar{L})\top}\boldsymbol{x}_{n+1}$, which can be constructed with 2 attention heads using the same architecture as Bai et al. (2023, Theorem 13)

    i. We apply the square loss as defined in Equation (6) to learn the 2SLS estimator, which simplifies the loss function's sum-of-ReLU representation.

    ii. The dimension of the input embedding is $D = qp + 3p + q + 3$, where the extra dimensions store the vectorized parameters $\boldsymbol{\Theta}^{(l)}, \boldsymbol{\beta}^{(l)}$, and the first stage estimate $\hat{\boldsymbol{x}}^{(l)}$.

    iii. We use a two-layer attention-only transformer block $\bar{\boldsymbol{\theta}}$ to implement a 2SLS GD update (7), with the first layer to update the current first-stage estimate $\hat{\boldsymbol{x}}^{(l)}$, and the second layer to update the parameters $\boldsymbol{\Theta}^{(l)}$ and $\boldsymbol{\beta}^{(l)}$.

    iv. For each transformer block, in the first layer, we equip 2 heads to update each dimension of $\hat{\boldsymbol{x}}_i^{(l)} \in \mathbb{R}^p$ for all samples. In the second layer, we equip 2 heads to update each column of $\boldsymbol{\Theta}^{(l)} \in \mathbb{R}^{q \times p}$ and $\boldsymbol{\beta}^{(l)} \in \mathbb{R}^p$.

## 3.4 PRETRAINING AND EXCESS LOSS BOUND

With slightly abuse of notations, we denote the (formulated) training prompt as:

$$
\boldsymbol{H}_k = \begin{bmatrix} \boldsymbol{z}_{1,k} & \cdots & \boldsymbol{z}_{n,k} & \boldsymbol{z}_{n+1,k} \\ \boldsymbol{x}_{1,k} & \cdots & \boldsymbol{x}_{n,k} & \boldsymbol{x}_{n+1,k} \\ y_{1,k} & \cdots & y_{n,k} & 0 \end{bmatrix} \in \mathbb{R}^{(p+q+1) \times (n+1)}, \quad k = 1, \ldots, N.
$$

Note that we denote each training prompt by the subscript $k = 1, \ldots, N$, where $N$ is the total number of prompts. Each training prompt consists of $n$ labeled training samples $\{(\boldsymbol{z}_i, \boldsymbol{x}_i, y_i)\}_{i=1}^n$, and one unlabeled query sample $(\boldsymbol{z}_{n+1}, \boldsymbol{x}_{n+1})$. Our goal is to predict $y_{n+1}$ given the context provided by the prompt.

We introduce the following ICL data generating scheme such that endogeneity occurs in the training samples, but does not extend to the query sample. Each training prompt is generated by the in-context distribution $\mathcal{P}$, described by Algorithm 1.

---

**Algorithm 1** In-Context Distribution $\mathcal{P}$

---

  1: **Parameters:** Sample size n, clipping thresholds $B_z, B_x, B_y$. Task parameters $\boldsymbol{\Theta}, \boldsymbol{\beta}, \boldsymbol{\Phi}, \boldsymbol{\phi}$, $\boldsymbol{\Sigma}_z, \boldsymbol{\Sigma}_u, \boldsymbol{\Sigma}_\omega, \sigma_\epsilon$ from meta distribution $\boldsymbol{\pi}$.
  2: **Output:** Training samples $\{(\boldsymbol{z}_i, \boldsymbol{x}_i, y_i)\}_{i=1}^n$, query sample $(\boldsymbol{z}_{n+1}, \boldsymbol{x}_{n+1}, \boldsymbol{y}_{n+1})$.
  3: **for** $i = 1, \ldots, n$ **do**
  4:     **Generate:** $\boldsymbol{z}_i \sim \mathcal{N}(0, \boldsymbol{\Sigma}_z), \boldsymbol{u}_i \sim \mathcal{N}(0, \boldsymbol{\Sigma}_u), \boldsymbol{\omega}_i \sim \mathcal{N}(0, \boldsymbol{\Sigma}_\omega), \epsilon_i \sim \mathcal{N}(0, \sigma_\epsilon^2)$.
  5:     **Compute:** $\boldsymbol{x}_i = \boldsymbol{\Theta}^\top \boldsymbol{z}_i + \boldsymbol{\Phi}^\top \boldsymbol{u}_i + \boldsymbol{\omega}_i$.
  6:     **Compute:** $y_i = \boldsymbol{\beta}^\top \boldsymbol{x}_i + \boldsymbol{\phi}^\top \boldsymbol{u}_i + \epsilon_i$.
  7: **end for**
  8: **Generate:** $\boldsymbol{z}_{n+1} \sim \mathcal{N}(0, \boldsymbol{\Sigma}_z), \boldsymbol{\omega}_{n+1} \sim \mathcal{N}(0, \boldsymbol{\Sigma}_\omega), \epsilon_{n+1} \sim \mathcal{N}(0, \sigma_\epsilon^2)$.
  9: **Compute:** $\boldsymbol{x}_{n+1} = \boldsymbol{\Theta}^\top \boldsymbol{z}_{n+1} + \boldsymbol{\omega}_{n+1}$.
10: **Compute:** $y_{n+1} = \boldsymbol{\beta}^\top \boldsymbol{x}_{n+1} + \epsilon_{n+1}$.
11: **Clip:** $\boldsymbol{z}_i = \mathsf{clip}_{B_z}(\boldsymbol{z}_i), \boldsymbol{x}_i = \mathsf{clip}_{B_x}(\boldsymbol{x}_i), y_i = \mathsf{clip}_{B_y}(y_i)$ for $i = 1, \ldots, n+1$.

---

In Algorithm 1, $\boldsymbol{u} \in \mathbb{R}^p$ is the source of endogenous error, $\boldsymbol{w} \in \mathbb{R}^p, \epsilon \in \mathbb{R}$ are the exogenous errors. Note that we have $\boldsymbol{\epsilon}_{1,i} = \boldsymbol{\phi}^\top \boldsymbol{u}_i + \epsilon_i$ and $\boldsymbol{\epsilon}_{2,i} = \boldsymbol{\Phi}^\top \boldsymbol{u}_i + \boldsymbol{\omega}_i$, corresponding to the notations in Equation (1). $\boldsymbol{\Theta} \in \mathbb{R}^{q \times p}, \boldsymbol{\beta} \in \mathbb{R}^p, \boldsymbol{\Phi} \in \mathbb{R}^{p \times p}, \boldsymbol{\phi} \in \mathbb{R}^p, \boldsymbol{\Sigma}_z \in \mathbb{R}^{q \times q}, \boldsymbol{\Sigma}_u \in \mathbb{R}^{p \times p}, \boldsymbol{\Sigma}_\omega \in \mathbb{R}^{p \times p}, \sigma_\epsilon \in \mathbb{R}$ are task-specific parameters following meta distribution $\boldsymbol{\pi}$. $\mathsf{clip}_B(\cdot)$ is a clipping operator to bound the norm of input within radius $B$. We say that the in-context samples $\{(\boldsymbol{z}_i, \boldsymbol{x}_i, y_i)\}_{i=1}^{n+1}$ are drawn from the in-context distribution $\mathcal{P}$, and $\mathcal{P} \sim \boldsymbol{\pi}$ if the task parameters $(\boldsymbol{\Theta}, \boldsymbol{\beta}, \boldsymbol{\Phi}, \boldsymbol{\phi}, \boldsymbol{\Sigma}_z, \boldsymbol{\Sigma}_u, \boldsymbol{\Sigma}_\omega, \sigma_\epsilon)$ are sampled from $\boldsymbol{\pi}$. One can check that Assumption 1 and Assumption 2(ii)(iii) are directly satisfied with the data generated from the in-context distribution $\mathcal{P}$.

Following the theoretical framework of (Bai et al., 2023), we define the population ICL loss[2]:

$$
L_{\mathsf{ICL}}(\boldsymbol{\theta}) = \mathbb{E}_\pi \mathbb{E}_\mathcal{P}[y_{n+1} - \mathsf{clip}_{B_y}(\mathsf{read}_y(\mathsf{TF}_{\boldsymbol{\theta}}^R(\boldsymbol{H}^{(0)})))]^2, \tag{10}
$$

---

[2]All the clipping operations are only for analytical purpose. In practice, the behavior of the trained transformer is consistent even without the clipping bounds.

where $\boldsymbol{H}^{(0)}$ is the embedded input as defined in Equation (9), $\mathsf{TF}_{\boldsymbol{\theta}}^{R}$ is the transformer model with parameter $\boldsymbol{\theta}$ and clipping operation $\mathsf{clip}_R(\cdot)$ applied to each layer output. For simplicity, we denote $\widetilde{\mathsf{TF}}_{\boldsymbol{\theta}}(\boldsymbol{H}) := \mathsf{clip}_{B_y}(\mathsf{read}_y(\mathsf{TF}_{\boldsymbol{\theta}}^{R}(\boldsymbol{H}^{(0)})))$.

The transformer is trained to minimize the in-context loss in Equation (10) with the following empirical loss:

$$\hat{L}_{\mathsf{ICL}}(\boldsymbol{\theta}) = \frac{1}{N} \sum_{k=1}^{N} (y_{n+1,k} - \widetilde{\mathsf{TF}}_{\boldsymbol{\theta}}(\boldsymbol{H}_k))^2. \tag{11}$$

We consider the following constrained optimization problem:

$$\hat{\boldsymbol{\theta}} := \underset{\boldsymbol{\theta} \in \boldsymbol{\vartheta}_{L,M,D',B_\theta}}{\arg\min} \quad \hat{L}_{\mathsf{ICL}}(\boldsymbol{\theta}),$$
$$\boldsymbol{\vartheta}_{L,M,D',B_\theta} := \{\boldsymbol{\theta} = (\boldsymbol{\theta}_{\mathsf{Attn}}^{(1:L)}, \boldsymbol{\theta}_{\mathsf{MLP}}^{(1:L)}) : \max_{l \in [L]} M^{(l)} \le M, \max_{l \in [L]} D^{(l)} \le D', \|\boldsymbol{\theta}\| \le B_\theta\}, \tag{12}$$

where $\|\boldsymbol{\theta}\| := \max_{l \in [L]} \{ \max_{m \in [M]} \{\|\boldsymbol{Q}_m^{(l)}\|, \|\boldsymbol{K}_m^{(l)}\|\} + \sum_{m=1}^{M} \|\boldsymbol{V}_m^{(l)}\| + \|\boldsymbol{W}_1^{(l)}\| + \|\boldsymbol{W}_2^{(l)}\|\}$.

We now establish excess loss bound for the trained transformer model.

**Theorem 3.3** (Excess loss bound for in-context pretrained transformer)**.** *Suppose condition (i) in Assumption 2 holds and the meta distribution $\boldsymbol{\pi}$ satisfies the following conditions:*

$$\mathbb{E}_{\boldsymbol{\pi}} \left[ \boldsymbol{\phi}^{\top} \boldsymbol{\Sigma}_u \boldsymbol{\phi} + \sigma_{\epsilon}^2 \right] \le \tilde{\sigma}^2 \text{ and } \mathbb{E}_{\boldsymbol{\pi}} \left[ \sigma_{\epsilon}^2 \right] \le \tilde{\sigma}_{\epsilon}^2. \tag{13}$$

*Let the in-context distribution $\mathcal{P} \sim \boldsymbol{\pi}$ such that the samples $(\boldsymbol{z}_i, \boldsymbol{x}_i, y_i)_{i=1}^{n+1}$ are drawn independently from $\mathcal{P}$ (ref. Algorithm 1). With training prompts $\boldsymbol{H}_k, k = 1, \dots, N$, under ICL loss (10), the trained transformer (12) with $L = 2\bar{L} + 1, M = 2(p + 1), D = qp + 3p + q + 3, D' = 0$ (attention-only) achieves the following excess loss with probability at least $1 - \zeta$:*

$$L_{\mathsf{ICL}}(\hat{\boldsymbol{\theta}}) - \mathbb{E}_{\boldsymbol{\pi}} \mathbb{E}_{\mathcal{P}} \left[ (y_{n+1} - \langle \boldsymbol{\beta}, \boldsymbol{x}_{n+1} \rangle)^2 \right] \le \mathcal{O}\left( (\Lambda^{\star})^{\bar{L}} \left( B_x^2 \sqrt{\frac{q}{n} \left( \frac{B_\beta^2}{K} + C^2(n)\tilde{\sigma}^2 \right)} + B_x \tilde{\sigma}_\epsilon \right) \right.$$

$$\left. + B_x^2 \left( \frac{q}{n} \left( \frac{B_\beta^2}{K} + C^2(n)\tilde{\sigma}^2 \right) + \mu_{\Lambda,2}^{\star} \right) + B_y^2 \sqrt{\frac{L^2 M D^2 \log(2 + \max\{B_\theta, R, B_y\}) + \log(1/\zeta)}{N}} \right),$$

*where $\Lambda^{\star} := \min_{\alpha, \eta} \mathbb{E}_{\boldsymbol{\pi}} \mathbb{E}_{\mathcal{P}}[\Lambda | \boldsymbol{H}, \alpha, \eta] < 1$, and $\mu_{\Lambda,2}^{\star} := \mathbb{E}_{\boldsymbol{\pi}} \mathbb{E}_{\mathcal{P}}[\Lambda^{2\bar{L}} | \boldsymbol{H}, \alpha^{\star}, \eta^{\star}]$ is close to 0.*

In practical training, the number of prompts $N$ is usually large enough such that the last term of the above bound is negligible. Thus, given a meta distribution $\boldsymbol{\pi}$, the excess loss is dominated by two factors: (i) number of attention layers, and (ii) number of in-context samples. The proof of Theorem 3.3 is provided in Appendix B.3.

### 3.5 EXTRACTING THE REGRESSION COEFFICIENTS

The primary goal of IV regression is to estimate the causal effect, i.e. the coefficient $\boldsymbol{\beta}$ under the stated endogeneity in Equation (1). For 2SLS, the estimated causal effect is given by the coefficients of the endogenous variable in the second stage regression (2). For transformer models, we propose a straightforward method to extract these estimated coefficients by differentiating the output with respect to each dimension of the endogenous variable. The specific approach is summarized in Algorithm 2. We observe that the choice of $\Delta$ within a reasonable range does not significantly affect the estimation of the coefficients. In practice, usually a slightly larger $\Delta$ (for example $\Delta = 5$) can lead to a more stable estimation, which is possibly due to the elimination of rounding errors during computation.

## 4 EXPERIMENTS

### 4.1 EXPERIMENT SETUP

We conduct a simulation study to evaluate the performance of the ICL-pretrained transformer model in handling endogeneity. We set the maximum input sample size to 51 ($n = 50$ training samples and

---

**Algorithm 2** Extracting the regression coefficients

---

1: **Input:** Trained transformer model $\mathsf{TF}_{\hat{\theta}}$, input matrix $\boldsymbol{H}$, perturbation $\Delta$.
2: **Output:** Estimated coefficient $\hat{\boldsymbol{\beta}}$.
3: **Procedure:**
4: Compute the output of the transformer model: $\hat{\boldsymbol{Y}} = \widetilde{\mathsf{TF}}_{\hat{\theta}}(\boldsymbol{H})$.
5: **for** each dimension $k = 1, \ldots, p$ **do**
6:     Copy $\boldsymbol{H}_{\Delta(k)} = \boldsymbol{H}$. Set the $k$-th dimension of $\boldsymbol{x}_{n+1}$ to be $(\boldsymbol{x}_{n+1})_k + \Delta$ for $\boldsymbol{H}_{\Delta(k)}$.
7:     Compute the new output value: $\hat{\boldsymbol{Y}}_{\Delta(k)} = \widetilde{\mathsf{TF}}_{\hat{\theta}}(\boldsymbol{H}_{\Delta(k)})$.
8:     Compute the estimated coefficient: $\hat{\beta}_k = \frac{\hat{\boldsymbol{Y}}_{\Delta(k)} - \hat{\boldsymbol{Y}}}{\Delta}$.
9: **end for**

---

one query sample), the dimension of endogenous variable $p = 5$, and the dimension of instrument $q = 10$. The training prompts are generated using Algorithm 1, with task parameters $\boldsymbol{\Theta}, \boldsymbol{\beta}, \boldsymbol{\Phi}, \boldsymbol{\phi}$ sampled from standard Gaussian distribution, and the covariance matrices $\boldsymbol{\Sigma}_z, \boldsymbol{\Sigma}_u, \boldsymbol{\Sigma}_\omega$ set to be identity matrices. The noise level $\sigma_\epsilon$ is set to 1. We ignore all the clipping bounds in the experiment $(B_\beta, B_\Theta, B_z, B_x, B_y, B_\theta, R$ set to infinity).

The backbone of the transformer block is initialized using GPT-2 settings, with 12 attention heads $(M = 12)$, 80-dimensional embedding space $(D = 80)$ and 2 layers $(L_0 = 2)$, following the theoretical guidelines in Theorem 3.2. We employ the looped transformer architecture, consisting of 10 identical cascading transformer blocks. The transformer model is trained under the ICL loss (11) with a batch size of $N = 64$, over a total of 300,000 training steps.

We evaluate the trained transformer model on test prompts that are not included during training. As benchmarks, we compare the transformer's performance against the 2SLS and the OLS estimators, which are obtained by directly fitting the training samples $\{(\boldsymbol{z}_i, \boldsymbol{x}_i, y_i)\}_{i=1}^n$ within the text prompts. In contrast, the same trained transformer model is used without any parameter adjustments for each task. We compare the performance of these models from two aspects: the in-context prediction error (ICPE) on the query sample $y_{n+1}$, and the mean squared error (MSE) on the coefficient $\boldsymbol{\beta}$.

## 4.2 RESULTS

We first investigate the performance of the trained transformer model over endogeneity tasks with varying training sample sizes from 20 to 50. The results are shown in Figure 1a. Under endogeneity, our transformer model achieves similar performance to that of the 2SLS estimator, with only small gaps in ICPE and MSE, both outperforming the OLS estimator.

Next, we examine the performance of the trained transformer model in handling varying levels of IV strength. The strength of an instrument is measured by the correlation between the IV and the endogenous variable. To vary the IV strength, we generate prompts with $\boldsymbol{z}_i$ and $\boldsymbol{x}_i$ following different correlation levels. Specifically, in Algorithm 1, we adjust the IV strength by multiplying $\boldsymbol{\Theta}$ by a factor $r \in (0, 2)$ when generating test prompts. The results are shown in Figure 1b.

Interestingly, the trained transformer model outperforms the 2SLS estimator in handling weaker IVs (when IV strength $< 0.5$). This suggests that, beyond merely mimicking 2SLS, the ICL training process may equip the transformer model with a more advanced mechanism for handling endogeneity with weak IVs than the 2SLS estimator. At the same time, when the IV is strong, the transformer model maintains performance comparable to that of the 2SLS estimator.

This finding motivates us to further examine the performance of the trained transformer model in non-standard endogeneity tasks. We consider two scenarios: (a) the IV has a quadratic effect on the endogenous variable, i.e. $\boldsymbol{x}_{i,k} = \boldsymbol{\Theta}_k^\top \boldsymbol{z}_{i,k}^2 + \mathsf{error}_{i,k}$ in Algorithm 1, and (b) the dimension of IV is not sufficient to identify the endogenous variable[3], where we set $q = 3$ (by zeroing out the remaining dimensions of $\boldsymbol{z}$ in test prompts) and $p = 5$.

We evaluate the same trained transformer model as before, with results presented in Figure 2a and Figure 2b, respectively. Once again, the trained transformer model consistently outperforms both

---

[3]For 2SLS estimate, the actual computation uses pseudoinverse to handle rank deficiency.

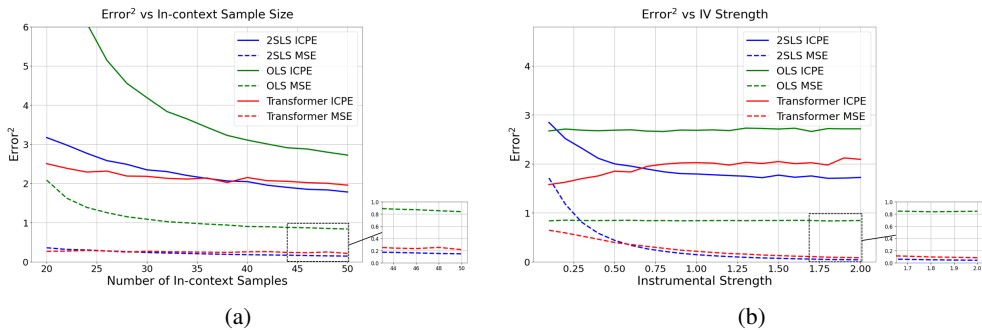

(a)                                            (b)

Figure 1: The ICL performance of the trained transformer model in endogeneity tasks. We compare in-context prediction error (ICPE) and coefficient MSE versus (a) the number of in-context samples; (b) the IV strength. The curves are averaged over 500 simulations.

2SLS and OLS estimators in handling these non-standard endogeneity tasks. All these results suggest that the trained transformer can be generalized effectively to a broader range of endogeneity tasks while still providing reliable in-context predictions and coefficient estimates. To further illustrate this capability, we also examine other cases including multicollinearity, complex non-linear IV, and varying endogeneity strengths, see Appendix C.3,C.4,C.5. We suspect that, in our pretraining scheme, although the 2SLS estimator already achieves small excess loss, a gap remains between the 2SLS estimator and the optimal predictor that the transformer model successfully bridges. Finally, we conclude that through ICL training, the transformer model performs at least as well as 2SLS and appears to be a promising tool for handling endogeneity in difficult scenarios.

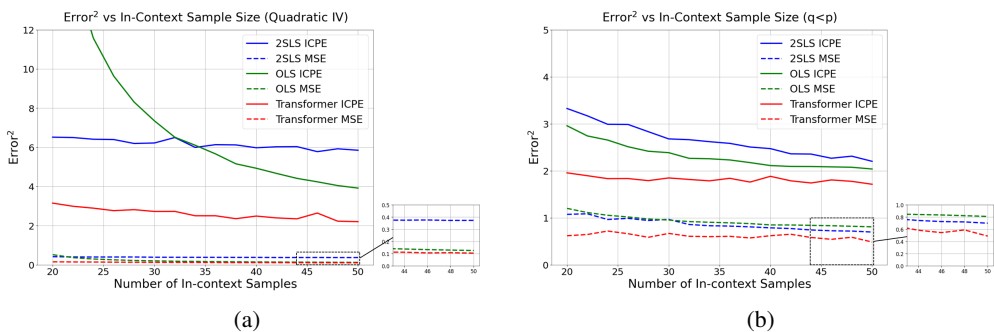

(a)                                            (b)

Figure 2: The ICL performance of the trained transformer model in non-standard endogeneity tasks: (a) The IV has quadratic effect on the endogenous variable; (b) The dimension of IV is not sufficient to identify the endogenous variable. The curves are averaged over 500 simulations.

## 5 CONCLUSION

This paper presents a novel perspective on the transformer model in its ability to handle endogeneity in in-context linear regression. We have theoretically shown that the transformer model exists an intrinsic structure that enables it to learn the 2SLS algorithm through an efficient GD procedure. We have further provided a theoretical guarantee that the trained transformer model can achieve a small excess loss over the optimal loss, under our proposed ICL training scheme. Our simulation study demonstrates that the trained transformer model can achieve comparable performance to the 2SLS estimator in handling standard endogeneity tasks. Furthermore, our investigation illustrates that it exhibits significantly better performances in handling complex scenarios such as weak instruments, non-linear IV, and underdetermined IV problems, compared to the 2SLS estimator. These results suggest that the ICL pre-trained transformer model is a promising tool for making reliable in-context predictions and coefficient estimates under endogeneity, especially when dealing with non-standard IV problems.

ACKNOWLEDGEMENTS

The work of H. Liang and L. Lai was supported by National Science Foundation (NSF) under grants CCF-2112504 and CCF-2232907. The work of K. Balasubramanian was supported by NSF under grants DMS-2413426 and DMS-2053918.

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

# A PROOFS FOR SECTION 2

## A.1 PROOF OF THEOREM 2.1

We first introduce the following lemmas that are used in the proof of Theorem 2.1.

**Lemma A.1** (Bernstein Inequality, from Theorem 6.1.1 in Tropp (2015))**.** Let $\boldsymbol{S}_1, \ldots, \boldsymbol{S}_n$ be independent, centered random matrices with common dimension $d_1 \times d_2$, and assume that each one is almost surely bounded:

$$\mathbb{E}[\boldsymbol{S}_i] = \mathbf{0}, \mathbb{P}(\|\boldsymbol{S}_i\| \leq b) = 1, \quad \forall i = 1, \ldots, n.$$

With the sum:

$$\boldsymbol{\Omega} = \sum_{i=1}^n \boldsymbol{S}_i,$$

and the matrix variance statistic of the sum:

$$\nu(\boldsymbol{\Omega}) := \max \left\{ \left\| \mathbb{E}(\boldsymbol{\Omega}\boldsymbol{\Omega}^\top) \right\|, \left\| \mathbb{E}(\boldsymbol{\Omega}^\top\boldsymbol{\Omega}) \right\| \right\},$$

then the following inequality holds:

$$\mathbb{P}\{\|\boldsymbol{\Omega}\| \geq \varepsilon\} \leq (d_1 + d_2) \cdot \exp\left(\frac{-\varepsilon^2/2}{\nu(\boldsymbol{\Omega}) + b\varepsilon/3}\right) \text{ for any } \varepsilon \geq 0.$$

**Lemma A.2** (Inverse Convergence, adapted from Lemma 2.1 in Jin et al. (2024))**.** Suppose we have a random invertible matrix $\boldsymbol{\Omega}$ and invertible matrix sequence $\{\hat{\boldsymbol{\Omega}}^{(n)}\}$ such that $\hat{\boldsymbol{\Omega}}^{(n)} \xrightarrow{\mathsf{P}} \boldsymbol{\Omega}$. If there exists a constant $\tilde{\lambda} > 0$ such that $\sigma_{\min}(\hat{\boldsymbol{\Omega}}) \geq \tilde{\lambda}$ almost surely, then it holds that:

$$(\hat{\boldsymbol{\Omega}}^{(n)})^{-1} \xrightarrow{\mathsf{P}} \boldsymbol{\Omega}^{-1}.$$

Further, given convergence rate

$$\mathbb{P}\left\{ \left\| \hat{\boldsymbol{\Omega}}^{(n)} - \boldsymbol{\Omega} \right\| \geq \varepsilon \right\} \leq \xi(n, \varepsilon),$$

then:

$$\mathbb{P}\left\{ \left\| (\hat{\boldsymbol{\Omega}}^{(n)})^{-1} - \boldsymbol{\Omega}^{-1} \right\| \geq \varepsilon \right\} \leq \xi(n, \tilde{\lambda}^2\varepsilon).$$

*Proof.* We have the following decomposition:

$$(\hat{\boldsymbol{\Omega}}^{(n)})^{-1} - \boldsymbol{\Omega}^{-1} = (\hat{\boldsymbol{\Omega}}^{(n)})^{-1}(\boldsymbol{\Omega} - \hat{\boldsymbol{\Omega}}^{(n)})\boldsymbol{\Omega}^{-1}.$$

It follows that:

$$\left\| (\hat{\boldsymbol{\Omega}}^{(n)})^{-1} - \boldsymbol{\Omega}^{-1} \right\| \leq \left\| (\hat{\boldsymbol{\Omega}}^{(n)})^{-1} \right\| \left\| \boldsymbol{\Omega} - \hat{\boldsymbol{\Omega}}^{(n)} \right\| \left\| \boldsymbol{\Omega}^{-1} \right\|$$

$$\leq \frac{1}{\tilde{\lambda}^2} \left\| \boldsymbol{\Omega} - \hat{\boldsymbol{\Omega}}^{(n)} \right\|.$$

Then

$$\mathbb{P}\left\{ \left\| (\hat{\boldsymbol{\Omega}}^{(n)})^{-1} - \boldsymbol{\Omega}^{-1} \right\| \geq \varepsilon \right\} \leq \mathbb{P}\left\{ \frac{1}{\tilde{\lambda}^2} \left\| \boldsymbol{\Omega} - \hat{\boldsymbol{\Omega}}^{(n)} \right\| \geq \varepsilon \right\}$$

$$\leq \xi(n, \tilde{\lambda}^2\varepsilon).$$

□

**Lemma A.3** (Product Convergence)**.** Let $\{\hat{\boldsymbol{\Omega}}_1^{(n)}\}, \{\hat{\boldsymbol{\Omega}}_2^{(n)}\}, \ldots, \{\hat{\boldsymbol{\Omega}}_K^{(n)}\}$ be $K$ sequences of matrices such that $\hat{\boldsymbol{\Omega}}_1^{(n)} \xrightarrow{\mathsf{P}} \boldsymbol{\Omega}_1, \hat{\boldsymbol{\Omega}}_2^{(n)} \xrightarrow{\mathsf{P}} \boldsymbol{\Omega}_2, \ldots, \hat{\boldsymbol{\Omega}}_K^{(n)} \xrightarrow{\mathsf{P}} \boldsymbol{\Omega}_K$, where each $\|\hat{\boldsymbol{\Omega}}_k^{(n)}\|$ is almost surely bounded for every $k = 1, \ldots, K$. If the dimensions match, then it holds that:

$$\hat{\boldsymbol{\Omega}}_1^{(n)}\hat{\boldsymbol{\Omega}}_2^{(n)} \cdots \hat{\boldsymbol{\Omega}}_K^{(n)} \xrightarrow{\mathsf{P}} \boldsymbol{\Omega}_1\boldsymbol{\Omega}_2 \cdots \boldsymbol{\Omega}_K.$$

Further, given convergence rates:

$$\mathbb{P}\left\{\left\|\hat{\boldsymbol{\Omega}}_1^{(n)} - \boldsymbol{\Omega}_1\right\| \geq \varepsilon\right\} \leq \xi_1(n, \varepsilon),$$

$$\mathbb{P}\left\{\left\|\hat{\boldsymbol{\Omega}}_2^{(n)} - \boldsymbol{\Omega}_2\right\| \geq \varepsilon\right\} \leq \xi_2(n, \varepsilon),$$

$$\vdots$$

$$\mathbb{P}\left\{\left\|\hat{\boldsymbol{\Omega}}_K^{(n)} - \boldsymbol{\Omega}_K\right\| \geq \varepsilon\right\} \leq \xi_K(n, \varepsilon),$$

then it holds that:

$$\mathbb{P}\left\{\left\|\hat{\boldsymbol{\Omega}}_1^{(n)}\hat{\boldsymbol{\Omega}}_2^{(n)}\cdots\hat{\boldsymbol{\Omega}}_K^{(n)} - \boldsymbol{\Omega}_1\boldsymbol{\Omega}_2\cdots\boldsymbol{\Omega}_K\right\| \geq \varepsilon\right\} \leq \sum_{i=1}^{K} \xi_i\left(n, \frac{\varepsilon}{K\prod_{k\neq i}^{K} M_k}\right), \qquad (14)$$

where $M_k$ is an upper bound such that $\|\hat{\boldsymbol{\Omega}}_k^{(n)}\| \leq M_k$ almost surely, $\forall k = 1, \ldots, K$.

*Proof.* We begin by showing the case of $K = 2$. By the triangle inequality, we have:

$$\left\|\hat{\boldsymbol{\Omega}}_1^{(n)}\hat{\boldsymbol{\Omega}}_2^{(n)} - \boldsymbol{\Omega}_1\boldsymbol{\Omega}_2\right\| \leq \left\|\hat{\boldsymbol{\Omega}}_1^{(n)}\hat{\boldsymbol{\Omega}}_2^{(n)} - \boldsymbol{\Omega}_1\hat{\boldsymbol{\Omega}}_2^{(n)}\right\| + \left\|\boldsymbol{\Omega}_1\hat{\boldsymbol{\Omega}}_2^{(n)} - \boldsymbol{\Omega}_1\boldsymbol{\Omega}_2\right\|$$

$$\leq \left\|\hat{\boldsymbol{\Omega}}_2^{(n)}\right\|\left\|\hat{\boldsymbol{\Omega}}_1^{(n)} - \boldsymbol{\Omega}_1\right\| + \left\|\hat{\boldsymbol{\Omega}}_2^{(n)} - \boldsymbol{\Omega}_2\right\|\|\boldsymbol{\Omega}_1\|$$

$$\leq M_2 \left\|\hat{\boldsymbol{\Omega}}_1^{(n)} - \boldsymbol{\Omega}_1\right\| + M_1 \left\|\hat{\boldsymbol{\Omega}}_2^{(n)} - \boldsymbol{\Omega}_2\right\|.$$

Using the union bound, we have:

$$\mathbb{P}\left\{\left\|\hat{\boldsymbol{\Omega}}_1^{(n)}\hat{\boldsymbol{\Omega}}_2^{(n)} - \boldsymbol{\Omega}_1\boldsymbol{\Omega}_2\right\| \geq \varepsilon\right\}$$

$$\leq \mathbb{P}\left\{M_2 \left\|\hat{\boldsymbol{\Omega}}_1^{(n)} - \boldsymbol{\Omega}_1\right\| + M_1 \left\|\hat{\boldsymbol{\Omega}}_2^{(n)} - \boldsymbol{\Omega}_2\right\| \geq \varepsilon\right\}$$

$$\leq \mathbb{P}\left\{M_2 \left\|\hat{\boldsymbol{\Omega}}_1^{(n)} - \boldsymbol{\Omega}_1\right\| \geq \varepsilon/2\right\} + \mathbb{P}\left\{M_1 \left\|\hat{\boldsymbol{\Omega}}_2^{(n)} - \boldsymbol{\Omega}_2\right\| \geq \varepsilon/2\right\}$$

$$\leq \xi_1\left(n, \frac{\varepsilon}{2M_2}\right) + \xi_2\left(n, \frac{\varepsilon}{2M_1}\right).$$

For any $K > 2$, suppose the statement (14) holds for $k = 2, \ldots, K - 1$. Observe that:

$$\left\|\hat{\boldsymbol{\Omega}}_1^{(n)}\hat{\boldsymbol{\Omega}}_2^{(n)}\cdots\hat{\boldsymbol{\Omega}}_K^{(n)} - \boldsymbol{\Omega}_1\boldsymbol{\Omega}_2\cdots\boldsymbol{\Omega}_K\right\|$$

$$\leq \left\|\hat{\boldsymbol{\Omega}}_1^{(n)}\hat{\boldsymbol{\Omega}}_2^{(n)}\cdots\hat{\boldsymbol{\Omega}}_K^{(n)} - \boldsymbol{\Omega}_1\boldsymbol{\Omega}_2\cdots\boldsymbol{\Omega}_{K-1}\hat{\boldsymbol{\Omega}}_K^{(n)}\right\| + \left\|\boldsymbol{\Omega}_1\boldsymbol{\Omega}_2\cdots\boldsymbol{\Omega}_{K-1}\hat{\boldsymbol{\Omega}}_K^{(n)} - \boldsymbol{\Omega}_1\boldsymbol{\Omega}_2\cdots\boldsymbol{\Omega}_K\right\|$$

$$\leq M_K \left\|\hat{\boldsymbol{\Omega}}_1^{(n)}\hat{\boldsymbol{\Omega}}_2^{(n)}\cdots\hat{\boldsymbol{\Omega}}_{K-1}^{(n)} - \boldsymbol{\Omega}_1\boldsymbol{\Omega}_2\cdots\boldsymbol{\Omega}_{K-1}\right\| + \prod_{k=1}^{K-1} M_k \left\|\hat{\boldsymbol{\Omega}}_K^{(n)} - \boldsymbol{\Omega}_K\right\|.$$

$$(15)$$

Then it follows that:

$$\mathbb{P}\left\{\left\|\hat{\boldsymbol{\Omega}}_1^{(n)}\hat{\boldsymbol{\Omega}}_2^{(n)}\cdots\hat{\boldsymbol{\Omega}}_K^{(n)}-\boldsymbol{\Omega}_1\boldsymbol{\Omega}_2\cdots\boldsymbol{\Omega}_K\right\|\geq\varepsilon\right\}$$

$$\leq\mathbb{P}\left\{M_K\left\|\hat{\boldsymbol{\Omega}}_1^{(n)}\hat{\boldsymbol{\Omega}}_2^{(n)}\cdots\hat{\boldsymbol{\Omega}}_{K-1}^{(n)}-\boldsymbol{\Omega}_1\boldsymbol{\Omega}_2\cdots\boldsymbol{\Omega}_{K-1}\right\|+\prod_{k=1}^{K-1}M_k\left\|\hat{\boldsymbol{\Omega}}_K^{(n)}-\boldsymbol{\Omega}_K\right\|\geq\varepsilon\right\}$$

$$\leq\mathbb{P}\left\{M_K\left\|\hat{\boldsymbol{\Omega}}_1^{(n)}\hat{\boldsymbol{\Omega}}_2^{(n)}\cdots\hat{\boldsymbol{\Omega}}_{K-1}^{(n)}-\boldsymbol{\Omega}_1\boldsymbol{\Omega}_2\cdots\boldsymbol{\Omega}_{K-1}\right\|\geq\frac{K-1}{K}\varepsilon\right\}$$

$$+\mathbb{P}\left\{\prod_{k=1}^{K-1}M_k\left\|\hat{\boldsymbol{\Omega}}_K^{(n)}-\boldsymbol{\Omega}_K\right\|\geq\frac{1}{K}\varepsilon\right\}$$

$$\leq\sum_{i=1}^{K-1}\xi_i\left(n,\frac{\varepsilon}{KM_K\prod_{k\neq i}^{K-1}M_k}\right)+\xi_K\left(n,\frac{\varepsilon}{K\prod_{k=1}^{K-1}M_k}\right)$$

$$=\sum_{i=1}^{K}\xi_i\left(n,\frac{\varepsilon}{K\prod_{k\neq i}^{K}M_k}\right).$$

Thus, by induction, the proof is complete. $\qquad\square$

**Remark A.1.** In Lemma A.3, consider the special case where $\boldsymbol{\Omega}_1=\boldsymbol{0}$. Then the inequality (15) can be simplified as follows:

$$\left\|\hat{\boldsymbol{\Omega}}_1^{(n)}\hat{\boldsymbol{\Omega}}_2^{(n)}\cdots\hat{\boldsymbol{\Omega}}_K^{(n)}-\boldsymbol{0}\right\|\leq\prod_{k=2}^{K}M_k\left\|\hat{\boldsymbol{\Omega}}_1^{(n)}\right\|.$$

And we have the following simplified form:

$$\mathbb{P}\left\{\left\|\hat{\boldsymbol{\Omega}}_1^{(n)}\hat{\boldsymbol{\Omega}}_2^{(n)}\cdots\hat{\boldsymbol{\Omega}}_K^{(n)}-\boldsymbol{0}\right\|\geq\varepsilon\right\}\leq\mathbb{P}\left\{\prod_{k=2}^{K}M_k\left\|\hat{\boldsymbol{\Omega}}_1^{(n)}\right\|\geq\varepsilon\right\}$$

$$\leq\xi_1\left(n,\frac{\varepsilon}{\prod_{k=2}^{K}M_k}\right).$$

**Lemma A.4** (Deviation Inequality for Minimum Eigenvalue of Projected Sample Covariance Matrix). Suppose Assumption 2 holds.

When $n\geq\max\left\{\frac{qe^{\frac{3}{2}}}{K},\frac{p^2(q+1)^2K}{qK_0^2}\right\}$, the following inequality holds with probability at least $1-\frac{3qe^{\frac{1}{2}}}{Kn}$:

$$\lambda_{\min}\left(\frac{1}{n}\boldsymbol{X}^\top\boldsymbol{P}_Z\boldsymbol{X}\right)\geq\lambda_z\left(\sigma_{\min}(\boldsymbol{\Theta})-\sqrt{\frac{2p(q+1)B_{\epsilon_2}^2\log(\frac{K}{q}n)}{\lambda_{\min}(\boldsymbol{\Sigma}_z)n}}\right)^2:=\lambda_{\tilde{x}},$$

where $K:=\frac{\lambda_{\min}(\boldsymbol{\Sigma}_z)}{6B_z^2}$, $K_0:=\frac{\lambda_{\min}(\boldsymbol{\Sigma}_z)\sigma_{\min}^2(\boldsymbol{\Theta})}{2B_{\epsilon_2}^2}$, $\boldsymbol{\Sigma}_z:=\mathbb{E}[\boldsymbol{z}\boldsymbol{z}^\top]$, $\boldsymbol{P}_Z:=\boldsymbol{Z}(\boldsymbol{Z}^\top\boldsymbol{Z})^{-1}\boldsymbol{Z}^\top$, and $\lambda_z$ is a lower bound of $\lambda_{\min}(\frac{\boldsymbol{Z}^\top\boldsymbol{Z}}{n})$.

*Proof.* Let

$$\boldsymbol{E}_{\parallel}:=\boldsymbol{P}_Z\boldsymbol{\mathcal{E}}_2,\boldsymbol{E}_{\perp}:=(\boldsymbol{I}-\boldsymbol{P}_Z)\boldsymbol{\mathcal{E}}_2.$$

We have the following decomposition:

$$\boldsymbol{X}=\boldsymbol{Z}\boldsymbol{\Theta}+\boldsymbol{\mathcal{E}}_2=\boldsymbol{Z}\boldsymbol{\Theta}+\boldsymbol{E}_{\parallel}+\boldsymbol{E}_{\perp}=\boldsymbol{Z}(\boldsymbol{\Theta}+(\boldsymbol{Z}^T\boldsymbol{Z})^{-1}\boldsymbol{Z}^T\boldsymbol{\mathcal{E}}_2)+\boldsymbol{E}_{\perp}.$$

Let $\boldsymbol{\Psi}:=(\boldsymbol{Z}^\top\boldsymbol{Z})^{-1}\boldsymbol{Z}^\top\boldsymbol{\mathcal{E}}_2$. Since $\boldsymbol{P}_Z\boldsymbol{E}_{\perp}=\boldsymbol{0}$, we have

$$\boldsymbol{X}^\top\boldsymbol{P}_Z\boldsymbol{X}=(\boldsymbol{Z}(\boldsymbol{\Theta}+\boldsymbol{\Psi}))^\top\boldsymbol{P}_Z(\boldsymbol{Z}(\boldsymbol{\Theta}+\boldsymbol{\Psi}))$$

$$=(\boldsymbol{\Theta}+\boldsymbol{\Psi})^\top\boldsymbol{Z}^\top\boldsymbol{Z}(\boldsymbol{\Theta}+\boldsymbol{\Psi}).$$

We can now write:

$$\lambda_{\min}\left(\frac{1}{n}\boldsymbol{X}^\top\boldsymbol{P}_Z\boldsymbol{X}\right) = \lambda_{\min}\left((\boldsymbol{\Theta}+\boldsymbol{\Psi})^\top\frac{\boldsymbol{Z}^\top\boldsymbol{Z}}{n}(\boldsymbol{\Theta}+\boldsymbol{\Psi})\right). \tag{16}$$

Note that in general, for a positive semi-definite matrix $\boldsymbol{A}$, we have

$$\lambda_{\min}(\boldsymbol{A}) = \min_{\boldsymbol{u}\neq\boldsymbol{0}}\frac{\boldsymbol{u}^\top\boldsymbol{A}\boldsymbol{u}}{\boldsymbol{u}^\top\boldsymbol{u}},$$

and

$$\begin{aligned}
\lambda_{\min}(\boldsymbol{B}^\top\boldsymbol{A}\boldsymbol{B}) &= \min_{\boldsymbol{u}\neq\boldsymbol{0}}\frac{(\boldsymbol{B}\boldsymbol{u})^\top\boldsymbol{A}\boldsymbol{B}\boldsymbol{u}}{\boldsymbol{u}^\top\boldsymbol{u}} \\
&\geq \min_{\boldsymbol{u}\neq\boldsymbol{0}}\lambda_{\min}(\boldsymbol{A})\frac{(\boldsymbol{B}\boldsymbol{u})^\top\boldsymbol{B}\boldsymbol{u}}{\boldsymbol{u}^\top\boldsymbol{u}} \\
&= \lambda_{\min}(\boldsymbol{A})\lambda_{\min}(\boldsymbol{B}^\top\boldsymbol{B}).
\end{aligned}$$

Thus, from Equation (16), with probability at least $1-\xi$, we have:

$$\begin{aligned}
\lambda_{\min}\left(\frac{1}{n}\boldsymbol{X}^\top\boldsymbol{P}_Z\boldsymbol{X}\right) &= \lambda_{\min}\left((\boldsymbol{\Theta}+\boldsymbol{\Psi})^\top\frac{\boldsymbol{Z}^\top\boldsymbol{Z}}{n}(\boldsymbol{\Theta}+\boldsymbol{\Psi})\right) \\
&\geq \lambda_z\lambda_{\min}\left((\boldsymbol{\Theta}+\boldsymbol{\Psi})^\top(\boldsymbol{\Theta}+\boldsymbol{\Psi})\right).
\end{aligned}$$

It now remains to bound $\lambda_{\min}\left((\boldsymbol{\Theta}+\boldsymbol{\Psi})^\top(\boldsymbol{\Theta}+\boldsymbol{\Psi})\right) = \sigma_{\min}^2(\boldsymbol{\Theta}+\boldsymbol{\Psi})$.

From (Hsu et al., 2014), for each $k\in[p]$ and any given $t>1$, with sample size satisfying

$$n \geq \frac{6B_z^2(\log q + t)}{\lambda_{\min}(\boldsymbol{\Sigma}_z)}, \tag{17}$$

we have the following holds with probability at least $1-3e^{-t}$:

$$\|\boldsymbol{\Psi}_k\|_{\boldsymbol{\Sigma}_z}^2 = \|\hat{\boldsymbol{\Theta}}_k - \boldsymbol{\Theta}_k\|_{\boldsymbol{\Sigma}_z}^2 \leq \frac{B_{\epsilon_2}^2\left(q + 2\sqrt{qt} + 2t\right)}{n} < \frac{B_{\epsilon_2}^2\left[q + 2(q+1)t\right]}{n}.$$

Note that

$$\|\boldsymbol{\Psi}_k\|_{\boldsymbol{\Sigma}_z}^2 = \boldsymbol{\Psi}_k^\top\boldsymbol{\Sigma}_z\boldsymbol{\Psi}_k = \boldsymbol{\Psi}_k^\top\boldsymbol{U}\boldsymbol{\Lambda}_z\boldsymbol{U}^\top\boldsymbol{\Psi}_k = \sum_{i=1}^q\lambda_{z,i}(\boldsymbol{U}^\top\boldsymbol{\Psi}_k)_i^2,$$

and

$$\|\boldsymbol{\Psi}_k\| = \boldsymbol{\Psi}_k^\top\boldsymbol{\Psi}_k = \boldsymbol{\Psi}_k^\top\boldsymbol{U}\boldsymbol{U}^\top\boldsymbol{\Psi}_k = \sum_{i=1}^q(\boldsymbol{U}^\top\boldsymbol{\Psi}_k)_i^2.$$

We have

$$\lambda_{\min}(\boldsymbol{\Sigma}_z)\|\boldsymbol{\Psi}_k\|^2 \leq \|\boldsymbol{\Psi}_k\|_{\boldsymbol{\Sigma}_z}^2 \leq \lambda_{\max}(\boldsymbol{\Sigma}_z)\|\boldsymbol{\Psi}_k\|^2.$$

Then

$$\|\boldsymbol{\Psi}\| \leq \|\boldsymbol{\Psi}\|_F = \sqrt{\sum_{k=1}^p\|\boldsymbol{\Psi}_k\|^2} \leq \sqrt{\sum_{k=1}^p\frac{1}{\lambda_{\min}(\boldsymbol{\Sigma}_z)}\|\boldsymbol{\Psi}_k\|_{\boldsymbol{\Sigma}_z}^2} < \sqrt{\frac{pB_{\epsilon_2}^2\left[q + 2(q+1)t\right]}{\lambda_{\min}(\boldsymbol{\Sigma}_z)n}}. \tag{18}$$

Hence, by Weyl's inequality, we have

$$\sigma_{\min}(\boldsymbol{\Theta}+\boldsymbol{\Psi}) \geq \sigma_{\min}(\boldsymbol{\Theta}) - \|\boldsymbol{\Psi}\| > \sigma_{\min}(\boldsymbol{\Theta}) - \sqrt{\frac{pB_{\epsilon_2}^2\left[q + 2(q+1)t\right]}{\lambda_{\min}(\boldsymbol{\Sigma}_z)n}}, \tag{19}$$

where $t$ is taken to be small enough such that the RHS $\geq 0$, i.e.

$$1 < t \leq \frac{K_0 n}{p(q+1)} - \frac{q}{2(q+1)}, \tag{20}$$

where $K_0 := \frac{\lambda_{\min}(\boldsymbol{\Sigma}_z)\sigma^2_{\min}(\boldsymbol{\Theta})}{2B^2_{\epsilon_2}}$. We now rewrite inequality Equation (19) in terms of $n$ only. From condition Equation (17), for any given sample size $n$, the range for $t$ is:

$$1 < t \leq Kn - \log q, \tag{21}$$

where $K := \frac{\lambda_{\min}(\boldsymbol{\Sigma}_z)}{6B^2_z}$. We take

$$t = \log(Kn) - \log q - \frac{1}{2} = \log\left(\frac{K}{q}n\right) - \frac{1}{2},$$

So that condition Equation (21) is satisfied when $n \geq \frac{qe^{\frac{3}{2}}}{K}$. To satisfy condition Equation (20), a sufficient condition is:

$$\log\left(\frac{K}{q}n\right) \leq \frac{K_0 n}{p(q+1)}.$$

Note that when $n \geq \frac{qe^{\frac{3}{2}}}{K}$, we also have:

$$\log\left(\frac{K}{q}n\right) \leq \sqrt{\frac{K}{q}n}.$$

So a sufficient condition to satisfy both Equation (20) and Equation (21) is:

$$n \geq \max\left\{\frac{qe^{\frac{3}{2}}}{K}, \frac{p^2(q+1)^2 K}{qK_0^2}\right\}.$$

Then the bound Equation (18) can be rewritten as:

$$\|\boldsymbol{\Psi}\| \leq \sqrt{\frac{pB^2_{\epsilon_2}\left[q + 2(q+1)\left(\log\left(\frac{K}{q}n\right) - \frac{1}{2}\right)\right]}{\lambda_{\min}(\boldsymbol{\Sigma}_z)n}} < \sqrt{\frac{2p(q+1)B^2_{\epsilon_2}\log\left(\frac{K}{q}n\right)}{\lambda_{\min}(\boldsymbol{\Sigma}_z)n}}. \tag{22}$$

Finally, from Equation (16),

$$\lambda_{\min}\left(\frac{1}{n}\boldsymbol{X}^\top\boldsymbol{P}_Z\boldsymbol{X}\right) \geq \lambda_z\left(\sigma_{\min}(\boldsymbol{\Theta}) - \sqrt{\frac{2p(q+1)B^2_{\epsilon_2}\log\left(\frac{K}{q}n\right)}{\lambda_{\min}(\boldsymbol{\Sigma}_z)n}}\right)^2.$$

$\square$

*Proof of Theorem 2.1.* We denote the observational values $(\boldsymbol{Z}, \boldsymbol{X}, \boldsymbol{Y}) = \{(\boldsymbol{z}_i, \boldsymbol{x}_i, y_i)\}_{i=1}^n$, and $\boldsymbol{\mathcal{E}}_1 = \{\epsilon_{1,i}\}_{i=1}^n, \boldsymbol{\mathcal{E}}_2 = \{\epsilon_{2,i}\}_{i=1}^n$. The 2SLS estimator is given by:

$$\begin{aligned}
\hat{\boldsymbol{\beta}}_{\text{2SLS}} &= \left(\hat{\boldsymbol{\Theta}}^\top\boldsymbol{Z}^\top\boldsymbol{Z}\hat{\boldsymbol{\Theta}}\right)^{-1}\hat{\boldsymbol{\Theta}}^\top\boldsymbol{Z}^\top\boldsymbol{Y} \\
&= \left[((\boldsymbol{Z}^\top\boldsymbol{Z})^{-1}\boldsymbol{Z}^\top\boldsymbol{X})^\top\boldsymbol{Z}^\top\boldsymbol{Z}(\boldsymbol{Z}^\top\boldsymbol{Z})^{-1}\boldsymbol{Z}^\top\boldsymbol{X}\right]^{-1}\left((\boldsymbol{Z}^\top\boldsymbol{Z})^{-1}\boldsymbol{Z}^\top\boldsymbol{X}\right)^\top\boldsymbol{Z}^\top\boldsymbol{Y} \\
&= \left(\boldsymbol{X}^\top\boldsymbol{Z}(\boldsymbol{Z}^\top\boldsymbol{Z})^{-1}\boldsymbol{Z}^\top\boldsymbol{X}\right)^{-1}\boldsymbol{X}^\top\boldsymbol{Z}(\boldsymbol{Z}^\top\boldsymbol{Z})^{-1}\boldsymbol{Z}^\top\boldsymbol{Y} \\
&= \boldsymbol{\beta} + \left(\boldsymbol{X}^\top\boldsymbol{Z}(\boldsymbol{Z}^\top\boldsymbol{Z})^{-1}\boldsymbol{Z}^\top\boldsymbol{X}\right)^{-1}\boldsymbol{X}^\top\boldsymbol{Z}(\boldsymbol{Z}^\top\boldsymbol{Z})^{-1}\boldsymbol{Z}^\top\boldsymbol{\mathcal{E}}_1.
\end{aligned} \tag{23}$$

Define constants $\lambda_z, \lambda_{\tilde{x}} > 0$, such that the following event $\mathcal{A}$ holds with probability at least $1 - \xi$:

$$\mathcal{A} = \left\{\lambda_{\min}\left(\frac{\boldsymbol{Z}^\top\boldsymbol{Z}}{n}\right) \geq \lambda_z, \lambda_{\min}\left(\frac{\boldsymbol{X}^\top\boldsymbol{P}_Z\boldsymbol{X}}{n}\right) \geq \lambda_{\tilde{x}}\right\}, \tag{24}$$

where $\lambda_{\min}(\cdot)$ denotes the smallest eigenvalue of a matrix, $\boldsymbol{P}_Z := \boldsymbol{Z}(\boldsymbol{Z}^\top \boldsymbol{Z})^{-1}\boldsymbol{Z}^\top$ denotes the projection matrix. We will first assume the existence of such $\lambda_z, \lambda_{\tilde{x}}$, with their values to be determined later.

We first consider the case when event $\mathcal{A}$ is true. Let $\boldsymbol{Q}_{zz} := \mathbb{E}[\boldsymbol{z}\boldsymbol{z}^\top|\mathcal{A}], \boldsymbol{Q}_{zx} := \mathbb{E}[\boldsymbol{z}\boldsymbol{x}^\top|\mathcal{A}],$ $\bar{\boldsymbol{\Omega}}_{zz} := \sum_{i=1}^n (\boldsymbol{z}_i\boldsymbol{z}_i^\top - \boldsymbol{Q}_{zz}), \bar{\boldsymbol{\Omega}}_{zx} := \sum_{i=1}^n (\boldsymbol{z}_i\boldsymbol{x}_i^\top - \boldsymbol{Q}_{zx}), \boldsymbol{\Omega}_{z\epsilon_1} := \sum_{i=1}^n \boldsymbol{z}_i\epsilon_{1,i}.$

Let $\bar{B}_{zz}, \bar{B}_{zx}, B_{zx}, B_{z\epsilon_1}$ be some upper bounds such that $\left\|\boldsymbol{z}_i\boldsymbol{z}_i^\top - \boldsymbol{Q}_{zz}\right\| \leq \bar{B}_{zz}, \left\|\boldsymbol{z}_i\boldsymbol{x}_i^\top - \boldsymbol{Q}_{zx}\right\| \leq \bar{B}_{zx}, \left\|\boldsymbol{z}_i\boldsymbol{x}_i^\top\right\| \leq B_{zx}, \left\|\boldsymbol{z}_i\epsilon_{1,i}\right\| \leq B_{z\epsilon_1}$ almost surely, for all $i = 1, \dots, n$. The existence of $\bar{B}_{zz}, \bar{B}_{zx}, B_{zx}, B_{z\epsilon_1}$ is guaranteed under Assumption 2(ii).

By Lemma A.1, we have:

$$
\begin{aligned}
\mathbb{P}\left\{\left\|\frac{\boldsymbol{Z}^\top \boldsymbol{Z}}{n} - \boldsymbol{Q}_{zz}\right\| \geq \varepsilon \middle| \mathcal{A}\right\} &= \mathbb{P}\left\{\left\|\frac{\sum_{i=1}^n \boldsymbol{z}_i\boldsymbol{z}_i^\top}{n} - \boldsymbol{Q}_{zz}\right\| \geq \varepsilon \middle| \mathcal{A}\right\} \\
&= \mathbb{P}\left\{\left\|\sum_{i=1}^n (\boldsymbol{z}_i\boldsymbol{z}_i^\top - \boldsymbol{Q}_{zz})\right\| \geq n\varepsilon \middle| \mathcal{A}\right\} \\
&\leq 2q\exp\left(-\frac{n^2\varepsilon^2/2}{\nu(\bar{\boldsymbol{\Omega}}_{zz}|\mathcal{A}) + \bar{B}_{zz}n\varepsilon/3}\right).
\end{aligned}
\tag{25}
$$

Similarly,

$$
\begin{aligned}
\mathbb{P}\left\{\left\|\frac{\boldsymbol{Z}^\top \boldsymbol{X}}{n} - \boldsymbol{Q}_{zx}\right\| \geq \varepsilon \middle| \mathcal{A}\right\} &= \mathbb{P}\left\{\left\|\frac{\sum_{i=1}^n \boldsymbol{z}_i\boldsymbol{x}_i^\top}{n} - \boldsymbol{Q}_{zx}\right\| \geq \varepsilon \middle| \mathcal{A}\right\} \\
&= \mathbb{P}\left\{\left\|\sum_{i=1}^n (\boldsymbol{z}_i\boldsymbol{x}_i^\top - \boldsymbol{Q}_{zx})\right\| \geq n\varepsilon \middle| \mathcal{A}\right\} \\
&\leq (p+q)\exp\left(-\frac{n^2\varepsilon^2/2}{\nu(\bar{\boldsymbol{\Omega}}_{zx}|\mathcal{A}) + \bar{B}_{zx}n\varepsilon/3}\right).
\end{aligned}
\tag{26}
$$

By Assumption 1(iii), the instrument $\boldsymbol{z}$ is uncorrelated with the error term $\epsilon_1$, which implies $\mathbb{E}[\boldsymbol{z}\epsilon_1|\mathcal{A}] = \boldsymbol{0}$. Applying Lemma A.1 again, we have:

$$
\begin{aligned}
\mathbb{P}\left\{\left\|\frac{\boldsymbol{Z}^\top \boldsymbol{\mathcal{E}}_1}{n}\right\| \geq \varepsilon \middle| \mathcal{A}\right\} &= \mathbb{P}\left\{\left\|\frac{\sum_{i=1}^n \boldsymbol{z}_i\epsilon_{1,i}}{n}\right\| \geq \varepsilon \middle| \mathcal{A}\right\} \\
&= \mathbb{P}\left\{\left\|\sum_{i=1}^n \boldsymbol{z}_i\epsilon_{1,i}\right\| \geq n\varepsilon \middle| \mathcal{A}\right\} \\
&\leq (q+1)\exp\left(-\frac{n^2\varepsilon^2/2}{\nu(\boldsymbol{\Omega}_{z\epsilon_1}|\mathcal{A}) + B_{z\epsilon_1}n\varepsilon/3}\right).
\end{aligned}
\tag{27}
$$

With Lemma A.2 and (25), we have:

$$
\begin{aligned}
\mathbb{P}\left\{\left\|n(\boldsymbol{Z}^\top \boldsymbol{Z})^{-1} - \boldsymbol{Q}_{ZZ}^{-1}\right\| \geq \varepsilon \middle| \mathcal{A}\right\} &\leq 2q\exp\left(-\frac{n^2(\lambda_z^2\varepsilon)^2/2}{\nu(\bar{\boldsymbol{\Omega}}_{zz}|\mathcal{A}) + \bar{B}_{zz}n(\lambda_z^2\varepsilon)/3}\right) \\
&= 2q\exp\left(-\frac{\lambda_z^4 n^2\varepsilon^2/2}{\nu(\bar{\boldsymbol{\Omega}}_{zz}|\mathcal{A}) + \lambda_z^2\bar{B}_{zz}n\varepsilon/3}\right).
\end{aligned}
\tag{28}
$$

Note that we have $\hat{\boldsymbol{\Theta}} = \boldsymbol{\Theta} + (\boldsymbol{Z}^\top \boldsymbol{Z})^{-1}\boldsymbol{Z}^\top \boldsymbol{\mathcal{E}}_2 := \boldsymbol{\Theta} + \boldsymbol{\Psi}$. Under event $\mathcal{A}$,

$$
\|\hat{\boldsymbol{\Theta}}\| = \|\boldsymbol{\Theta} + \boldsymbol{\Psi}\| \leq \|\boldsymbol{\Theta}\| + \|\boldsymbol{\Psi}\| \leq B_\Theta + B_\Psi := B_{\hat{\Theta}}.
\tag{29}
$$

With Lemma A.3 (Remark A.1), combining (27)(29), we have:

$$
\begin{aligned}
\mathbb{P}\left\{\left\|\frac{1}{n}\boldsymbol{X}^\top \boldsymbol{Z}(\boldsymbol{Z}^\top \boldsymbol{Z})^{-1}\boldsymbol{Z}^\top \boldsymbol{\mathcal{E}}_1 - \boldsymbol{0}\right\| \geq \varepsilon \middle| \mathcal{A}\right\} &\leq (q+1)\exp\left(-\frac{n^2(\frac{\varepsilon}{B_{\hat{\Theta}}})^2/2}{\nu(\boldsymbol{\Omega}_{z\epsilon_1}|\mathcal{A}) + B_{z\epsilon_1}n(\frac{\varepsilon}{B_{\hat{\Theta}}})/3}\right) \\
&= (q+1)\exp\left(-\frac{n^2\varepsilon^2/2}{B_{\hat{\Theta}}^2\nu(\boldsymbol{\Omega}_{z\epsilon_1}|\mathcal{A}) + B_{\hat{\Theta}}B_{z\epsilon_1}n\varepsilon/3}\right).
\end{aligned}
\tag{30}
$$

Additionally, with Lemma A.3, combining (26)(28), we have:

$$\mathbb{P}\left\{\left\|\frac{1}{n}\boldsymbol{X}^\top \boldsymbol{Z}(\boldsymbol{Z}^\top \boldsymbol{Z})^{-1}\boldsymbol{Z}^\top \boldsymbol{X} - \boldsymbol{Q}_{zx}^\top \boldsymbol{Q}_{zz}^{-1}\boldsymbol{Q}_{zx}\right\| \geq \varepsilon \Big| \mathcal{A}\right\}$$

$$\leq 2(p+q)\exp\left(-\frac{n^2(\frac{\lambda_z \varepsilon}{3B_{zx}})^2/2}{\nu(\bar{\boldsymbol{\Omega}}_{zx}|\mathcal{A}) + \bar{B}_{zx}n(\frac{\lambda_z \varepsilon}{3B_{zx}})/3}\right) + 2q\exp\left(-\frac{\lambda_z^4 n^2(\frac{\varepsilon}{3B_{zx}^2})^2/2}{\nu(\bar{\boldsymbol{\Omega}}_{zz}|\mathcal{A}) + \lambda_z^2 \bar{B}_{zz}n(\frac{\varepsilon}{3B_{zx}^2})/3}\right)$$

$$= 2(p+q)\exp\left(-\frac{\lambda_z^2 n^2 \varepsilon^2/2}{9B_{zx}^2 \nu(\bar{\boldsymbol{\Omega}}_{zx}|\mathcal{A}) + \lambda_z B_{zx}\bar{B}_{zx}n\varepsilon}\right) + 2q\exp\left(-\frac{\lambda_z^4 n^2 \varepsilon^2/2}{9B_{zx}^4 \nu(\bar{\boldsymbol{\Omega}}_{zz}|\mathcal{A}) + \lambda_z^2 B_{zx}^2 \bar{B}_{zz}n\varepsilon}\right).$$

Applying Lemma A.2 again, we have:

$$\mathbb{P}\left\{\left\|n\left(\boldsymbol{X}^\top \boldsymbol{Z}(\boldsymbol{Z}^\top \boldsymbol{Z})^{-1}\boldsymbol{Z}^\top \boldsymbol{X}\right)^{-1} - (\boldsymbol{Q}_{ZX}^\top \boldsymbol{Q}_{zz}^{-1}\boldsymbol{Q}_{zx})^{-1}\right\| \geq \varepsilon \Big| \mathcal{A}\right\}$$

$$\leq 2(p+q)\exp\left(-\frac{\lambda_z^2 n^2(\lambda_{\tilde{x}}^2 \varepsilon)^2/2}{9B_{zx}^2 \nu(\bar{\boldsymbol{\Omega}}_{zx}|\mathcal{A}) + \lambda_z B_{zx}\bar{B}_{zx}n(\lambda_{\tilde{x}}^2 \varepsilon)}\right) + 2q\exp\left(-\frac{\lambda_z^4 n^2(\lambda_{\tilde{x}}^2 \varepsilon)^2/2}{9B_{zx}^4 \nu(\bar{\boldsymbol{\Omega}}_{zz}|\mathcal{A}) + \lambda_z^2 B_{zx}^2 \bar{B}_{zz}n(\lambda_{\tilde{x}}^2 \varepsilon)}\right)$$

$$= 2(p+q)\exp\left(-\frac{\lambda_z^2 \lambda_{\tilde{x}}^4 n^2 \varepsilon^2/2}{9B_{zx}^2 \nu(\bar{\boldsymbol{\Omega}}_{zx}|\mathcal{A}) + \lambda_z \lambda_{\tilde{x}}^2 B_{zx}\bar{B}_{zx}n\varepsilon}\right) + 2q\exp\left(-\frac{\lambda_z^4 \lambda_{\tilde{x}}^4 n^2 \varepsilon^2/2}{9B_{zx}^4 \nu(\bar{\boldsymbol{\Omega}}_{zz}|\mathcal{A}) + \lambda_z^2 \lambda_{\tilde{x}}^2 B_{zx}^2 \bar{B}_{zz}n\varepsilon}\right). \tag{31}$$

Therefore, we have shown that under event $\mathcal{A}$,

$$n\left(\boldsymbol{X}^\top \boldsymbol{Z}(\boldsymbol{Z}^\top \boldsymbol{Z})^{-1}\boldsymbol{Z}^\top \boldsymbol{X}\right)^{-1} \xrightarrow{\mathsf{p}} (\boldsymbol{Q}_{ZX}^\top \boldsymbol{Q}_{zz}^{-1}\boldsymbol{Q}_{zx})^{-1}.$$

From equation (23), combining (30) and (31) with Lemma A.3 (Remark A.1), we have:

$$\mathbb{P}\left\{\left\|\hat{\boldsymbol{\beta}}_{\mathsf{2SLS}} - \boldsymbol{\beta}\right\| \geq \varepsilon \Big| \mathcal{A}\right\}$$

$$= \mathbb{P}\left\{\left\|\left(\boldsymbol{X}^\top \boldsymbol{Z}(\boldsymbol{Z}^\top \boldsymbol{Z})^{-1}\boldsymbol{Z}^\top \boldsymbol{X}\right)^{-1}\boldsymbol{X}^\top \boldsymbol{Z}(\boldsymbol{Z}^\top \boldsymbol{Z})^{-1}\boldsymbol{Z}^\top \boldsymbol{\mathcal{E}}_1 - \boldsymbol{0}\right\| \geq \varepsilon \Big| \mathcal{A}\right\}$$

$$\leq (q+1)\exp\left(-\frac{n^2(\lambda_{\tilde{x}}\varepsilon)^2/2}{B_{\hat{\Theta}}^2 \nu(\boldsymbol{\Omega}_{z\epsilon_1}|\mathcal{A}) + B_{\hat{\Theta}}B_{z\epsilon_1}n(\lambda_{\tilde{x}}\varepsilon)/3}\right)$$

$$= (q+1)\exp\left(-\frac{\lambda_{\tilde{x}}^2 n^2 \varepsilon^2/2}{B_{\hat{\Theta}}^2 \nu(\boldsymbol{\Omega}_{z\epsilon_1}|\mathcal{A}) + \lambda_{\tilde{x}}B_{\hat{\Theta}}B_{z\epsilon_1}n\varepsilon/3}\right).$$

For the second part of the theorem, let $c := \left(\frac{3B_{\hat{\Theta}}\nu(\boldsymbol{\Omega}_{z\epsilon_1}|\mathcal{A})}{\lambda_{\tilde{x}}B_{z\epsilon_1}n}\right)^2$, we have:

$$\mathbb{E}\left[\|\mathsf{clip}_{B_\beta}(\hat{\boldsymbol{\beta}}_{\mathsf{2SLS}}) - \boldsymbol{\beta}\|^2\right]$$

$$= \mathbb{E}\left[\|\mathsf{clip}_{B_\beta}(\hat{\boldsymbol{\beta}}_{\mathsf{2SLS}}) - \boldsymbol{\beta}\|^2 \Big| \mathcal{A}\right]\mathbb{P}\{\mathcal{A}\} + \mathbb{E}\left[\|\mathsf{clip}_{B_\beta}(\hat{\boldsymbol{\beta}}_{\mathsf{2SLS}}) - \boldsymbol{\beta}\|^2 \Big| \mathcal{A}^c\right]\mathbb{P}\{\mathcal{A}^c\}$$

$$\leq \mathbb{E}\left[\|\hat{\boldsymbol{\beta}}_{\mathsf{2SLS}} - \boldsymbol{\beta}\|^2 \Big| \mathcal{A}\right]\mathbb{P}\{\mathcal{A}\} + \mathbb{E}\left[\|\mathsf{clip}_{B_\beta}(\hat{\boldsymbol{\beta}}_{\mathsf{2SLS}}) - \boldsymbol{\beta}\|^2 \Big| \mathcal{A}^c\right]\mathbb{P}\{\mathcal{A}^c\} \tag{32}$$

$$\leq \mathbb{E}\left[\|\hat{\boldsymbol{\beta}}_{\mathsf{2SLS}} - \boldsymbol{\beta}\|^2 \Big| \mathcal{A}\right] + \mathbb{E}\left[\|\mathsf{clip}_{B_\beta}(\hat{\boldsymbol{\beta}}_{\mathsf{2SLS}}) - \boldsymbol{\beta}\|^2 \Big| \mathcal{A}^c\right]\cdot \xi,$$

where

$$\mathbb{E}\left[\|\mathsf{clip}_{B_\beta}(\hat{\boldsymbol{\beta}}_{\mathsf{2SLS}}) - \boldsymbol{\beta}\|^2 \Big| \mathcal{A}^c\right] \leq 4B_\beta^2, \tag{33}$$

and

$$\mathbb{E}\left[\|\hat{\boldsymbol{\beta}}_{\mathsf{2SLS}} - \boldsymbol{\beta}\|^2 \Big| \mathcal{A}\right]$$

$$= \int_0^\infty \mathbb{P}\left\{\|\hat{\boldsymbol{\beta}}_{\mathsf{2SLS}} - \boldsymbol{\beta}\|^2 \geq \varepsilon \Big| \mathcal{A}\right\} d\varepsilon$$

$$= \int_0^\infty \mathbb{P}\left\{\|\hat{\boldsymbol{\beta}}_{\mathsf{2SLS}} - \boldsymbol{\beta}\| \geq \sqrt{\varepsilon} \Big| \mathcal{A}\right\} d\varepsilon$$

$$\leq \int_0^\infty (q+1)\exp\left(-\frac{\lambda_{\tilde{x}}^2 n^2 \varepsilon/2}{B_{\hat{\Theta}}^2 \nu(\boldsymbol{\Omega}_{z\epsilon_1}|\mathcal{A}) + \lambda_{\tilde{x}} B_{\hat{\Theta}} B_{z\epsilon_1} n\sqrt{\varepsilon}/3}\right) d\varepsilon$$

$$\leq (q+1)\left[\int_0^c \exp\left(-\frac{\lambda_{\tilde{x}}^2 n^2 \varepsilon/2}{2B_{\hat{\Theta}}^2 \nu(\boldsymbol{\Omega}_{z\epsilon_1}|\mathcal{A})}\right) d\varepsilon + \int_c^\infty \exp\left(-\frac{\lambda_{\tilde{x}} n\sqrt{\varepsilon}/2}{2B_{\hat{\Theta}} B_{z\epsilon_1}/3}\right) d\varepsilon\right]$$

$$= (q+1)\left[\frac{4B_{\hat{\Theta}}^2 \nu(\boldsymbol{\Omega}_{z\epsilon_1}|\mathcal{A})}{\lambda_{\tilde{x}}^2 n^2}\left(1 - \exp\left(-\frac{9\nu(\boldsymbol{\Omega}_{z\epsilon_1}|\mathcal{A})}{4B_{z\epsilon_1}^2}\right)\right) + \left(\frac{8B_{\hat{\Theta}}^2 \nu(\boldsymbol{\Omega}_{z\epsilon_1}|\mathcal{A})}{\lambda_{\tilde{x}}^2 n^2} + \frac{32B_{\hat{\Theta}}^2 B_{z\epsilon_1}^2}{9\lambda_{\tilde{x}}^2 n^2}\right)\exp\left(-\frac{9\nu(\boldsymbol{\Omega}_{z\epsilon_1}|\mathcal{A})}{4B_{z\epsilon_1}^2}\right)\right]$$

$$\leq (q+1)\left[\frac{4B_{\hat{\Theta}}^2 \nu(\boldsymbol{\Omega}_{z\epsilon_1}|\mathcal{A})}{\lambda_{\tilde{x}}^2 n^2} + \frac{8B_{\hat{\Theta}}^2 \nu(\boldsymbol{\Omega}_{z\epsilon_1}|\mathcal{A})}{\lambda_{\tilde{x}}^2 n^2} + \frac{32B_{\hat{\Theta}}^2 B_{z\epsilon_1}^2}{9\lambda_{\tilde{x}}^2 n^2}\right]$$

$$= \frac{(q+1)B_{\hat{\Theta}}^2}{\lambda_{\tilde{x}}^2 n^2}\left[12\nu(\boldsymbol{\Omega}_{z\epsilon_1}|\mathcal{A}) + \frac{32B_z^2 B_{\epsilon_1}^2}{9}\right]. \tag{34}$$

Note that we further have the following bound:

$$\nu(\boldsymbol{\Omega}_{z\epsilon_1}|\mathcal{A}) = \max\left\{\left\|\mathbb{E}\left[\left(\sum_{i=1}^n \boldsymbol{z}_i \epsilon_{1,i}\right)^\top\left(\sum_{j=1}^n \boldsymbol{z}_j \epsilon_{1,j}\right)\Big|\mathcal{A}\right]\right\|, \left\|\mathbb{E}\left[\left(\sum_{i=1}^n \boldsymbol{z}_i \epsilon_{1,i}\right)\left(\sum_{j=1}^n \boldsymbol{z}_j \epsilon_{1,j}\right)^\top\Big|\mathcal{A}\right]\right\|\right\}$$

$$= \max\left\{\left\|\mathbb{E}\left[\sum_{i=1}^n \epsilon_{1,i}^2 \boldsymbol{z}_i^\top \boldsymbol{z}_i\Big|\mathcal{A}\right]\right\|, \left\|\mathbb{E}\left[\sum_{i=1}^n \epsilon_{1,i}^2 \boldsymbol{z}_i \boldsymbol{z}_i^\top\Big|\mathcal{A}\right]\right\|\right\}$$

$$\leq nB_z^2 \sigma_1^2. \tag{35}$$

It now remains to determine $\lambda_z$, $\lambda_{\tilde{x}}$, and $B_{\hat{\Theta}}$.

From Theorem 4.6.1 of (Vershynin, 2018), when $n \geq c^2 B_z^4 (\sqrt{q} + \sqrt{t})^2$, with probability at least $1 - 2e^{-t}$, we have:

$$\lambda_{\min}\left(\frac{1}{n}\boldsymbol{Z}^\top \boldsymbol{Z}\right) \geq \lambda_{\min}(\boldsymbol{\Sigma}_z)\left(1 - \frac{cB_z^2(\sqrt{q} + \sqrt{t})}{\sqrt{n}}\right)^2,$$

where c is an absolute constant. We rewrite the theorem by taking $t = \log(\frac{K}{q}n) - \frac{1}{2}$ (similar to the proof in Lemma A.4). Then we need the following condition to be satisfied:

$$n \geq c^2 B_z^4 \left(\sqrt{q} + \sqrt{\log\left(\frac{K}{q}n\right) - \frac{1}{2}}\right)^2. \tag{36}$$

We can bound the RHS of Equation (36) as follows:

$$c^2 B_z^4 \left(\sqrt{q} + \sqrt{\log\left(\frac{K}{q}n\right) - \frac{1}{2}}\right)^2 \leq 2c^2 B_z^4 \left(q + \log\left(\frac{K}{q}n\right) - \frac{1}{2}\right)$$

$$\leq \frac{n}{2} + 2c^2 B_z^4 \left(q + \log\left(\frac{4c^2 B_z^4 K}{q}\right) - \frac{3}{2}\right),$$

where the second line follows from the inequality $\log(x) \leq \frac{x}{C} + \log(C) - 1$, with $x = \frac{K}{q}n$ and $C = \frac{4c^2 B_z^4 K}{q}$. So a sufficient condition for Equation (36) to hold is:

$$n \geq 4c^2 B_z^4 \left( q + \log\left(\frac{4c^2 B_z^4 K}{q}\right) - \frac{3}{2} \right). \tag{37}$$

With condition Equation (37), we have the following bound holds with probability at least $1 - \frac{2qe^{\frac{1}{2}}}{Kn}$:

$$\lambda_{\min}\left(\frac{1}{n}\boldsymbol{Z}^\top \boldsymbol{Z}\right) \geq \lambda_{\min}(\boldsymbol{\Sigma}_z)\left(1 - \frac{cB_z^2\left(\sqrt{q} + \sqrt{\log\left(\frac{K}{q}n\right) - \frac{1}{2}}\right)}{\sqrt{n}}\right)^2 := \lambda_z. \tag{38}$$

Furthermore, with Lemma A.4, when $n \geq \max\left\{\frac{qe^{\frac{3}{2}}}{K}, \frac{p^2(q+1)^2 K}{qK_0^2}\right\}$, we have the following holds with probability at least $1 - \frac{5qe^{\frac{1}{2}}}{Kn}$:

$$\lambda_{\min}\left(\frac{1}{n}\boldsymbol{X}^\top \boldsymbol{P}_Z \boldsymbol{X}\right)$$

$$\geq \lambda_z \left(\sigma_{\min}(\boldsymbol{\Theta}) - \sqrt{\frac{2p(q+1)B_{\epsilon_2}^2 \log\left(\frac{K}{q}n\right)}{\lambda_{\min}(\boldsymbol{\Sigma}_z)n}}\right)^2$$

$$= \lambda_{\min}(\boldsymbol{\Sigma}_z)\left(1 - \frac{cB_z^2\left(\sqrt{q} + \sqrt{\log\left(\frac{K}{q}n\right) - \frac{1}{2}}\right)}{\sqrt{n}}\right)^2 \left(\sigma_{\min}(\boldsymbol{\Theta}) - \sqrt{\frac{2p(q+1)B_{\epsilon_2}^2 \log\left(\frac{K}{q}n\right)}{\lambda_{\min}(\boldsymbol{\Sigma}_z)n}}\right)^2$$

$$:= \lambda_{\tilde{x}}. \tag{39}$$

From Equation (22) and Equation (29), we have:

$$B_{\hat{\Theta}} = B_{\Theta} + \sqrt{\frac{2p(q+1)B_{\epsilon_2}^2 \log\left(\frac{K}{q}n\right)}{\lambda_{\min}(\boldsymbol{\Sigma}_z)n}}. \tag{40}$$

With $\xi = \frac{5qe^{\frac{1}{2}}}{Kn}$, putting together Equations (35)(38)(39)(40) into Equation (34), and Equations (33)(34) into Equation (32), we have:

$$\mathbb{E}\left[\|\text{clip}_{B_\beta}(\hat{\boldsymbol{\beta}}_{\text{2SLS}}) - \boldsymbol{\beta}\|^2\right] \leq \mathbb{E}\left[\|\hat{\boldsymbol{\beta}}_{\text{2SLS}} - \boldsymbol{\beta}\|^2 \Big| \mathcal{A}\right] + 4B_\beta^2 \xi$$

$$\leq \frac{(q+1)B_{\hat{\Theta}}^2}{\lambda_{\tilde{x}}^2 n^2}\left[12nB_z^2\sigma_1^2 + \frac{32B_z^2 B_{\epsilon_1}^2}{9}\right] + \frac{20qe^{\frac{1}{2}}B_\beta^2}{Kn} \tag{41}$$

$$\leq \mathcal{O}\left(\frac{q}{n}\left(\frac{B_\beta^2}{K} + C^2(n)\sigma_1^2\right)\right),$$

where

$$C(n) := \frac{\left(B_{\Theta} + \sqrt{\frac{2p(q+1)B_{\epsilon_2}^2 \log\left(\frac{K}{q}n\right)}{\lambda_{\min}(\boldsymbol{\Sigma}_z)n}}\right)B_z}{\lambda_{\min}(\boldsymbol{\Sigma}_z)\left(1 - \frac{cB_z^2\left(\sqrt{q} + \sqrt{\log\left(\frac{K}{q}n\right) - \frac{1}{2}}\right)}{\sqrt{n}}\right)^2 \left(\sigma_{\min}(\boldsymbol{\Theta}) - \sqrt{\frac{2p(q+1)B_{\epsilon_2}^2 \log\left(\frac{K}{q}n\right)}{\lambda_{\min}(\boldsymbol{\Sigma}_z)n}}\right)^2}, \tag{42}$$

thus completing the proof. $\qquad\square$

## B   PROOFS FOR SECTION 3

### B.1   PROOF OF THEOREM 3.1

**Lemma B.1.** Suppose $\{\boldsymbol{\Omega}^{(1)}, \ldots, \boldsymbol{\Omega}^{(t)}, \ldots\}$ is a $d \times d$-matrix sequence decaying with exponential rate $r$, i.e. for some constant $c > 0$ and $0 < r < 1$,

$$\left\| \boldsymbol{\Omega}^{(t)} \right\|_F \le cr^t.$$

Then for any $\varepsilon > 0$, there exists a finite constant:

$$T_0 = \left\lceil \log_r \frac{(1-r)(\varepsilon/d)}{c\left(1 + (1-r)(\varepsilon/d)\right)} \right\rceil,$$

such that

$$\left\| \prod_{t=T_0}^{\infty} \left(\boldsymbol{I} + \boldsymbol{\Omega}^{(t)}\right) - \boldsymbol{I} \right\|_F < \varepsilon,$$

and hence

$$\left\| \prod_{t=T_0}^{\infty} \left(\boldsymbol{I} + \boldsymbol{\Omega}^{(t)}\right) \right\|_F < \sqrt{d} + \varepsilon.$$

*Proof.* By definition,

$$\left\| \boldsymbol{\Omega}^{(k)} \right\|_F = \sqrt{\sum_{i,j=1}^{p} \boldsymbol{\Omega}_{ij}^{(k)2}} \le cr^k,$$

which implies:

$$\left| \boldsymbol{\Omega}_{ij}^{(k)} \right| \le cr^k, \quad \forall i, j, k.$$

Consider the product of any two matrices. By sub-multiplicativity,

$$\left\| \boldsymbol{\Omega}^{(k)} \boldsymbol{\Omega}^{(l)} \right\|_F \le \left\| \boldsymbol{\Omega}^{(k)} \right\|_F \left\| \boldsymbol{\Omega}^{(l)} \right\|_F \le c^2 r^{k+l},$$

which implies:

$$\left| \left[ \boldsymbol{\Omega}^{(k)} \boldsymbol{\Omega}^{(l)} \right]_{ij} \right| \le c^2 r^{k+l}, \quad \forall i, j, k, l.$$

Similarly, for the product of any number of matrices:

$$\left| \left[ \boldsymbol{\Omega}^{(k_1)} \boldsymbol{\Omega}^{(k_2)} \cdots \boldsymbol{\Omega}^{(k_n)} \right]_{ij} \right| \le c^n r^{k_1 + k_2 + \cdots + k_n}, \quad \forall i, j, k_1, \ldots, k_n.$$

Thus

$$
\begin{aligned}
&\left\| \prod_{t=t_1}^{t_2} \left(\boldsymbol{I} + \boldsymbol{\Omega}^{(t)}\right) - \boldsymbol{I} \right\|_F \\
&= \left\| \left(\boldsymbol{I} + \boldsymbol{\Omega}^{(t_1)}\right) \left(\boldsymbol{I} + \boldsymbol{\Omega}^{(t_1+1)}\right) \cdots \left(\boldsymbol{I} + \boldsymbol{\Omega}^{(t_2)}\right) - \boldsymbol{I} \right\|_F \\
&= \left\| \sum_{t_1 \le k \le t_2} \boldsymbol{\Omega}^{(k)} + \sum_{t_1 \le k < l \le t_2} \boldsymbol{\Omega}^{(k)} \boldsymbol{\Omega}^{(l)} + \cdots + \boldsymbol{\Omega}^{(t_1)} \boldsymbol{\Omega}^{(t_1+1)} \cdots \boldsymbol{\Omega}^{(t_2)} \right\|_F \\
&\le \left\| \sum_{t_1 \le k \le t_2} cr^k \boldsymbol{1}\boldsymbol{1}^\top + \sum_{t_1 \le k < l \le t_2} c^2 r^{k+l} \boldsymbol{1}\boldsymbol{1}^\top + \cdots + c^{t2-t1+1} r^{t_1 + \cdots + t_2} \boldsymbol{1}\boldsymbol{1}^\top \right\|_F.
\end{aligned}
\tag{43}
$$

Note that the last inequality can be checked by comparing matrix elements of both sides. For any $\varepsilon > 0$, we take $T_0 = \lceil \log_r \frac{(1-r)(\varepsilon/d)}{c(1+(1-r)(\varepsilon/d))} \rceil$. Consider $t_1 = T_0$ and $t_2 \to \infty$ in (43). For notation convenience, let

$$\mathbf{\Xi} := \sum_{T_0 \leq k} cr^k \mathbf{1}\mathbf{1}^\top + \sum_{T_0 \leq k < l} c^2 r^{k+l} \mathbf{1}\mathbf{1}^\top + \sum_{T_0 \leq k < l < m} c^3 r^{k+l+m} \mathbf{1}\mathbf{1}^\top + \cdots .$$

Then

$$
\begin{aligned}
\mathbf{\Xi}_{ij} &= \sum_{T_0 \leq k} cr^k + \sum_{T_0 \leq k < l} c^2 r^{k+l} + \sum_{T_0 \leq k < l < m} c^3 r^{k+l+m} + \cdots \\
&< c \sum_{k \geq T_0} r^k + c^2 r^{T_0} \sum_{k \geq T_0} r^k + c^3 r^{2T_0} \sum_{k \geq T_0} r^k + \cdots \\
&= \frac{cr^{T_0}}{1-r} + \frac{c^2 r^{2T_0}}{1-r} + \frac{c^3 r^{3T_0}}{1-r} + \cdots \\
&= \frac{cr^{T_0}}{(1-r)(1 - cr^{T_0})} \\
&\leq \frac{\varepsilon}{d}.
\end{aligned}
$$

Thus

$$\left\| \prod_{t=T_0}^{\infty} (\mathbf{I} + \mathbf{\Omega}^{(t)}) - \mathbf{I} \right\|_F = \|\mathbf{\Xi}\|_F = \sqrt{\sum_{i,j=1}^{d} \mathbf{\Xi}_{ij}^2} \leq \varepsilon.$$

Hence completes the proof. $\qquad\square$

*Proof of Theorem 3.1.* In the following proof, we treat $\mathbf{Z}, \mathbf{X}, \mathbf{Y}$ as deterministic matrices.

We begin by checking the inner loop (7a):

$$
\begin{aligned}
\mathbf{\Theta}^{(t)} - \hat{\mathbf{\Theta}} &= \mathbf{\Theta}^{(t-1)} - \hat{\mathbf{\Theta}} - \eta \mathbf{Z}^\top \left( \mathbf{Z}\mathbf{\Theta}^{(t-1)} - \mathbf{X} \right) \\
&= \left( \mathbf{I} - \eta \mathbf{Z}^\top \mathbf{Z} \right) \left( \mathbf{\Theta}^{(t-1)} - \hat{\mathbf{\Theta}} \right) + \eta \mathbf{Z}^\top \left( \mathbf{X} - \mathbf{Z}\hat{\mathbf{\Theta}} \right) \\
&= \left( \mathbf{I} - \eta \mathbf{Z}^\top \mathbf{Z} \right)^2 \left( \mathbf{\Theta}^{(t-2)} - \hat{\mathbf{\Theta}} \right) + \eta \mathbf{Z}^\top \left( \mathbf{X} - \mathbf{Z}\hat{\mathbf{\Theta}} \right) + \eta \left( \mathbf{I} - \eta \mathbf{Z}^\top \mathbf{Z} \right) \mathbf{Z}^\top \left( \mathbf{X} - \mathbf{Z}\hat{\mathbf{\Theta}} \right) \\
&\qquad\qquad \vdots \\
&= \left( \mathbf{I} - \eta \mathbf{Z}^\top \mathbf{Z} \right)^t \left( \mathbf{\Theta}^{(0)} - \hat{\mathbf{\Theta}} \right) + \sum_{i=0}^{t-1} \eta \left( \mathbf{I} - \eta \mathbf{Z}^\top \mathbf{Z} \right)^{t-1-i} \mathbf{Z}^\top \left( \mathbf{X} - \mathbf{Z}\hat{\mathbf{\Theta}} \right) \\
&= \left( \mathbf{I} - \eta \mathbf{Z}^\top \mathbf{Z} \right)^t \left( \mathbf{\Theta}^{(0)} - \hat{\mathbf{\Theta}} \right) + \eta \left[ \mathbf{I} - (\mathbf{I} - \eta \mathbf{Z}^\top \mathbf{Z})^t \right] \left( \eta \mathbf{Z}^\top \mathbf{Z} \right)^{-1} \mathbf{Z}^\top \left( \mathbf{X} - \mathbf{Z}\hat{\mathbf{\Theta}} \right) \\
&= \left( \mathbf{I} - \eta \mathbf{Z}^\top \mathbf{Z} \right)^t \left( \mathbf{\Theta}^{(0)} - \hat{\mathbf{\Theta}} \right) + \left[ \mathbf{I} - (\mathbf{I} - \eta \mathbf{Z}^\top \mathbf{Z})^t \right] \left[ (\mathbf{Z}^\top \mathbf{Z})^{-1} \mathbf{Z}^\top \mathbf{X} - \hat{\mathbf{\Theta}} \right] \\
&= \left( \mathbf{I} - \eta \mathbf{Z}^\top \mathbf{Z} \right)^t \left( \mathbf{\Theta}^{(0)} - \hat{\mathbf{\Theta}} \right) .
\end{aligned}
$$

With learning rate $0 < \eta < \frac{2}{\sigma_{\max}^2(\mathbf{Z})}$, let $\kappa(\eta) := \rho \left( \mathbf{I} - \eta \mathbf{Z}^\top \mathbf{Z} \right)$, where $\rho(\cdot)$ denotes the spectral radius. Then it follows that $0 < \kappa(\eta) < 1$. We have:

$$
\begin{aligned}
\left\| \mathbf{\Theta}^{(t)} - \hat{\mathbf{\Theta}} \right\| &= \left\| (\mathbf{I} - \eta \mathbf{Z}^\top \mathbf{Z})^t (\mathbf{\Theta}^{(0)} - \hat{\mathbf{\Theta}}) \right\| \\
&\leq \left\| (\mathbf{I} - \eta \mathbf{Z}^\top \mathbf{Z})^t \right\| \left\| \mathbf{\Theta}^{(0)} - \hat{\mathbf{\Theta}} \right\| \\
&\leq \kappa(\eta)^t \left\| \mathbf{\Theta}^{(0)} - \hat{\mathbf{\Theta}} \right\| \\
&= \mathcal{O}(\kappa(\eta)^t).
\end{aligned}
$$

Thus $\{\boldsymbol{\Theta}^{(t)}\}$ converges to $\hat{\boldsymbol{\Theta}}$ exponentially with rate $\kappa(\eta)$.

For the outer loop (7b), we have:

$$
\begin{aligned}
\boldsymbol{\beta}^{(t)} - \hat{\boldsymbol{\beta}}_{\text{2SLS}} &= \boldsymbol{\beta}^{(t-1)} - \hat{\boldsymbol{\beta}}_{\text{2SLS}} - \alpha \boldsymbol{\Theta}^{(t-1)\top} \boldsymbol{Z}^\top \left( \boldsymbol{Z} \boldsymbol{\Theta}^{(t-1)} \boldsymbol{\beta}^{(t-1)} - \boldsymbol{Y} \right) \\
&= \left( \boldsymbol{I} - \alpha \boldsymbol{\Theta}^{(t-1)\top} \boldsymbol{Z}^\top \boldsymbol{Z} \boldsymbol{\Theta}^{(t-1)} \right) \left( \boldsymbol{\beta}^{(t-1)} - \hat{\boldsymbol{\beta}}_{\text{2SLS}} \right) + \alpha \boldsymbol{\Theta}^{(t-1)\top} \boldsymbol{Z}^\top \left( \boldsymbol{Y} - \boldsymbol{Z} \boldsymbol{\Theta}^{(t-1)} \hat{\boldsymbol{\beta}}_{\text{2SLS}} \right) \\
&\quad \vdots \\
&= \underbrace{\prod_{i=0}^{t-1} \left( \boldsymbol{I} - \alpha \boldsymbol{\Theta}^{(i)\top} \boldsymbol{Z}^\top \boldsymbol{Z} \boldsymbol{\Theta}^{(i)} \right) \left( \boldsymbol{\beta}^{(0)} - \hat{\boldsymbol{\beta}}_{\text{2SLS}} \right)}_{\boldsymbol{\Delta}_1 \boldsymbol{\beta}^{(t)}} \\
&\quad + \underbrace{\sum_{i=0}^{t-1} \alpha \left[ \prod_{j=i+1}^{t-1} \left( \boldsymbol{I} - \alpha \boldsymbol{\Theta}^{(j)\top} \boldsymbol{Z}^\top \boldsymbol{Z} \boldsymbol{\Theta}^{(j)} \right) \right] \boldsymbol{\Theta}^{(i)\top} \boldsymbol{Z}^\top \left( \boldsymbol{Y} - \boldsymbol{Z} \boldsymbol{\Theta}^{(i)} \hat{\boldsymbol{\beta}}_{\text{2SLS}} \right)}_{\boldsymbol{\Delta}_2 \boldsymbol{\beta}^{(t)}} .
\end{aligned}
\tag{44}
$$

To simplify notations, let

$$
\begin{aligned}
\boldsymbol{R}^{(t)} &:= \boldsymbol{\Theta}^{(t)} - \hat{\boldsymbol{\Theta}} = \left( \boldsymbol{I} - \eta \boldsymbol{Z}^\top \boldsymbol{Z} \right)^t \left( \boldsymbol{\Theta}^{(0)} - \hat{\boldsymbol{\Theta}} \right), \\
\boldsymbol{V}^{(t)} &:= \left( \boldsymbol{I} - \alpha \hat{\boldsymbol{\Theta}}^\top \boldsymbol{Z}^\top \boldsymbol{Z} \hat{\boldsymbol{\Theta}} \right)^t, \\
\boldsymbol{W}^{(t)} &:= \boldsymbol{R}^{(t)\top} \boldsymbol{Z}^\top \boldsymbol{Z} \hat{\boldsymbol{\Theta}} + \hat{\boldsymbol{\Theta}}^\top \boldsymbol{Z}^\top \boldsymbol{Z} \boldsymbol{R}^{(t)} + \boldsymbol{R}^{(t)\top} \boldsymbol{Z}^\top \boldsymbol{Z} \boldsymbol{R}^{(t)}.
\end{aligned}
$$

With learning rates $0 < \alpha < \frac{2}{\sigma_{\max}^2(\boldsymbol{Z}\hat{\boldsymbol{\Theta}})}$, $0 < \eta < \frac{2}{\sigma_{\max}^2(\boldsymbol{Z})}$, let $\gamma(\alpha) := \rho \left( \boldsymbol{I} - \alpha \hat{\boldsymbol{\Theta}}^\top \boldsymbol{Z}^\top \boldsymbol{Z} \hat{\boldsymbol{\Theta}} \right)$. Then it follows that $0 < \gamma(\alpha) < 1$. We have:

$$
\left\| \boldsymbol{R}^{(t)} \right\| \leq \kappa(\eta)^t \left\| \boldsymbol{\Theta}^{(0)} - \hat{\boldsymbol{\Theta}} \right\|,
\tag{45}
$$

$$
\left\| \boldsymbol{V}^{(t)} \right\| \leq \gamma(\alpha)^t,
$$

and

$$
\begin{aligned}
\left\| \boldsymbol{W}^{(t)} \right\| &= \left\| \boldsymbol{R}^{(t)\top} \boldsymbol{Z}^\top \boldsymbol{Z} \hat{\boldsymbol{\Theta}} + \hat{\boldsymbol{\Theta}}^\top \boldsymbol{Z}^\top \boldsymbol{Z} \boldsymbol{R}^{(t)} + \boldsymbol{R}^{(t)\top} \boldsymbol{Z}^\top \boldsymbol{Z} \boldsymbol{R}^{(t)} \right\| \\
&\leq 2 \left\| \hat{\boldsymbol{\Theta}}^\top \boldsymbol{Z}^\top \boldsymbol{Z} \right\| \left\| \boldsymbol{R}^{(t)} \right\| + \left\| \boldsymbol{Z}^\top \boldsymbol{Z} \right\| \left\| \boldsymbol{R}^{(t)} \right\|^2 \\
&\leq 2 \kappa(\eta)^t \left\| \hat{\boldsymbol{\Theta}} \boldsymbol{Z}^\top \boldsymbol{Z} \right\| \left\| \boldsymbol{\Theta}^{(0)} - \hat{\boldsymbol{\Theta}} \right\| + \kappa(\eta)^{2t} \left\| \boldsymbol{Z}^\top \boldsymbol{Z} \right\| \left\| \boldsymbol{\Theta}^{(0)} - \hat{\boldsymbol{\Theta}} \right\|^2 \\
&\leq \kappa(\eta)^t \left( 2 \left\| \hat{\boldsymbol{\Theta}} \boldsymbol{Z}^\top \boldsymbol{Z} \right\| \left\| \boldsymbol{\Theta}^{(0)} - \hat{\boldsymbol{\Theta}} \right\| + \left\| \boldsymbol{Z}^\top \boldsymbol{Z} \right\| \left\| \boldsymbol{\Theta}^{(0)} - \hat{\boldsymbol{\Theta}} \right\|^2 \right) \\
&= \mathcal{O}(\kappa(\eta)^t).
\end{aligned}
$$

Then from Equation (44), we have:

$$\boldsymbol{\Delta}_1\boldsymbol{\beta}^{(t)} = \prod_{i=0}^{t-1} \left(\boldsymbol{I} - \alpha\boldsymbol{\Theta}^{(i)\top}\boldsymbol{Z}^\top\boldsymbol{Z}\boldsymbol{\Theta}^{(i)}\right)\left(\boldsymbol{\beta}^{(0)} - \hat{\boldsymbol{\beta}}_{\mathsf{2SLS}}\right)$$

$$= \prod_{i=0}^{t-1}\left[\boldsymbol{I} - \alpha\hat{\boldsymbol{\Theta}}^\top\boldsymbol{Z}^\top\boldsymbol{Z}\hat{\boldsymbol{\Theta}} - \alpha\left(\boldsymbol{R}^{(i)\top}\boldsymbol{Z}^\top\boldsymbol{Z}\hat{\boldsymbol{\Theta}} + \hat{\boldsymbol{\Theta}}^\top\boldsymbol{Z}^\top\boldsymbol{Z}\boldsymbol{R}^{(i)} + \boldsymbol{R}^{(i)\top}\boldsymbol{Z}^\top\boldsymbol{Z}\boldsymbol{R}^{(i)}\right)\right]\left(\boldsymbol{\beta}^{(0)} - \hat{\boldsymbol{\beta}}_{\mathsf{2SLS}}\right)$$

$$= \prod_{i=0}^{t-1}\left[\boldsymbol{I} - \alpha\hat{\boldsymbol{\Theta}}^\top\boldsymbol{Z}^\top\boldsymbol{Z}\hat{\boldsymbol{\Theta}} - \alpha\boldsymbol{W}^{(i)}\right]\left(\boldsymbol{\beta}^{(0)} - \hat{\boldsymbol{\beta}}_{\mathsf{2SLS}}\right)$$

$$= \left(\boldsymbol{I} - \alpha\hat{\boldsymbol{\Theta}}^\top\boldsymbol{Z}^\top\boldsymbol{Z}\hat{\boldsymbol{\Theta}}\right)^t\prod_{i=0}^{t-1}\left[\boldsymbol{I} - \alpha\left(\boldsymbol{I} - \alpha\hat{\boldsymbol{\Theta}}^\top\boldsymbol{Z}^\top\boldsymbol{Z}\hat{\boldsymbol{\Theta}}\right)^{-1}\boldsymbol{W}^{(i)}\right]\left(\boldsymbol{\beta}^{(0)} - \hat{\boldsymbol{\beta}}_{\mathsf{2SLS}}\right)$$

$$= \boldsymbol{V}^{(t)}\prod_{i=0}^{t-1}\left[\boldsymbol{I} - \alpha\left(\boldsymbol{I} - \alpha\hat{\boldsymbol{\Theta}}^\top\boldsymbol{Z}^\top\boldsymbol{Z}\hat{\boldsymbol{\Theta}}\right)^{-1}\boldsymbol{W}^{(i)}\right]\left(\boldsymbol{\beta}^{(0)} - \hat{\boldsymbol{\beta}}_{\mathsf{2SLS}}\right).$$

We denote $\boldsymbol{\Psi} := \alpha\left(\boldsymbol{I} - \alpha\hat{\boldsymbol{\Theta}}^\top\boldsymbol{Z}^\top\boldsymbol{Z}\hat{\boldsymbol{\Theta}}\right)^{-1}$. By Lemma B.1, we take $\varepsilon = 1$, $c_0$ be a constant such that $\left\|\boldsymbol{W}^{(t)}\right\|_F \le c_0\kappa(\eta)^t$, and $T_0 = \lceil\log_{\kappa(\eta)}\frac{(1-\kappa(\eta))}{\|\boldsymbol{\Psi}\|_F c_0(p+(1-\kappa(\eta)))}\rceil$.

Then we have:

$$\left\|\prod_{i=T_0}^{t-1}\left(\boldsymbol{I} - \boldsymbol{\Psi}\boldsymbol{W}^{(i)}\right)\right\| \le \left\|\prod_{i=T_0}^{t-1}\left(\boldsymbol{I} - \boldsymbol{\Psi}\boldsymbol{W}^{(i)}\right)\right\|_F < \sqrt{p} + 1. \tag{46}$$

Hence

$$\left\|\boldsymbol{\Delta}_1\boldsymbol{\beta}^{(t)}\right\| = \left\|\boldsymbol{V}^{(t)}\prod_{i=0}^{t-1}(\boldsymbol{I} - \boldsymbol{\Psi}\boldsymbol{W}^{(i)})(\boldsymbol{\beta}^{(0)} - \hat{\boldsymbol{\beta}}_{\mathsf{2SLS}})\right\|$$

$$\le \left\|\boldsymbol{V}^{(t)}\right\|\left\|\prod_{i=0}^{T_0-1}\left(\boldsymbol{I} - \boldsymbol{\Psi}\boldsymbol{W}^{(i)}\right)\right\|\left\|\prod_{i=T_0}^{t-1}(\boldsymbol{I} - \boldsymbol{\Psi}\boldsymbol{W}^{(i)})\right\|\left\|\boldsymbol{\beta}^{(0)} - \hat{\boldsymbol{\beta}}_{\mathsf{2SLS}}\right\| \tag{47}$$

$$< \gamma(\alpha)^t\left\|\prod_{i=0}^{T_0-1}\left(\boldsymbol{I} - \boldsymbol{\Psi}\boldsymbol{W}^{(i)}\right)\right\|(\sqrt{p} + 1)\left\|\boldsymbol{\beta}^{(0)} - \hat{\boldsymbol{\beta}}_{\mathsf{2SLS}}\right\|$$

$$= \mathcal{O}(\gamma(\alpha)^t).$$

Next we consider $\boldsymbol{\Delta}_2\boldsymbol{\beta}^{(t)}$:

$$\boldsymbol{\Delta}_2\boldsymbol{\beta}^{(t)} = \sum_{i=0}^{t-1}\alpha\left[\prod_{j=i+1}^{t-1}\left(\boldsymbol{I} - \alpha\boldsymbol{\Theta}^{(j)\top}\boldsymbol{Z}^\top\boldsymbol{Z}\boldsymbol{\Theta}^{(j)}\right)\right]\boldsymbol{\Theta}^{(i)\top}\boldsymbol{Z}^\top\left(\boldsymbol{Y} - \boldsymbol{Z}\boldsymbol{\Theta}^{(i)}\hat{\boldsymbol{\beta}}_{\mathsf{2SLS}}\right)$$

$$= \sum_{i=0}^{t-1}\alpha\left[\prod_{j=i+1}^{t-1}\left(\boldsymbol{I} - \alpha\hat{\boldsymbol{\Theta}}^\top\boldsymbol{Z}^\top\boldsymbol{Z}\hat{\boldsymbol{\Theta}} - \alpha\boldsymbol{W}^{(j)}\right)\right]\left(\boldsymbol{R}^{(i)} + \hat{\boldsymbol{\Theta}}\right)^\top\boldsymbol{Z}^\top\left[\boldsymbol{Y} - \boldsymbol{Z}\left(\boldsymbol{R}^{(i)} + \hat{\boldsymbol{\Theta}}\right)\hat{\boldsymbol{\beta}}_{\mathsf{2SLS}}\right]$$

$$= \sum_{i=0}^{t-1}\alpha\left(\boldsymbol{I} - \alpha\hat{\boldsymbol{\Theta}}^\top\boldsymbol{Z}^\top\boldsymbol{Z}\hat{\boldsymbol{\Theta}}\right)^{t-1-i}\prod_{j=i+1}^{t-1}\left[\boldsymbol{I} - \alpha\left(\boldsymbol{I} - \alpha\hat{\boldsymbol{\Theta}}^\top\boldsymbol{Z}^\top\boldsymbol{Z}\hat{\boldsymbol{\Theta}}\right)^{-1}\boldsymbol{W}^{(j)}\right]$$

$$\cdot\left(\boldsymbol{R}^{(i)} + \hat{\boldsymbol{\Theta}}\right)^\top\boldsymbol{Z}^\top\left[\boldsymbol{Y} - \boldsymbol{Z}\left(\boldsymbol{R}^{(i)} + \hat{\boldsymbol{\Theta}}\right)\hat{\boldsymbol{\beta}}_{\mathsf{2SLS}}\right]$$

$$= \sum_{i=0}^{t-1}\alpha\left[\boldsymbol{V}^{(t-1-i)}\prod_{j=i+1}^{t-1}\left(\boldsymbol{I} - \boldsymbol{\Psi}\boldsymbol{W}^{(j)}\right)\right]\left(\boldsymbol{R}^{(i)} + \hat{\boldsymbol{\Theta}}\right)^\top\boldsymbol{Z}^\top\left[\boldsymbol{Y} - \boldsymbol{Z}\left(\boldsymbol{R}^{(i)} + \hat{\boldsymbol{\Theta}}\right)\hat{\boldsymbol{\beta}}_{\mathsf{2SLS}}\right].$$

For convenience, let $\boldsymbol{\Delta}_2\boldsymbol{\beta}^{(t)} := \boldsymbol{\Delta}_{21}\boldsymbol{\beta}^{(t)} + \boldsymbol{\Delta}_{22}\boldsymbol{\beta}^{(t)}$, where

$$\boldsymbol{\Delta}_{21}\boldsymbol{\beta}^{(t)} := \sum_{i=0}^{t-1}\alpha\left[\boldsymbol{V}^{(t-1-i)}\prod_{j=i+1}^{t-1}\left(\boldsymbol{I} - \boldsymbol{\Psi}\boldsymbol{W}^{(j)}\right)\right]\boldsymbol{R}^{(i)\top}\boldsymbol{Z}^\top\left[\boldsymbol{Y} - \boldsymbol{Z}\left(\boldsymbol{R}^{(i)} + \hat{\boldsymbol{\Theta}}\right)\hat{\boldsymbol{\beta}}_{\mathsf{2SLS}}\right],$$

$$\boldsymbol{\Delta}_{22}\boldsymbol{\beta}^{(t)} := \sum_{i=0}^{t-1} \alpha \left[ \boldsymbol{V}^{(t-1-i)} \prod_{j=i+1}^{t-1} \left( \boldsymbol{I} - \boldsymbol{\Psi}\boldsymbol{W}^{(j)} \right) \right] \hat{\boldsymbol{\Theta}}^{\top}\boldsymbol{Z}^{\top} \left[ \boldsymbol{Y} - \boldsymbol{Z}\left( \boldsymbol{R}^{(i)} + \hat{\boldsymbol{\Theta}} \right) \hat{\boldsymbol{\beta}}_{\text{2SLS}} \right].$$

Suppose $\tilde{M}_1, \tilde{M}_2$ are the upper bounds such that

$$\left\| \boldsymbol{Z}^{\top} \left[ \boldsymbol{Y} - \boldsymbol{Z}\left( \boldsymbol{R}^{(i)} + \hat{\boldsymbol{\Theta}} \right) \hat{\boldsymbol{\beta}}_{\text{2SLS}} \right] \right\| \le \tilde{M}_1, \quad \forall i = 0, \dots, t-1,$$

$$\left\| \prod_{j=i+1}^{t-1} \left( \boldsymbol{I} - \boldsymbol{\Psi}\boldsymbol{W}^{(j)} \right) \right\| \le \tilde{M}_2, \quad \forall i = 0, \dots, t-1.$$

We know such $\tilde{M}_1, \tilde{M}_2$ exist because of the bounds given by (45) and (46). Let $\tilde{M} = \tilde{M}_1\tilde{M}_2$. Then

$$\begin{aligned}
\left\| \boldsymbol{\Delta}_{21}\boldsymbol{\beta}^{(t)} \right\| &\le \tilde{M} \left\| \sum_{i=0}^{t-1} \alpha \boldsymbol{V}^{(t-1-i)} \boldsymbol{R}^{(i)\top} \right\| \\
&\le \tilde{M}\alpha \sum_{i=0}^{t-1} \left\| \boldsymbol{V}^{(t-1-i)} \right\| \left\| \boldsymbol{R}^{(i)} \right\| \\
&\le \tilde{M}\alpha \left\| \boldsymbol{\Theta}^{(0)} - \hat{\boldsymbol{\Theta}} \right\| \sum_{i=0}^{t-1} \gamma(\alpha)^{t-1-i}\kappa(\eta)^{i} \\
&= \tilde{M}\alpha \left\| \boldsymbol{\Theta}^{(0)} - \hat{\boldsymbol{\Theta}} \right\| \sum_{i=0}^{t-1} \gamma(\alpha)^{t-1} \left( \frac{\kappa(\eta)}{\gamma(\alpha)} \right)^{i} \\
&= \mathcal{O}\left( \frac{\gamma(\alpha)^{t} - \kappa(\eta)^{t}}{\gamma(\alpha) - \kappa(\eta)} \right) \\
&\le \mathcal{O}(\max\{\gamma(\alpha)^{t}, \kappa(\eta)^{t}\}),
\end{aligned}$$

and similarly,

$$\begin{aligned}
\left\| \boldsymbol{\Delta}_{22}\boldsymbol{\beta}^{(t)} \right\| &\le \tilde{M} \left\| \sum_{i=0}^{t-1} \alpha \boldsymbol{V}^{(t-1-i)} \hat{\boldsymbol{\Theta}}^{\top} \right\| \\
&\le \tilde{M}\alpha \left\| \hat{\boldsymbol{\Theta}} \right\| \sum_{i=0}^{t-1} \left\| \boldsymbol{V}^{(t-1-i)} \right\| \\
&\le \tilde{M}\alpha \left\| \hat{\boldsymbol{\Theta}} \right\| \sum_{i=0}^{t-1} \gamma(\alpha)^{t-1-i} \\
&= \mathcal{O}(\gamma(\alpha)^{t}).
\end{aligned}$$

Thus

$$\begin{aligned}
\left\| \boldsymbol{\Delta}_{2}\boldsymbol{\beta}^{(t)} \right\| &= \left\| \boldsymbol{\Delta}_{21}\boldsymbol{\beta}^{(t)} + \boldsymbol{\Delta}_{22}\boldsymbol{\beta}^{(t)} \right\| \\
&\le \left\| \boldsymbol{\Delta}_{21}\boldsymbol{\beta}^{(t)} \right\| + \left\| \boldsymbol{\Delta}_{22}\boldsymbol{\beta}^{(t)} \right\| \\
&\le \mathcal{O}(\max\{\gamma(\alpha)^{t}, \kappa(\eta)^{t}\}).
\end{aligned} \tag{48}$$

Therefore, plugging (47) and (48) into (44), we have:

$$\left\| \boldsymbol{\beta}^{(t)} - \hat{\boldsymbol{\beta}}_{\text{2SLS}} \right\| \le \mathcal{O}(\max\{\gamma(\alpha)^{t}, \kappa(\eta)^{t}\}),$$

Hence completes the proof. $\qquad\square$

## B.2 PROOF OF THEOREM 3.2

*Proof of Theorem 3.2.* For ease of notations, we ignore $l$ in the following proof. Consider the input matrix taking the form:

$$
\boldsymbol{H}^{(0)} = \begin{bmatrix}
\boldsymbol{z}_1 & \cdots & \boldsymbol{z}_n & \boldsymbol{z}_{n+1} \\
\boldsymbol{x}_1 & \cdots & \boldsymbol{x}_n & \boldsymbol{x}_{n+1} \\
y_1 & \cdots & y_n & 0 \\
\boldsymbol{\Theta}^{(0)}_{:,1} & \cdots & \boldsymbol{\Theta}^{(0)}_{:,1} & \boldsymbol{\Theta}^{(0)}_{:,1} \\
\vdots & \vdots & \vdots & \vdots \\
\boldsymbol{\Theta}^{(0)}_{:,p} & \cdots & \boldsymbol{\Theta}^{(0)}_{:,p} & \boldsymbol{\Theta}^{(0)}_{:,p} \\
\boldsymbol{\beta}^{(0)} & \cdots & \boldsymbol{\beta}^{(0)} & \boldsymbol{\beta}^{(0)} \\
\hat{\boldsymbol{x}}^{(0)}_1 & \cdots & \hat{\boldsymbol{x}}^{(0)}_n & \hat{\boldsymbol{x}}^{(0)}_{n+1} \\
1 & \cdots & 1 & 1 \\
1 & \cdots & 1 & 0
\end{bmatrix} \in \mathbb{R}^{D \times (n+1)},
$$

i.e., element-wise,

$$
\boldsymbol{h}^{(0)}_i = \left( \boldsymbol{z}_i, \boldsymbol{x}_i, y_i t_i, \boldsymbol{\Theta}^{(0)}_{:,1}, \ldots, \boldsymbol{\Theta}^{(0)}_{:,p}, \boldsymbol{\beta}^{(0)}, \hat{\boldsymbol{x}}^{(0)}_i, 1, t_i \right)^\top, \quad i = 1, \ldots, n+1,
$$

where $D = qp + 3p + q + 3$, $t_i := \mathbb{1}\{i \leq n\}$ is the indicator for training sample. We can take any initialization for $\boldsymbol{\Theta}^{(0)}$, $\boldsymbol{\beta}^{(0)}$ and $\hat{\boldsymbol{x}}^{(0)}$. To avoid abuse of notations, we omit the superscript of those parameters to be updated in the following proof.

Recall the definitions (4) and (5). Our goal is to show that there exists a series of attention parameters $\boldsymbol{\theta}^{(1:2)}_{\text{ATTN}} = \{(\boldsymbol{Q}^{(1:2)}_m, \boldsymbol{K}^{(1:2)}_m, \boldsymbol{V}^{(1:2)}_m)\}_{m \in [M]} \subset \mathbb{R}^{D \times D}$ such that $\boldsymbol{\theta}^{(1:2)}_{\text{ATTN}}$ updates $\boldsymbol{\Theta}, \boldsymbol{\beta}$ on the corresponding rows. i.e, if we denote $D_0 := q + p + 1$, the updates on row $D_0 + 1$ to row $D_0 + qp$ correspond to $\boldsymbol{\Theta}$, and the updates on row $D_0 + qp + 1$ to row $D_0 + qp + p$ correspond to $\boldsymbol{\beta}$.

**1) In the first layer, the transformer updates the current first-stage estimate $\hat{x}$.**

For $m = 2k - 1, k = 1, \ldots, p$, define $\boldsymbol{Q}^{(1)}_m, \boldsymbol{K}^{(1)}_m, \boldsymbol{V}^{(1)}_m$ such that:

$$
\boldsymbol{Q}^{(1)}_m \boldsymbol{h}^{(0)}_i = \begin{bmatrix} z_{i1} \\ \vdots \\ z_{iq} \\ \hat{x}^{(0)}_{ik} \\ \boldsymbol{0} \end{bmatrix}, \quad \boldsymbol{K}^{(1)}_m \boldsymbol{h}^{(0)}_j = \begin{bmatrix} \boldsymbol{\Theta}^{(0)}_{1k} \\ \vdots \\ \boldsymbol{\Theta}^{(0)}_{qk} \\ -1 \\ \boldsymbol{0} \end{bmatrix}, \quad \boldsymbol{V}^{(1)}_m \boldsymbol{h}^{(0)}_j = \boldsymbol{e}_{D_0 + qp + p + k}. \tag{49}
$$

For $m = 2k, k = 1, \ldots, p$, define $\boldsymbol{Q}^{(1)}_m, \boldsymbol{K}^{(1)}_m, \boldsymbol{V}^{(1)}_m$ such that:

$$
\boldsymbol{Q}^{(1)}_m \boldsymbol{h}^{(0)}_i = \begin{bmatrix} -z_{i1} \\ \vdots \\ -z_{iq} \\ \hat{x}^{(0)}_{ik} \\ \boldsymbol{0} \end{bmatrix}, \quad \boldsymbol{K}^{(1)}_m \boldsymbol{h}^{(0)}_j = \begin{bmatrix} \boldsymbol{\Theta}^{(0)}_{1k} \\ \vdots \\ \boldsymbol{\Theta}^{(0)}_{qk} \\ 1 \\ \boldsymbol{0} \end{bmatrix}, \quad \boldsymbol{V}^{(1)}_m \boldsymbol{h}^{(0)}_j = -\boldsymbol{e}_{D_0 + qp + p + k}, \tag{50}
$$

where $\boldsymbol{e}_j \in \mathbb{R}^D$ is the standard unit vector with only one 1 at the $j$-th coordinate. Note that the above are just linear transformations on $\boldsymbol{h}_i$ or $\boldsymbol{h}_j$, hence such matrices $\boldsymbol{Q}^{(1)}_m, \boldsymbol{K}^{(1)}_m, \boldsymbol{V}^{(1)}_m$ must exist.

Then we have:

$$
\begin{aligned}
\boldsymbol{h}_i^{(1)} &= \boldsymbol{h}_i^{(0)} + \sum_{m=1}^{2p} \frac{1}{n+1} \sum_{j=1}^{n+1} \sigma\left(\langle \boldsymbol{Q}_m^{(1)}\boldsymbol{h}_i^{(0)}, \boldsymbol{K}_m^{(1)}\boldsymbol{h}_j^{(0)}\rangle\right) \cdot \boldsymbol{V}_m^{(1)}\boldsymbol{h}_j^{(0)} \\
&= \boldsymbol{h}_i^{(0)} + \sum_{k=1}^{p} \frac{1}{n+1} \sum_{j=1}^{n+1} \left[\sigma\left(\sum_{l=1}^{q} z_{il}\boldsymbol{\Theta}_{lk}^{(0)} - \hat{x}_{ik}^{(0)}\right) - \sigma\left(-\sum_{l=1}^{q} z_{il}\boldsymbol{\Theta}_{lk}^{(0)} + \hat{x}_{ik}^{(0)}\right)\right] \cdot \boldsymbol{e}_{D_0+qp+p+k} \\
&= \boldsymbol{h}_i^{(0)} + \sum_{k=1}^{p} \left[\sum_{l=1}^{q} z_{il}\boldsymbol{\Theta}_{lk}^{(0)} - \hat{x}_{ik}^{(0)}\right] \boldsymbol{e}_{D_0+qp+p+k} \\
&= \boldsymbol{h}_i^{(0)} + \sum_{k=1}^{p} \left(\hat{x}_{ik}^{(1)} - \hat{x}_{ik}^{(0)}\right) \boldsymbol{e}_{D_0+qp+p+k}.
\end{aligned}
$$

Thus this layer correctly updates the first-stage prediction values $\hat{\boldsymbol{x}}_1^{(1)}, \ldots, \hat{\boldsymbol{x}}_{n+1}^{(1)}$, where $\hat{\boldsymbol{x}}_i^{(1)} := [\boldsymbol{Z}\boldsymbol{\Theta}^{(0)}]_{i,:} = \sum_{l=1}^{q} z_{il}\boldsymbol{\Theta}_{l,:}^{(0)}$. We will have:

$$
\boldsymbol{H}^{(1)} = \begin{bmatrix}
\boldsymbol{z}_1 & \cdots & \boldsymbol{z}_n & \boldsymbol{z}_{n+1} \\
\boldsymbol{x}_1 & \cdots & \boldsymbol{x}_n & \boldsymbol{x}_{n+1} \\
y_1 & \cdots & y_n & 0 \\
\boldsymbol{\Theta}_{:,1}^{(0)} & \cdots & \boldsymbol{\Theta}_{:,1}^{(0)} & \boldsymbol{\Theta}_{:,1}^{(0)} \\
\vdots & \vdots & \vdots & \vdots \\
\boldsymbol{\Theta}_{:,p}^{(0)} & \cdots & \boldsymbol{\Theta}_{:,p}^{(0)} & \boldsymbol{\Theta}_{:,p}^{(0)} \\
\boldsymbol{\beta}^{(0)} & \cdots & \boldsymbol{\beta}^{(0)} & \boldsymbol{\beta}^{(0)} \\
\hat{\boldsymbol{x}}_1^{(1)} & \cdots & \hat{\boldsymbol{x}}_n^{(1)} & \hat{\boldsymbol{x}}_{n+1}^{(1)} \\
1 & \cdots & 1 & 1 \\
1 & \cdots & 1 & 0
\end{bmatrix}.
$$

**2) In the second layer, the transformer does the gradient updates on the parameters $\boldsymbol{\Theta}$ and $\boldsymbol{\beta}$.**

For $m = 2k-1, k = 1, \ldots, p$, define $\boldsymbol{Q}_m^{(2)}, \boldsymbol{K}_m^{(2)}, \boldsymbol{V}_m^{(2)}$ such that:

$$
\boldsymbol{Q}_m^{(2)}\boldsymbol{h}_i^{(1)} = \begin{bmatrix} \boldsymbol{\Theta}_{:,k}^{(0)} \\ -1 \\ -1 \\ \vdots \\ \boldsymbol{0} \end{bmatrix}, \quad \boldsymbol{K}_m^{(2)}\boldsymbol{h}_j^{(1)} = \begin{bmatrix} \boldsymbol{z}_j \\ x_{jk}t_j \\ R(1-t_j) \\ \vdots \\ \boldsymbol{0} \end{bmatrix}, \quad \boldsymbol{V}_m^{(2)}\boldsymbol{h}_j^{(1)} = -(n+1)\eta \sum_{l=1}^{q} z_{jl}\boldsymbol{e}_{D_0+(k-1)q+l}.
$$

$$(51)$$

For $m = 2k, k = 1, \ldots, p$, define $\boldsymbol{Q}_m^{(2)}, \boldsymbol{K}_m^{(2)}, \boldsymbol{V}_m^{(2)}$ such that:

$$
\boldsymbol{Q}_m^{(2)}\boldsymbol{h}_i^{(1)} = \begin{bmatrix} -\boldsymbol{\Theta}_{:,k}^{(0)} \\ 1 \\ -1 \\ \vdots \\ \boldsymbol{0} \end{bmatrix}, \quad \boldsymbol{K}_m^{(2)}\boldsymbol{h}_j^{(1)} = \begin{bmatrix} \boldsymbol{z}_j \\ x_{jk}t_j \\ R(1-t_j) \\ \vdots \\ \boldsymbol{0} \end{bmatrix}, \quad \boldsymbol{V}_m^{(2)}\boldsymbol{h}_j^{(1)} = (n+1)\eta \sum_{l=1}^{q} z_{jl}\boldsymbol{e}_{D_0+(k-1)q+l},
$$

$$(52)$$

where $R = \max\limits_{\substack{i=1,\ldots,n+1 \\ t=0,1,\ldots}} \{\|\boldsymbol{\Theta}^{(t)\top}\boldsymbol{z}_i\|\}$. Then we have:

$$
\begin{aligned}
\sigma\left(\langle \boldsymbol{Q}_{2k-1}^{(2)}\boldsymbol{h}_i^{(1)}, \boldsymbol{K}_{2k-1}^{(2)}\boldsymbol{h}_j^{(1)}\rangle\right) &= \sigma\left(\boldsymbol{\Theta}_{:,k}^{(0)\top}\boldsymbol{z}_j - x_{jk}t_j - R(1-t_j)\right) \\
&= \sigma\left(\boldsymbol{\Theta}_{:,k}^{(0)\top}\boldsymbol{z}_j - x_{jk}\right)\mathbb{1}\{t_j = 1\} \\
&= \sigma\left(\boldsymbol{\Theta}_{:,k}^{(0)\top}\boldsymbol{z}_j - x_{jk}\right)t_j,
\end{aligned}
$$

and

$$\sigma\left(\langle \boldsymbol{Q}_{2k}^{(2)}\boldsymbol{h}_i^{(1)}, \boldsymbol{K}_{2k}^{(2)}\boldsymbol{h}_j^{(1)}\rangle\right) = \sigma\left(-\boldsymbol{\Theta}_{:,k}^{(0)\top}\boldsymbol{z}_j + x_{jk}t_j - R(1-t_j)\right)$$

$$= \sigma\left(-\boldsymbol{\Theta}_{:,k}^{(0)\top}\boldsymbol{z}_j + x_{jk}\right)\mathbb{1}\{t_j = 1\}$$

$$= \sigma\left(-\boldsymbol{\Theta}_{:,k}^{(0)\top}\boldsymbol{z}_j + x_{jk}\right)t_j.$$

So that

$$\sum_{m=1}^{2p} \sigma\left(\langle \boldsymbol{Q}_m^{(2)}\boldsymbol{h}_i^{(1)}, \boldsymbol{K}_m^{(2)}\boldsymbol{h}_j^{(1)}\rangle\right)\boldsymbol{V}_m^{(2)}\boldsymbol{h}_j^{(1)}$$

$$= -(n+1)t_j\eta\sum_{k=1}^{p}\left[\sigma\left(\boldsymbol{\Theta}_{:,k}^{(0)\top}\boldsymbol{z}_j - x_{jk}\right) - \sigma\left(-\boldsymbol{\Theta}_{:,k}^{(0)\top}\boldsymbol{z}_j + x_{jk}\right)\right]\cdot\sum_{l=1}^{q}z_{jl}\boldsymbol{e}_{D_0+(k-1)q+l}$$

$$= -(n+1)t_j\eta\sum_{k=1}^{p}\sum_{l=1}^{q}z_{jl}\left(\boldsymbol{\Theta}_{:,k}^{(0)\top}\boldsymbol{z}_j - x_{jk}\right)\boldsymbol{e}_{D_0+(k-1)q+l}.$$

Similarly, for $m = 2p+1, 2p+2$, define $\boldsymbol{Q}_m^{(2)}, \boldsymbol{K}_m^{(2)}, \boldsymbol{V}_m^{(2)}$ such that:

$$\boldsymbol{Q}_{2p+1}^{(2)}\boldsymbol{h}_i^{(1)} = \begin{bmatrix}\boldsymbol{\beta}^{(0)} \\ -1 \\ -1 \\ \vdots \\ \boldsymbol{0}\end{bmatrix}, \boldsymbol{K}_{2p+1}^{(2)}\boldsymbol{h}_j^{(1)} = \begin{bmatrix}\hat{\boldsymbol{x}}_j^{(1)} \\ y_jt_j \\ R'(1-t_j) \\ \vdots \\ \boldsymbol{0}\end{bmatrix}, \boldsymbol{V}_{2p+1}^{(2)}\boldsymbol{h}_j^{(1)} = -(n+1)\alpha\sum_{l=1}^{p}\hat{x}_{jl}^{(1)}\boldsymbol{e}_{D_0+qp+l},$$

(53)

$$\boldsymbol{Q}_{2p+2}^{(2)}\boldsymbol{h}_i^{(1)} = \begin{bmatrix}-\boldsymbol{\beta}^{(0)} \\ 1 \\ -1 \\ \vdots \\ \boldsymbol{0}\end{bmatrix}, \boldsymbol{K}_{2p+2}^{(2)}\boldsymbol{h}_j^{(1)} = \begin{bmatrix}\hat{\boldsymbol{x}}_j^{(1)} \\ y_jt_j \\ R'(1-t_j) \\ \vdots \\ \boldsymbol{0}\end{bmatrix}, \boldsymbol{V}_{2p+2}^{(2)}\boldsymbol{h}_j^{(1)} = (n+1)\alpha\sum_{l=1}^{p}\hat{x}_{jl}^{(1)}\boldsymbol{e}_{D_0+qp+l},$$

(54)

where $R' = \max\limits_{\substack{i=1,\dots,n+1 \\ t=0,1,\dots}}\{|\boldsymbol{\beta}^{(t)\top}\boldsymbol{x}_i|\}$. Then

$$\sigma\left(\langle \boldsymbol{Q}_{2p+1}^{(2)}\boldsymbol{h}_i^{(1)}, \boldsymbol{K}_{2p+1}^{(2)}\boldsymbol{h}_j^{(1)}\rangle\right) = \sigma\left(\boldsymbol{\beta}^{(0)\top}\hat{\boldsymbol{x}}_j^{(1)} - y_jt_j - R'(1-t_j)\right)$$

$$= \sigma\left(\boldsymbol{\beta}^{(0)\top}\hat{\boldsymbol{x}}_j^{(1)} - y_j\right)\mathbb{1}\{t_j = 1\}$$

$$= \sigma\left(\boldsymbol{\beta}^{(0)\top}\hat{\boldsymbol{x}}_j^{(1)} - y_j\right)t_j,$$

and

$$\sigma\left(\langle \boldsymbol{Q}_{2p+2}^{(2)}\boldsymbol{h}_i^{(1)}, \boldsymbol{K}_{2p+2}^{(2)}\boldsymbol{h}_j^{(1)}\rangle\right) = \sigma\left(-\boldsymbol{\beta}^{(0)\top}\hat{\boldsymbol{x}}_j^{(1)} + y_jt_j - R'(1-t_j)\right)$$

$$= \sigma\left(-\boldsymbol{\beta}^{(0)\top}\hat{\boldsymbol{x}}_j^{(1)} + y_j\right)\mathbb{1}\{t_j = 1\}$$

$$= \sigma\left(-\boldsymbol{\beta}^{(0)\top}\hat{\boldsymbol{x}}_j^{(1)} + y_j\right)t_j.$$

So that

$$\sum_{m=2p+1}^{2p+2} \sigma\left(\langle \boldsymbol{Q}_m^{(2)}\boldsymbol{h}_i^{(1)}, \boldsymbol{K}_m^{(2)}\boldsymbol{h}_j^{(1)}\rangle\right)\boldsymbol{V}_m^{(2)}\boldsymbol{h}_j^{(1)}$$

$$= -(n+1)t_j\alpha\left[\sigma\left(\boldsymbol{\beta}^{(0)\top}\hat{\boldsymbol{x}}_j^{(1)} - y_j\right) - \sigma\left(-\boldsymbol{\beta}^{(0)\top}\hat{\boldsymbol{x}}_j^{(1)} + y_j\right)\right]\cdot\sum_{l=1}^{p}\hat{x}_{jl}^{(1)}\boldsymbol{e}_{D_0+qp+l}$$

$$= -(n+1)t_j\alpha\sum_{l=1}^{p}\hat{x}_{jl}^{(1)}\left(\boldsymbol{\beta}^{(0)\top}\hat{\boldsymbol{x}}_j^{(1)} - y_j\right)\boldsymbol{e}_{D_0+qp+l}.$$

Thus the final output, for $i = 1, \dots, n+1$:

$$\boldsymbol{h}_i^{(2)} = \boldsymbol{h}_i^{(1)} + \sum_{m=1}^{2p+2} \frac{1}{n+1} \sum_{j=1}^{n} \sigma\left(\langle \boldsymbol{Q}_m^{(2)} \boldsymbol{h}_i^{(1)}, \boldsymbol{K}_m^{(2)} \boldsymbol{h}_j^{(1)} \rangle\right) \boldsymbol{V}_m^{(2)} \boldsymbol{h}_j^{(1)}$$

$$= \begin{bmatrix} \boldsymbol{z}_i \\ \boldsymbol{x}_i \\ y_i t_i \\ \boldsymbol{\Theta}_{:,1}^{(0)} - \eta \sum_{j=1}^{n} \boldsymbol{z}_j \left(\boldsymbol{z}_j^\top \boldsymbol{\Theta}_{:,1}^{(0)} - x_{j1}\right) \\ \vdots \\ \boldsymbol{\Theta}_{:,p}^{(0)} - \eta \sum_{j=1}^{n} \boldsymbol{z}_j \left(\boldsymbol{z}_j^\top \boldsymbol{\Theta}_{:,p}^{(0)} - x_{jp}\right) \\ \boldsymbol{\beta}^{(0)} - \alpha \sum_{j=1}^{n} \hat{\boldsymbol{x}}_j^{(1)} \left(\hat{\boldsymbol{x}}_j^{(1)\top} \boldsymbol{\beta}^{(0)} - y_j\right) \\ \hat{\boldsymbol{x}}_i^{(1)} \\ 1 \\ t_i \end{bmatrix}$$

$$= \begin{bmatrix} \boldsymbol{z}_i \\ \boldsymbol{x}_i \\ y_i t_i \\ \boldsymbol{\Theta}_{:,1}^{(0)} - \eta \boldsymbol{Z}^\top \left[\boldsymbol{Z}\boldsymbol{\Theta}^{(0)} - \boldsymbol{X}\right]_{:,1} \\ \vdots \\ \boldsymbol{\Theta}_{:,p}^{(0)} - \eta \boldsymbol{Z}^\top \left[\boldsymbol{Z}\boldsymbol{\Theta}^{(0)} - \boldsymbol{X}\right]_{:,p} \\ \boldsymbol{\beta}^{(0)} - \alpha \boldsymbol{\Theta}^{(0)\top} \boldsymbol{Z}^\top \left(\boldsymbol{Z}\boldsymbol{\Theta}^{(0)} \boldsymbol{\beta}^{(0)} - \boldsymbol{y}\right) \\ \hat{\boldsymbol{x}}_i^{(1)} \\ 1 \\ t_i \end{bmatrix}.$$

This corresponds to a one-step 2SLS GD update of $\boldsymbol{\Theta}, \boldsymbol{\beta}$, according to (7). Therefore, the final output matrix is:

$$\boldsymbol{H}^{(2)} = \begin{bmatrix} \boldsymbol{z}_1 & \cdots & \boldsymbol{z}_n & \boldsymbol{z}_{n+1} \\ \boldsymbol{x}_1 & \cdots & \boldsymbol{x}_n & \boldsymbol{x}_{n+1} \\ y_1 & \cdots & y_n & 0 \\ \boldsymbol{\Theta}_{:,1}^{(1)} & \cdots & \boldsymbol{\Theta}_{:,1}^{(1)} & \boldsymbol{\Theta}_{:,1}^{(1)} \\ \vdots & \vdots & \vdots & \vdots \\ \boldsymbol{\Theta}_{:,p}^{(1)} & \cdots & \boldsymbol{\Theta}_{:,p}^{(1)} & \boldsymbol{\Theta}_{:,p}^{(1)} \\ \boldsymbol{\beta}^{(1)} & \cdots & \boldsymbol{\beta}^{(1)} & \boldsymbol{\beta}^{(1)} \\ \hat{\boldsymbol{x}}_1^{(1)} & \cdots & \hat{\boldsymbol{x}}_n^{(1)} & \hat{\boldsymbol{x}}_{n+1}^{(1)} \\ 1 & \cdots & 1 & 1 \\ 1 & \cdots & 1 & 0 \end{bmatrix}.$$

Thus the proof is complete. We further note that in construction steps (49)(50)(51)(52)(53)(54), regardless of the initial values of $\boldsymbol{\Theta}^{(0)}, \boldsymbol{\beta}^{(0)}$, and $\hat{\boldsymbol{x}}^{(0)}$, the matrices $\boldsymbol{Q}_m^{(1:2)}, \boldsymbol{K}_m^{(1:2)}, \boldsymbol{V}_m^{(1:2)}$ do the same linear transformations on the input vectors. Therefore they are identical across different layers. $\qquad\square$

### B.3 PROOF OF THEOREM 3.3

**Lemma B.2** (Generalization of pretraining, from Theorem 20 in Bai et al. (2023))**.** Given optimization problm (12), with probability at least $1 - \zeta$, the solution $\hat{\boldsymbol{\theta}}$ satisfies:

$$L_{\mathsf{ICL}}(\hat{\boldsymbol{\theta}}) \le \inf_{\boldsymbol{\theta} \in \vartheta} L_{\mathsf{ICL}}(\boldsymbol{\theta}) + \mathcal{O}\left(B_y^2 \sqrt{\frac{L^2(MD^2 + DD')\log(2 + \max\{B_\theta, R, B_y\}) + \log(1/\zeta)}{N}}\right).$$

*Proof of Theorem 3.3.* We begin by showing the (clipped) 2SLS predictor achieves small excess loss under in-context distribution $\mathcal{P}$:

$$\mathbb{E}_{\mathcal{P}}\left[\left(\langle \text{clip}_{B_\beta}(\hat{\boldsymbol{\beta}}_{\text{2SLS}}), \boldsymbol{x}_{n+1}\rangle - y_{n+1}\right)^2\right]$$

$$= \mathbb{E}_{\mathcal{P}}\left[\left(\langle \text{clip}_{B_\beta}(\hat{\boldsymbol{\beta}}_{\text{2SLS}}) - \boldsymbol{\beta}, \boldsymbol{x}_{n+1}\rangle + \langle \boldsymbol{\beta}, \boldsymbol{x}_{n+1}\rangle - y_{n+1}\right)^2\right]$$

$$= \mathbb{E}_{\mathcal{P}}\left[\langle \text{clip}_{B_\beta}(\hat{\boldsymbol{\beta}}_{\text{2SLS}}) - \boldsymbol{\beta}, \boldsymbol{x}_{n+1}\rangle^2\right] + 2E_{\mathcal{P}}\left[\langle \text{clip}_{B_\beta}(\hat{\boldsymbol{\beta}}_{\text{2SLS}}) - \boldsymbol{\beta}, \boldsymbol{x}_{n+1}\rangle(\langle \boldsymbol{\beta}, \boldsymbol{x}_{n+1}\rangle - y_{n+1})\right]$$

$$\qquad + \mathbb{E}_{\mathcal{P}}\left[\left(\langle \boldsymbol{\beta}, \boldsymbol{x}_{n+1}\rangle - y_{n+1}\right)^2\right]$$

$$= \underbrace{\mathbb{E}_{\mathcal{P}}\left[\langle \text{clip}_{B_\beta}(\hat{\boldsymbol{\beta}}_{\text{2SLS}}) - \boldsymbol{\beta}, \boldsymbol{x}_{n+1}\rangle^2\right]}_{\text{Excess Loss}} + \mathbb{E}_{\mathcal{P}}\left[\left(\langle \boldsymbol{\beta}, \boldsymbol{x}_{n+1}\rangle - y_{n+1}\right)^2\right],$$

where $\mathbb{E}_{\mathcal{P}}[\langle \text{clip}_{B_\beta}(\hat{\boldsymbol{\beta}}_{\text{2SLS}}) - \boldsymbol{\beta}, \boldsymbol{x}_{n+1}\rangle(\langle \boldsymbol{\beta}, \boldsymbol{x}_{n+1}\rangle - y_{n+1})] = 0$ follows from the independence between $\langle \text{clip}_{B_\beta}(\hat{\boldsymbol{\beta}}_{\text{2SLS}}) - \boldsymbol{\beta}, \boldsymbol{x}_{n+1}\rangle$ and $(\langle \boldsymbol{\beta}, \boldsymbol{x}_{n+1}\rangle - y_{n+1})$ with $\mathbb{E}_{\mathcal{P}}[\langle \boldsymbol{\beta}, \boldsymbol{x}_{n+1}\rangle - y_{n+1}] = \mathbb{E}_{\mathcal{P}}[\epsilon_{n+1}] = 0$.

To bound the excess loss, we have

$$\mathbb{E}_{\mathcal{P}}\left[\langle \text{clip}_{B_\beta}(\hat{\boldsymbol{\beta}}_{\text{2SLS}}) - \boldsymbol{\beta}, \boldsymbol{x}_{n+1}\rangle^2\right] = \mathbb{E}_{\mathcal{P}}\left[\left\|\boldsymbol{x}_{n+1}^\top\left(\text{clip}_{B_\beta}(\hat{\boldsymbol{\beta}}_{\text{2SLS}}) - \boldsymbol{\beta}\right)\right\|^2\right]$$

$$\le \mathbb{E}_{\mathcal{P}}\left[\|\boldsymbol{x}_{n+1}\|^2 \left\|\text{clip}_{B_\beta}(\hat{\boldsymbol{\beta}}_{\text{2SLS}}) - \boldsymbol{\beta}\right\|^2\right]$$

$$= \mathbb{E}_{\mathcal{P}}\left[\|\boldsymbol{x}_{n+1}\|^2\right] \mathbb{E}_{\mathcal{P}}\left[\left\|\text{clip}_{B_\beta}(\hat{\boldsymbol{\beta}}_{\text{2SLS}}) - \boldsymbol{\beta}\right\|^2\right] \qquad (55)$$

$$\le \mathcal{O}\left(B_x^2\left(\frac{q}{n}\left(\frac{B_\beta^2}{K} + C^2(n)(\boldsymbol{\phi}^\top \boldsymbol{\Sigma}_u \boldsymbol{\phi} + \sigma_\epsilon^2)\right)\right)\right),$$

where the last inequality follows from (3).

Next, for the ICL loss, we have

$$L_{\text{ICL}}(\boldsymbol{\theta})$$

$$= \mathbb{E}_\pi \mathbb{E}_{\mathcal{P}}\left[\left(\widetilde{\text{TF}}_{\boldsymbol{\theta}}(\boldsymbol{H}) - y_{n+1}\right)^2\right]$$

$$= \mathbb{E}_\pi \mathbb{E}_{\mathcal{P}}\left[\left(\widetilde{\text{TF}}_{\boldsymbol{\theta}}(\boldsymbol{H}) - \langle \text{clip}_{B_\beta}(\hat{\boldsymbol{\beta}}_{\text{2SLS}}), \boldsymbol{x}_{n+1}\rangle + \langle \text{clip}_{B_\beta}(\hat{\boldsymbol{\beta}}_{\text{2SLS}}), \boldsymbol{x}_{n+1}\rangle - y_{n+1}\right)^2\right]$$

$$= \mathbb{E}_\pi\left\{\mathbb{E}_{\mathcal{P}}\left[\left(\widetilde{\text{TF}}_{\boldsymbol{\theta}}(\boldsymbol{H}) - \langle \text{clip}_{B_\beta}(\hat{\boldsymbol{\beta}}_{\text{2SLS}}), \boldsymbol{x}_{n+1}\rangle\right)^2\right] + \mathbb{E}_{\mathcal{P}}\left[\left(\langle \text{clip}_{B_\beta}(\hat{\boldsymbol{\beta}}_{\text{2SLS}}), \boldsymbol{x}_{n+1}\rangle - y_{n+1}\right)^2\right]\right.$$

$$\left. + 2\mathbb{E}_{\mathcal{P}}\left[\left(\widetilde{\text{TF}}_{\boldsymbol{\theta}}(\boldsymbol{H}) - \langle \text{clip}_{B_\beta}(\hat{\boldsymbol{\beta}}_{\text{2SLS}}), \boldsymbol{x}_{n+1}\rangle\right)\left(\langle \text{clip}_{B_\beta}(\hat{\boldsymbol{\beta}}_{\text{2SLS}}), \boldsymbol{x}_{n+1}\rangle - y_{n+1}\right)\right]\right\}$$

$$\le \mathbb{E}_\pi\left\{\mathbb{E}_{\mathcal{P}}\left[\left(\widetilde{\text{TF}}_{\boldsymbol{\theta}}(\boldsymbol{H}) - \langle \text{clip}_{B_\beta}(\hat{\boldsymbol{\beta}}_{\text{2SLS}}), \boldsymbol{x}_{n+1}\rangle\right)^2\right] + \mathbb{E}_{\mathcal{P}}\left[\langle \text{clip}_{B_\beta}(\hat{\boldsymbol{\beta}}_{\text{2SLS}}) - \boldsymbol{\beta}, \boldsymbol{x}_{n+1}\rangle^2\right] + \mathbb{E}_{\mathcal{P}}\left[\left(\langle \boldsymbol{\beta}, \boldsymbol{x}_{n+1}\rangle - y_{n+1}\right)^2\right]\right.$$

$$\left. + 2\mathbb{E}_{\mathcal{P}}\left[\left|\widetilde{\text{TF}}_{\boldsymbol{\theta}}(\boldsymbol{H}) - \langle \text{clip}_{B_\beta}(\hat{\boldsymbol{\beta}}_{\text{2SLS}}), \boldsymbol{x}_{n+1}\rangle\right|\right]\mathbb{E}_{\mathcal{P}}\left[\left|\langle \text{clip}_{B_\beta}(\hat{\boldsymbol{\beta}}_{\text{2SLS}}), \boldsymbol{x}_{n+1}\rangle - y_{n+1}\right|\right]\right\}$$

$$\le \mathbb{E}_\pi\left\{\mathbb{E}_{\mathcal{P}}\left[\left(\widetilde{\text{TF}}_{\boldsymbol{\theta}}(\boldsymbol{H}) - \langle \text{clip}_{B_\beta}(\hat{\boldsymbol{\beta}}_{\text{2SLS}}), \boldsymbol{x}_{n+1}\rangle\right)^2\right] + \mathbb{E}_{\mathcal{P}}\left[\langle \text{clip}_{B_\beta}(\hat{\boldsymbol{\beta}}_{\text{2SLS}}) - \boldsymbol{\beta}, \boldsymbol{x}_{n+1}\rangle^2\right] + \mathbb{E}_{\mathcal{P}}\left[\left(\langle \boldsymbol{\beta}, \boldsymbol{x}_{n+1}\rangle - y_{n+1}\right)^2\right]\right.$$

$$\left. + 2\mathbb{E}_{\mathcal{P}}\left[\left|\widetilde{\text{TF}}_{\boldsymbol{\theta}}(\boldsymbol{H}) - \langle \text{clip}_{B_\beta}(\hat{\boldsymbol{\beta}}_{\text{2SLS}}), \boldsymbol{x}_{n+1}\rangle\right|\right]\left(\mathbb{E}_{\mathcal{P}}\left[\left|\langle \text{clip}_{B_\beta}(\hat{\boldsymbol{\beta}}_{\text{2SLS}}) - \boldsymbol{\beta}, \boldsymbol{x}_{n+1}\rangle\right|\right] + \mathbb{E}_{\mathcal{P}}\left[\left|\langle \boldsymbol{\beta}, \boldsymbol{x}_{n+1}\rangle - y_{n+1}\right|\right]\right)\right\}.$$

$$(56)$$

From Corollary 3.1, we know that there exists a $L = 2\bar{L} + 1$-layer attention-only transformer model $\boldsymbol{\theta}$, with $M = 2(p + 1)$ heads, and embedding dimension $D = qp + 3p + q + 3$, such that for any $\boldsymbol{H}$, given any learning rates $\alpha, \eta$ and $\Lambda$ as defined in Equation (8), the following holds[4]:

$$\left| \widetilde{\mathsf{TF}}_{\boldsymbol{\theta}}(\boldsymbol{H}) - \langle \mathsf{clip}_{B_\beta}(\hat{\boldsymbol{\beta}}_{\mathsf{2SLS}}), \boldsymbol{x}_{n+1} \rangle \right| \leq \mathcal{O}\left( B_x \Lambda^{\bar{L}} \right).$$

Denote $\Lambda^\star := \min_{\alpha, \eta} \mathbb{E}_\pi \mathbb{E}_\mathcal{P} [\Lambda | \boldsymbol{H}, \alpha, \eta]$, then under $\alpha^\star, \eta^\star$, we have:

$$\mathbb{E}_\pi \mathbb{E}_\mathcal{P} \left[ \left| \widetilde{\mathsf{TF}}_{\boldsymbol{\theta}}(\boldsymbol{H}) - \langle \mathsf{clip}_{B_\beta}(\hat{\boldsymbol{\beta}}_{\mathsf{2SLS}}), \boldsymbol{x}_{n+1} \rangle \right| \right] \leq \mathcal{O}\left( B_x (\Lambda^\star)^{\bar{L}} \right), \tag{57}$$

and

$$\mathbb{E}_\pi \mathbb{E}_\mathcal{P} \left[ \left( \widetilde{\mathsf{TF}}_{\boldsymbol{\theta}}(\boldsymbol{H}) - \langle \mathsf{clip}_{B_\beta}(\hat{\boldsymbol{\beta}}_{\mathsf{2SLS}}), \boldsymbol{x}_{n+1} \rangle \right)^2 \right] \leq \mathcal{O}\left( B_x^2 \mu_{\Lambda,2}^\star \right), \tag{58}$$

where $\mu_{\Lambda,2}^\star := \mathbb{E}_\pi \mathbb{E}_\mathcal{P} \left[ \Lambda^{2\bar{L}} | \boldsymbol{H}, \alpha^\star, \eta^\star \right]$ is close to 0.

With condition (13), from Cauchy-Schwarz inequality, we have:

$$\mathbb{E}_\pi \mathbb{E}_\mathcal{P} \left[ \left| \langle \boldsymbol{\beta}, \boldsymbol{x}_{n+1} \rangle - y_{n+1} \right| \right] \leq \mathbb{E}_\pi \left[ \sqrt{\mathbb{E}_\mathcal{P} \left( \epsilon_{n+1}^2 \right)} \right] = \mathbb{E}_\pi [\sigma_\epsilon] \leq \tilde{\sigma}_\epsilon. \tag{59}$$

Also, from (55), we have:

$$\mathbb{E}_\pi \mathbb{E}_\mathcal{P} \left[ \langle \mathsf{clip}_{B_\beta}(\hat{\boldsymbol{\beta}}_{\mathsf{2SLS}}) - \boldsymbol{\beta}, \boldsymbol{x}_{n+1} \rangle^2 \right] \leq \mathcal{O}\left( B_x^2 \left( \frac{q}{n} \left( \frac{B_\beta^2}{K} + C^2(n)\tilde{\sigma}^2 \right) \right) \right). \tag{60}$$

Further,

$$\mathbb{E}_\pi \mathbb{E}_\mathcal{P} \left[ \left| \langle \mathsf{clip}_{B_\beta}(\hat{\boldsymbol{\beta}}_{\mathsf{2SLS}}) - \boldsymbol{\beta}, \boldsymbol{x}_{n+1} \rangle \right| \right] \leq \sqrt{\mathbb{E}_\pi \mathbb{E}_\mathcal{P} \left[ \langle \mathsf{clip}_{B_\beta}(\hat{\boldsymbol{\beta}}_{\mathsf{2SLS}}) - \boldsymbol{\beta}, \boldsymbol{x}_{n+1} \rangle^2 \right]}$$
$$\leq \mathcal{O}\left( B_x \sqrt{\frac{q}{n} \left( \frac{B_\beta^2}{K} + C^2(n)\tilde{\sigma}^2 \right)} \right). \tag{61}$$

Finally, with (57)(58)(59)(60)(61), rearranging the terms in (56), we have:

$$L_{\mathsf{ICL}}(\boldsymbol{\theta}) - \mathbb{E}_\pi \mathbb{E}_\mathcal{P} \left[ (y_{n+1} - \langle \boldsymbol{\beta}, \boldsymbol{x}_{n+1} \rangle)^2 \right]$$
$$\leq \mathcal{O}\left( B_x^2 \left( \mu_{\Lambda,2}^\star + (\Lambda^\star)^{\bar{L}} \sqrt{\frac{q}{n} \left( \frac{B_\beta^2}{K} + C^2(n)\tilde{\sigma}^2 \right)} + \frac{q}{n} \left( \frac{B_\beta^2}{K} + C^2(n)\tilde{\sigma}^2 \right) \right) + B_x (\Lambda^\star)^{\bar{L}} \tilde{\sigma}_\epsilon \right).$$
$$\leq \mathcal{O}\left( (\Lambda^\star)^{\bar{L}} \left( B_x^2 \sqrt{\frac{q}{n} \left( \frac{B_\beta^2}{K} + C^2(n)\tilde{\sigma}^2 \right)} + B_x \tilde{\sigma}_\epsilon \right) + B_x^2 \left( \frac{q}{n} \left( \frac{B_\beta^2}{K} + C^2(n)\tilde{\sigma}^2 \right) + \mu_{\Lambda,2}^\star \right) \right).$$
$$\tag{62}$$

Thus combining Lemma B.2 with (62) completes the proof. $\qquad\square$

## C    ADDITIONAL EXPERIMENTS AND DISCUSSIONS

For all experiments in this section, to be consistent with our main experiment in Section 4, we generate $n = 50$ training samples with $p = 5, q = 10$, following the data generating process described in Algorithm 1. The task parameters $\boldsymbol{\Theta}, \boldsymbol{\beta}, \boldsymbol{\Phi}, \boldsymbol{\phi}$ are sampled from standard Gaussian distribution, the covariance matrices $\boldsymbol{\Sigma}_z, \boldsymbol{\Sigma}_u, \boldsymbol{\Sigma}_\omega$ are set to be identity matrices, and the noise level $\sigma_\epsilon$ is set to 1.

---

[4]The clipping bound on $\hat{\boldsymbol{\beta}}_{\mathsf{2SLS}}$ can be matched by adjusting the clipping threshold on the last layer of $\widetilde{\mathsf{TF}}_{\boldsymbol{\theta}}$.

## C.1 Simulations Verifying Theorem 3.1

We use the GD-based 2SLS method (7) to estimate the causal effect $\beta$. For the simulated data, we calculate the following metrics:

$$\frac{2}{\sigma_{\max}^2(\boldsymbol{Z}\hat{\boldsymbol{\Theta}})} = 0.0016, \frac{2}{\sigma_{\max}^2(\boldsymbol{Z})} = 0.0212.$$

By Theorem 3.1, the gradient descent converges when $\alpha \in (0, 0.0016)$ and $\eta \in (0, 0.0212)$. The overall convergence rate is determined by $\Lambda := \max\{\gamma(\alpha), \kappa(\eta)\}$, where

$$\gamma(\alpha) := \rho\left(\boldsymbol{I} - \alpha\hat{\boldsymbol{\Theta}}^\top \boldsymbol{Z}^\top \boldsymbol{Z}\hat{\boldsymbol{\Theta}}\right),$$
$$\kappa(\eta) := \rho\left(\boldsymbol{I} - \eta\boldsymbol{Z}^\top \boldsymbol{Z}\right).$$

We first set $\alpha = 0.0012$ and vary $\eta$. The corresponding convergence rates are determined by $\Lambda = \max(0.87, \kappa(\eta))$. Next, we set $\eta = 0.01$ and vary $\alpha$. The corresponding convergence curves are determined by $\Lambda = \max(\gamma(\alpha), 0.82)$. We compare the estimates $\hat{\beta}^{(t)}$ with the 2SLS estimate $\hat{\beta}_{2SLS}$ as the iteration proceeds. The convergence results are shown in Figure 3.

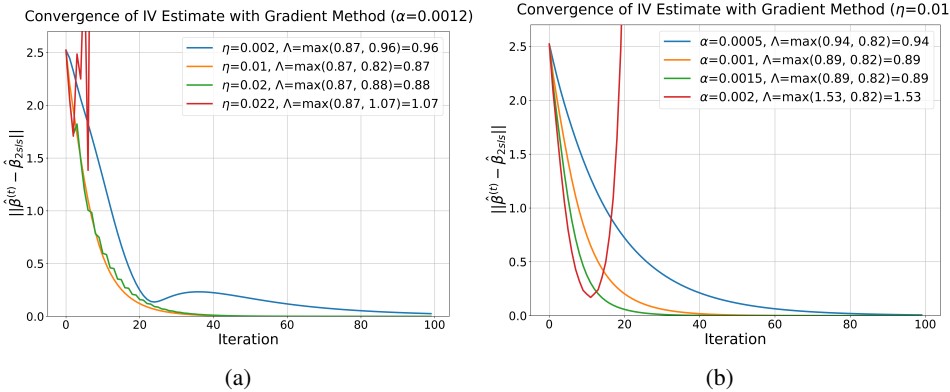

Figure 3: The convergence of the GD-based 2SLS method with (a) fixed $\alpha = 0.0012$ and varying $\eta$ and (b) fixed $\eta = 0.01$ and varying $\alpha$.

The results in Figure 3 are consistent with our theoretical analysis in Theorem 3.1. It is worth noting that in Figure 3a, when $\eta$ is relatively large (or small), the convergence curves exhibit some suiggly patterns. This is due to the innerloop updates (7a) are converging faster (or slower) than the outer loop updates (7b). However, the overall convergence rate is still determined by $\Lambda$. This pattern doesn't appear in Figure 3b as we set $\eta$ to be a moderate value, which ensures that the inner loop and outer loop converge synchronously.

Next, we show the bias of the GD estimator. For better convergence, we set $\alpha^\star = \frac{1}{\sigma_{\max}^2(\boldsymbol{Z}\hat{\boldsymbol{\Theta}})}$ and $\eta^\star = \frac{1}{\sigma_{\max}^2(\boldsymbol{Z})}$. We compare the biases of the GD estimator with $n = 50, 100, 150$ in-context samples. The results are shown in Figure 4.

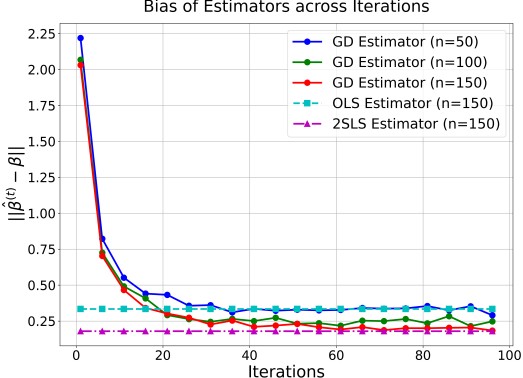

Figure 4: The convergence of the GD-based 2SLS method with $\alpha^\star = \frac{1}{\sigma^2_{\max}(\mathbf{Z}\hat{\mathbf{\Theta}})}$ and $\eta^\star = \frac{1}{\sigma^2_{\max}(\mathbf{Z})}$. The biases of 2SLS estimator and OLS estimator at $n = 150$ are plotted for comparison.

## C.2    DISCUSSIONS ON 2SLS WITH $\ell_2$-REGULARIZATION

In this section, we briefly discuss a generalization of our analysis to the case where the 2SLS estimator is regularized by the $\ell_2$ penalty (Ridge 2SLS). For this case, the bi-level optimization problem Equation (6) is modified as follows:

$$\min_{\boldsymbol{\beta}} \quad \mathcal{L}(\boldsymbol{\beta}) = \frac{1}{n}\sum_{i=1}^{n}(y_i - \boldsymbol{z}_i^\top \hat{\mathbf{\Theta}}\boldsymbol{\beta})^2 + \frac{1}{2}\lambda\|\boldsymbol{\beta}\|^2,$$

$$\text{where} \quad \hat{\mathbf{\Theta}} := \underset{\mathbf{\Theta}}{\arg\min} \quad \frac{1}{n}\sum_{j=1}^{n}(\boldsymbol{x}_j - \boldsymbol{z}_j^\top \mathbf{\Theta})^2 + \frac{1}{2}\tau\|\mathbf{\Theta}\|_F^2,$$

where $\lambda, \tau \geq 0$ are regularization parameters. To solve this problem, the GD updates in Equation (7) is modified as follows:

$$\mathbf{\Theta}^{(t+1)} = \mathbf{\Theta}^{(t)} - \eta\left[\mathbf{Z}^\top(\mathbf{Z}\mathbf{\Theta}^{(t)} - \mathbf{X}) + \tau\mathbf{\Theta}^{(t)}\right], \tag{63a}$$

$$\boldsymbol{\beta}^{(t+1)} = \boldsymbol{\beta}^{(t)} - \alpha\left[\mathbf{\Theta}^{(t)\top}\mathbf{Z}^\top(\mathbf{Z}\mathbf{\Theta}^{(t)}\boldsymbol{\beta}^{(t)} - \mathbf{Y}) + \lambda\boldsymbol{\beta}^{(t)}\right]. \tag{63b}$$

The only difference between Equation (7) and Equation (63) is the additional terms $\tau\mathbf{\Theta}^{(t)}$ in Equation (63a), and $\lambda\boldsymbol{\beta}^{(t)}$ in Equation (63b). The convergence analysis of the $\ell_2$-regularized GD updates in Equation (63) can be conducted in a similar manner as in Theorem 3.1. The only difference is that the convergence rate $\Lambda$ is now determined by the spectral radiuses of $\mathbf{I} - \eta(\mathbf{Z}^\top\mathbf{Z} + \tau\mathbf{I})$ and $\mathbf{I} - \alpha(\hat{\mathbf{\Theta}}^\top\mathbf{Z}^\top\mathbf{Z}\hat{\mathbf{\Theta}} + \lambda\mathbf{I})$, respectively.

With the same configuration as in Theorem 3.2 but adding $p + 1$ attention heads in the second layer (i.e. the second layer needs $3p + 3$ heads), we can show that transformers are able to implement the $\ell_2$-regularized GD updates in Equation (63). The proof follows directly from Appendix B.2, with the construction of the new attention heads in the second layer as follows.

For $m = 2p + 2 + k, k = 1, \ldots, p$, define $\mathbf{Q}_m^{(2)}, \mathbf{K}_m^{(2)}, \mathbf{V}_m^{(2)}$ such that:

$$\mathbf{Q}_m^{(2)}\boldsymbol{h}_i^{(1)} = \begin{bmatrix}1\\\mathbf{0}\end{bmatrix}, \mathbf{K}_m^{(2)}\boldsymbol{h}_i^{(1)} = \begin{bmatrix}1\\\mathbf{0}\end{bmatrix}, \mathbf{V}_m^{(2)}\boldsymbol{h}_i^{(1)} = -\eta\tau\sum_{l=1}^{q}\Theta_{lk}^{(0)}\boldsymbol{e}_{D_0+(k-1)q+l}.$$

For $m = 3p + 3$, define $\mathbf{Q}_{3p+3}^{(2)}, \mathbf{K}_{3p+3}^{(2)}, \mathbf{V}_{3p+3}^{(2)}$ such that:

$$\mathbf{Q}_{3p+3}^{(2)}\boldsymbol{h}_i^{(1)} = \begin{bmatrix}1\\\mathbf{0}\end{bmatrix}, \mathbf{K}_{3p+3}^{(2)}\boldsymbol{h}_i^{(1)} = \begin{bmatrix}1\\\mathbf{0}\end{bmatrix}, \mathbf{V}_{3p+3}^{(2)}\boldsymbol{h}_i^{(1)} = -\alpha\lambda\sum_{l=1}^{q}\beta_l^{(0)}\boldsymbol{e}_{D_0+qp+l}.$$

Then the remaining proof follows exactly the same as Appendix B.2. This result indicates that transformers are potentially capable of handling multicollinearity in IV regression problem. We conduct experiments to validate this and the results are shown in Appendix C.3.

### C.3 Experiments on Multicollinearity IV Problem

As a supplement to Section 4.2, we examine the case where multicollinearity occurs in the IV regression problem. We generate the test prompts in the same way using Algorithm 1, but introduce multicollinearity in the endogenous variable $x$ and instrument $z$.

Specifically, we first generate 4 columns of $X$, and 9 columns of $Z$, and then set $X_{:,5} \sim \mathcal{N}(2X_{:,4}, 10^{-6}I_n)$, and $Z_{:,10} \sim \mathcal{N}(2Z_{:,9}, 10^{-6}I_n)$. The results are shown in Figure 5a. As shown in Figure 5a, both ordinary OLS and 2SLS estimators fail to estimate the coefficients, while the trained transformer model is still able to provide consistent predictions and coefficient estimates.

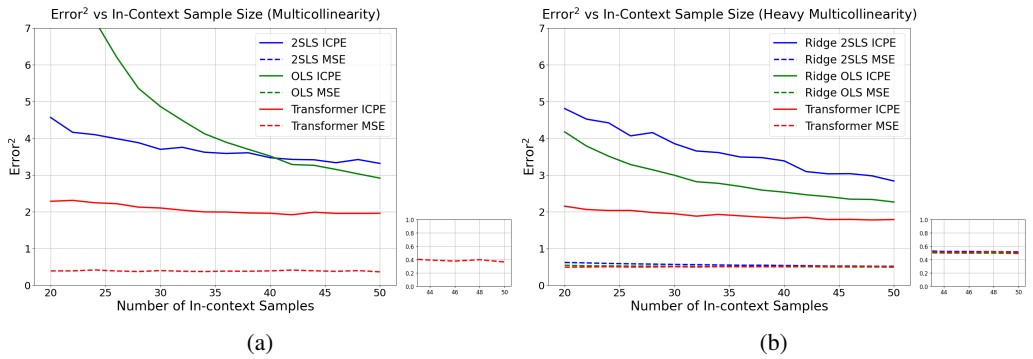

(a)                                     (b)

Figure 5: The ICL performance of the trained transformer model in endogeneity tasks with multi-collinearity. (a) 1 collinear column in $X$, and 1 collinear column in $Z$. Note that the coefficient MSEs for 2SLS and OLS are both out of range. (b) 2 collinear columns in $X$, and 5 collinear columns in $Z$. We compare the performance to the $\ell_2$-regularized 2SLS and OLS estimators. The curves are averaged over 500 simulations.

We further examine a more difficult case where heavy multicollinearity occurs. We first generate 3 column of $X$, and 5 column of $Z$, and then set $X_{:,j} \sim \mathcal{N}(2X_{:,j-2}, 10^{-6}I_n)$ for $j = 4, 5$, and $Z_{:,j} \sim \mathcal{N}(2Z_{:,j-5}, 10^{-6}I_n)$ for $j = 6, 7, 8, 9, 10$. For better comparisons, we now compare the performance of the trained transformer model to the $\ell_2$-regularized 2SLS and OLS estimators (with all regularization parameters set to 1). The results are shown in Figure 5b.

These results suggest that the trained transformer model is capable to handle multicollinearity in IV regression problems, even though it has not been specifically trained with multicollinearity tasks.

### C.4 Experiments on Complex Non-Linear IV Problem

As a supplement to Section 4.2, we examine a more complex scenario where the instrument $z$ has non-linear effect on the endogenous regressor $x$. We consider the following data generating process:

$$y = \langle \beta, x \rangle + \epsilon_1, \quad \text{and} \quad x = g(z) + \epsilon_2,$$

where $g : \mathbb{R}^q \to \mathbb{R}^p$ is a two-layer fully connected neural network with ReLU activation function. Similar to Section 4, the test prompts are generated using Algorithm 1, with all task parameters and weights of neural network sampled from standard Gaussian distribution. The results are shown in Figure 6. From this figure, we can see that the trained transformer model still achieves optimal performance in this complex non-linear setting.

### C.5 Experiments on Varying Endogeneity Strength

As a supplement to Section 4.2, we examine the performance of the trained transformer model in standard IV tasks with varying endogeneity strengths. The strength of endogeneity is determined

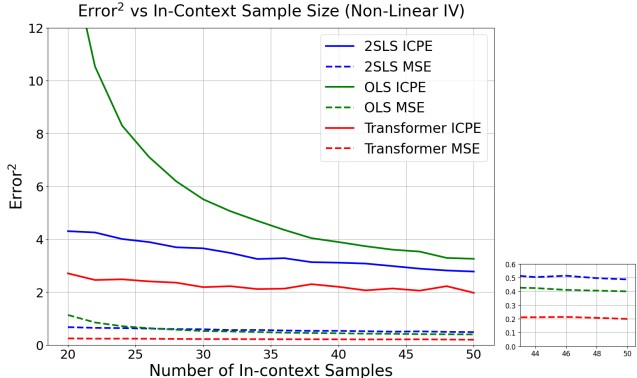

Figure 6: The ICL performance of the trained transformer model in complex non-linear endogeneity tasks where the IV has non-linear effect on the endogenous variable. The curves are averaged over 500 simulations.

by the correlation between $x$ and the endogenous error $\epsilon_1$. To vary the endogeneity strength, in Algorithm 1, we multiply $u$ by a factor $r \in (0, 2)$ when generating test prompts. The results are shown in Figure 7, which illustrates that the trained transformer model is comparable with the optimal 2SLS estimator in these standard IV tasks, regardless of the endogeneity level.

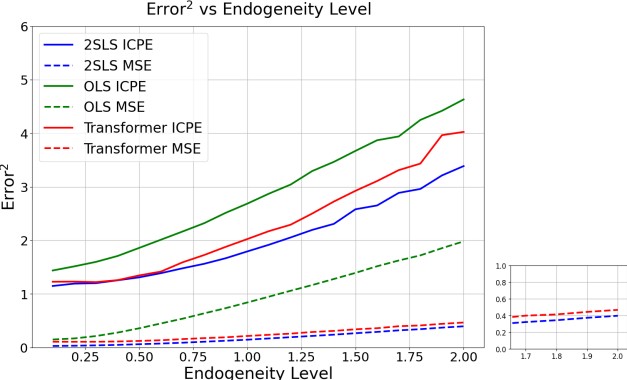

Figure 7: The ICL performance of the trained transformer model in tasks with varying endogeneity strengths. The curves are averaged over 500 simulations.

## C.6   EXPERIMENTS ON REAL-WORLD DATASET

In this section we provide an example to illustrate how the pretrained transformer model can be applied to a real-world dataset. We use the dataset from the study of Angrist & Evans (1998). This study investigates the effect of childbearing on labor supply. For demonstration purpose, we consider a simplified setup. We focus on a subset of the dataset that contains 6421 samples from Alabama. The outcome variable $y$ is mother's labor supply (number of working weeks in a year divided by 52), the endogenous variable $x$ is the number of children ($\geq 2$), and the instrument $z$ is an indicator variable of whether the first and second children are of the same sex[5].

---

[5]Research found that parents of same-sex siblings are significantly more likely to go on to have an additional child (Westoff & Parke, 1972), while it is not directly correlated with mother's labor supply as mixture of sex of the first two children can be considered as randomly assigned.

For each run we randomly select 50 samples from the dataset, and make the boxplot of the estimated $\beta$ over 500 runs[6]. As the ground truth effect $\beta$ is unknown, we compare them to the OLS and 2SLS estimates over all samples. The results are shown in Figure 8.

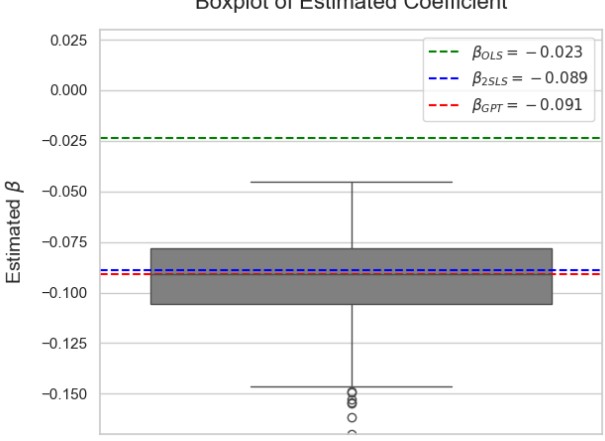

Figure 8: The boxplot of transformer's estimates over 500 runs on the labor supply dataset, comparing to the OLS and 2SLS estimates. $\beta_{\text{GPT}}$ is taken to be the median of all runs. The gray box represents the interquartile range, where the middle 50% of the estimated values fall. The whiskers of the box indicate the spread of the estimates. Any points falling outside of the whisker can be considered as outliers.

The final estimate $\beta_{\text{GPT}} = -0.091$, which suggests that with each increase in the number of children, the mother's labor supply is expected to drop 9.1% (approximately 4.73 weeks per year). This result is closer to the 2SLS estimate $\beta_{\text{2SLS}} = -0.089$ than the OLS estimate $\beta_{\text{OLS}} = -0.023$. This example demonstrates the potential of the pretrained transformer model in handling real-world IV problems.

## C.7 EXPERIMENTAL DETAIL

The training of the transformer in our experiment was conducted on a Windows 11 machine with the following specifications:

- GPU: NVIDIA GeForce RTX 4090
- CPU: Intel Core i9-14900KF
- Memory: 32 GB DDR5, 5600MHz

The training process took around 10 hours.

---

[6]For large enough model that can fit in the entire dataset, this step can be ignored. As shown in the simulation study in Section 4, a single estimate is expected to perform at least as good as 2SLS estimator, given the same number of samples.

