# OpenReview forum: "Transformers Handle Endogeneity in In-Context Linear Regression"
_ICLR.cc/2025/Conference — ICLR 2025 Poster_

### Official Review · Reviewer_XFtG · 2024-11-01

**Soundness:** 3
**Presentation:** 3
**Contribution:** 3
**Rating:** 6
**Confidence:** 3

**Summary:**

The paper investigates the ability of transformers to address endogeneity in in-context linear regression, proposing that transformers can emulate the two-stage least squares (2SLS) method through gradient-based optimization. Key contributions include demonstrating how transformers can handle endogeneity using instrumental variables, proposing an in-context pretraining scheme with theoretical guarantees for minimal excess loss, and showcasing robust performance in various endogeneity scenarios, including weak and non-linear instrumental variables​.

**Strengths:**

(1) This paper creatively combines transformer architectures with econometric techniques, specifically instrumental variables, to address endogeneity—a novel approach that extends transformers' applicability beyond traditional machine learning domains.

(2) The authors provide rigorous theoretical backing, including a bi-level optimization framework, and offer non-asymptotic error bounds, supporting their claims with comprehensive experiments that validate the model's performance against standard benchmarks like 2SLS.

(3)  By demonstrating that transformers can not only handle endogeneity but also generalize to complex scenarios (e.g., weak and non-linear IV), this work highlights the potential of transformers as a robust tool in econometrics, broadening the scope of in-context learning applications.

**Weaknesses:**

(1) The paper provides strong theoretical foundations but lacks practical guidance for implementation. More details on the parameter settings, model configurations, and optimization process would enhance reproducibility and help readers better understand how to apply the proposed method.

(2) While the theoretical contributions are thorough, the presentation is complex and could benefit from simplification or visual aids. This would make the bi-level optimization framework and convergence properties more accessible to a broader audience.

(3)  The paper does not address the scalability of the proposed transformer model for handling endogeneity. Including an analysis of computational efficiency or memory requirements would help clarify its suitability for larger datasets or higher-dimensional problems.

**Questions:**

(1) How does the proposed transformer model handle cases with highly correlated instruments? In many practical applications, instrumental variables may exhibit multicollinearity, which could impact the model's stability and coefficient estimates. Could the authors clarify whether the model’s training or architecture includes mechanisms to address such cases?

(2) What led to the choice of hyperparameters in the pretraining scheme? The paper presents a theoretical guarantee of minimal excess loss but does not detail the rationale behind the selection of specific hyperparameters, such as learning rates or number of transformer layers. Could the authors provide insights or guidelines on choosing these parameters for optimal performance?

(3) How sensitive is the model to changes in the strength of endogeneity? While the experiments demonstrate robustness to varying IV strengths, it would be valuable to understand if there is a threshold or particular cases where the transformer’s performance degrades. Could the authors elaborate on the model's sensitivity to different degrees of endogeneity?

---

> ### Author Response · Authors · 2024-11-22
> **Author Response**
>
> Thank you for your valuable comments on our paper and the positive evaluation. We appreciate the opportunity to address your concerns.
>
> **[W3] Scalability Analysis**
>
> Thank you for your suggestion. As a reference, the training of the transformer in our experiment was conducted on a Windows 11 machine with the following specifications:
>
> - GPU: NVIDIA GeForce RTX 4090
> - CPU: Intel Core i9-14900KF
> - Memory: 32 GB DDR5, 5600MHz
>
> The training process took around 10 hours.  These are mentioned in our supplementary "readme" file. We also added this information to Appendix C.8.
>
> We also emphasize that our current work is focused on the existence of transformer implementing 2SLS. In practice, the parameters of transformer are learned by gradient descent-type algorithm. Understanding the training dynamics, including scalability, computational complexity and memory efficiency is certainly an important question that we plan to work on in the future. However, it is clearly beyond the scope of the current paper.
>
> **[Q1] How does the proposed transformer model handle cases with highly correlated instruments?**
>
> The issue of multicollinearity can be resolved by ridge regression. Although not directly specified in our paper, previous works have studied that transformers have intrinsic mechanism to implement ridge regression in-context (for example, see Bai et al. (2024), Section 3.1). This mechanism can be easily extended to 2SLS case by considering each stage as a constrained optimization problem. We provide a dicussion in Appendix C.3, and we add some experiments to examine the capability of transformer in handling  IV regression with multicollinearity in Appendix C.4.
>
> **[Q2] What led to the choice of hyperparameters in the pretraining scheme?**
>
> Our hyperparameter settings of number of attention heads, embedding space dimension, and number of transformer layer in each transformer block follow the theoretical results of Theorem 3.2. For the number of transformer blocks needed, one may refer to Appendix C.2. The empirical results suggest that 20 iterations (blocks) are sufficient for the algorithm to converge. We set the number of blocks to be 10 in our experiments which is mainly out of the consideration of training time, and it turns out that the pretrained model is powerful enough. While we expect the model performance to be further improved with a larger model size, our experiment focuses on the minimum sufficient case directed by our theoretical results.
>
> Note that the learning rates $\alpha, \eta$ in GD-2SLS are not hyperparameters of our transformer model. Instead, for any values of $\alpha$ and $\eta$, there exists a corresponding transformer architecture to implement the GD-2SLS algorithm, so we claim that there exists a transformer architecture that can implement GD-2SLS with global optimum $\alpha$ and $\eta$, determined by the training data distributions. For specific training of the transformer model, as a common practice, we use the Adam optimizer with default initial learning rate 0.001.
>
> **[Q3] How sensitive is the model to changes in the strength of endogeneity?**
>
> Thank you for this good question. Based on your question, we provide an experiment in Appendix C.6 to examine the sensitivity of the transformer model to the strength of endogeneity. The results show that the transformer model is robust to different endogeneity levels, and the performance is comparable with the optimal 2SLS estimator.
>
> **We sincerely hope that we have addressed many of your questions, in which case, we would greatly appreciate if you could increase your score as you see fit.**

---

> > ### Author Response · Authors · 2024-11-25
> > **Gentle reminder**
> >
> > Dear Reviewer XFtG,
> >
> > As we are getting closer to the end of the discussion period (Nov 26th), we were wondering if you could let us know if we have sufficiently addressed your questions (we would also greatly appreciate if you could increase your score as you see fit to reflect this).
> >
> > Please let us know if you have any additional questions and we will be happy to address them as much as we can before the deadline. Thank you and look forward to hearing from you.
> >
> > Best regards,\
> > Authors

---

> > > ### Author Response · Authors · 2024-11-29
> > > **gentle second reminder**
> > >
> > > Dear Reviewer XFtG,
> > >
> > > As we are getting closer to the end of the updated discussion period (Dec 2nd), we were wondering if you could let us know if we have sufficiently addressed your questions (we would also greatly appreciate if you could increase your score as you see fit to reflect this).
> > >
> > > **We would greatly appreciate your opinion given that you are most confident reviewer for our submission.**
> > >
> > > Please let us know if you have any additional questions and we will be happy to address them as much as we can before the deadline. Thank you and look forward to hearing from you.
> > >
> > > Best regards,\
> > > Authors

---

### Official Review · Reviewer_aQCE · 2024-11-04

**Soundness:** 3
**Presentation:** 3
**Contribution:** 2
**Rating:** 6
**Confidence:** 2

**Summary:**

This paper first analyzes how to use IV regression for endogeneity, where the IV regression is estimated with 2SLS. Then, an error bound is proposed for this process. Furthermore, the paper shows that the transformer can achieve the gradient-based 2SLS for in-context linear regression.

**Strengths:**

I am not familiar with IV regression and endogeneity, so I haven't reviewed the mathematical accuracy of the theorem. My comments are based solely on a basic understanding of the motivation, overall contribution, and presentation. Please consider them with low weights.

Overall, I find the paper well-motivated, with clear writing. The background information and literature review appear thorough.

Understanding the mechanism of the Transformer could be valuable for advancing future research in this area.

**Weaknesses:**

I understand that theoretical analysis requires specific assumptions. I am, however, curious whether it might be possible to extend the theoretical analysis to the non-linear case, as most real-world scenarios tend to be non-linear.

Additionally, could you provide an example of real-world applications where the proposed analysis could be beneficial? If such examples exist, is it feasible to validate the analysis experimentally?

**Questions:**

Please refer to the weaknesses section.

---

> ### Author Response · Authors · 2024-11-22
> **Author Response**
>
> Thank you for your valuable comments on our paper and the positive evaluation. We appreciate the opportunity to address your concerns.
>
> **[W1] Can the theoretical analysis be extended to nonlinear case?**
>
> We agree that most of the real-world problems are non-linear in nature. However, we also emphasize that for various applications in econometrics, it is common to assume linearity between **y**  and **x**, for the interpretability reasons. Regarding the relation between **x** and **z**, our experiments in the original draft show that the pre-trained transformer model provides consistent coefficient estimates when the relationship between **x** and **z** is quadratic. In the revised draft, we additionally examine a more complicated non-linear case in Appendix C.5.
>
> While it is an interesting question to extend our theoretical analysis to the case of non-linear relationships (both between **y** and **x**, and **x** and **z**), some of the underlying assumptions need to be revisited, and we expect there will be structural changes in the proof. We leave this as a future work.
>
> **[W2] Real-world applications and experiments.**
>
> Thank you for your suggestion. To illustrate the performance of pre-trained transformers on real-world data, we now include an example in Appendix C.7. In this example, we analyze the effect of childbearing on mother's labor supply, with instrument being the indicator variable of whether the first two siblings are of the same sex. The transformer gives an estimate that with each increase in the number of children, the mother's labor supply is expected to drop 4.73 weeks per year. This result is statistically close to the 2SLS estimate, a widely agreed upon benchmark for this dataset.
>
> **We sincerely hope that we have addressed many of your questions, in which case, we would greatly appreciate if you could increase your score as you see fit.**

---

> > ### Author Response · Authors · 2024-11-25
> > **Gentle reminder**
> >
> > Dear Reviewer aQCE,
> >
> > As we are getting closer to the end of the discussion period (Nov 26th), we were wondering if you could let us know if we have sufficiently addressed your questions (we would also greatly appreciate if you could increase your score as you see fit to reflect this).
> >
> > Please let us know if you have any additional questions and we will be happy to address them as much as we can before the deadline. Thank you and look forward to hearing from you.
> >
> > Best regards,
> > Authors

---

> > ### Comment · Reviewer_aQCE · 2024-11-25
> > **Thanks for the author's response**
> >
> > Thank you for the author’s response.
> >
> > I would like to reiterate that I am not well-versed in IV regression and endogeneity. While I am making efforts to learn about these concepts during the rebuttal phase, I still do not feel confident enough to assess the correctness and contribution of this work comprehensively. Thus, I will maintain my score.
> >
> > Additionally, during my learning process, I came across a related paper that may be useful for discussion: Causal Interpretation of Self-Attention in Pre-Trained Transformers, NeurIPS 2023.

---

### Official Review · Reviewer_feNa · 2024-11-05

**Soundness:** 3
**Presentation:** 3
**Contribution:** 3
**Rating:** 6
**Confidence:** 1

**Summary:**

The paper extends the theoretical analysis done in previous work that looks at the class of functions that transformers can learn (e.g. simple linear regression) to a more complex set -- those with endogeneity and corresponding instrument variables. The authors show that transformers can learn this function and do as well as the direct solvers in most cases and potentially better in more challenging cases. They supplement this with a theoretical analysis.

**Strengths:**

I want to caveat this review by saying that I have flagged to the AC that this is not my area of expertise.

The paper seems interesting: it extends the theoretical analysis done in previous work that looks at the class of functions that transformers can learn (e.g. simple linear regression) to a more complex set -- those with endogeneity and corresponding instrument variables. The authors show that transformers can learn this function and do as well as the direct solvers in most cases and potentially better in more challenging cases. They supplement this with a theoretical analysis.

**Weaknesses:**

I want to caveat this review by saying that I have flagged to the AC that this is not my area of expertise.

I do not know enough about this area to understand the potential flaws in their statements / etc or more subtle points.
At a broad level, their intro / overview makes sense and seems convincing.

**Questions:**

I want to caveat this review by saying that I have flagged to the AC that this is not my area of expertise.

I wonder how this work helps one to understand what capabilities a transformer should / should not be capable of? Can we say something more concrete / general around what class of functions we can argue that a transformer should be able to solve? How does this relate to larger models ?

---

> ### Author Response · Authors · 2024-11-22
> **Author Response**
>
> Thank you for your valuable comments on our paper and the positive evaluation. We appreciate the opportunity to address your concerns.
>
>  **[Q1] I wonder how this work helps one to understand what capabilities a transformer should / should not be capable of?**
>
> We emphasize that our work provides a theoretical analysis of the transformer model's ability to perform in-context instrumental variable (IV) regression and deliver reliable coefficient estimates under endogeneity. Specifically, we demonstrate the existence of an intrinsic attention structure that enables transformers to execute GD-based 2SLS. We then propose a pretraining scheme and prove that under this scheme, the pretrained transformer achieves a small excess loss. Furthermore, our experiments reveal that the pretrained transformer model not only effectively addresses standard IV tasks but also generalizes well to more challenging scenarios, providing superior estimates compared to standard 2SLS and OLS estimators.
>
> At a general level, your question is quite open-ended. Various studies, such as Wei et al. (2022) and Sanford et al. (2024), have explored the general algorithmic representational capabilities of transformers. However, the results in these works are necessarily coarse, as they operate in a broad and general framework. In contrast, our work is motivated by a recent line of research, including Bai et al. (2024), Akyürek et al. (2023), Von Oswald et al. (2023), Li et al. (2023), Fu et al. (2023), and Ahn et al. (2024), which focus on specific statistical problems and provide more refined, fine-grained results. However these prior works do not address the challenge of handling endogeneity. Our work fills this gap by establishing precise bounds on the in-context learning capabilities of transformers for addressing endogeneity in linear models through instrumental variable regression.
>
>  **[Q2] Can we say something more concrete / general around what class of functions we can argue that a transformer should be able to solve?**
>
> In contrast to several works demonstrating that transformers can implement gradient descent and Newton's method for standard single-level optimization problems (e.g., argmin-type tasks), our analysis fundamentally argues that transformers are capable of executing a single step of bi-level gradient descent using two attention layers. Based on our findings, we believe that the proposed framework can be extended to other tasks that can be effectively formulated as bi-level optimization problems. These include various statistical learning challenges, such as hyperparameter tuning, adversarial machine learning, and fairness-related problems.
>
>  **[Q3] How does this relate to larger models?**
>
> It is somewhat unclear to us what you mean by "larger models." If you are referring to larger transformer architectures, works such as Wei et al. (2022) and Sanford et al. (2024) offer insights in that direction. We would greatly appreciate it if you could clarify your question by providing more specific details.
>
> Wei, Colin, Yining Chen, and Tengyu Ma. "Statistically meaningful approximation: a case study on approximating turing machines with transformers." Advances in Neural Information Processing Systems 35 (2022): 12071-12083.
>
> Sanford, Clayton, Daniel J. Hsu, and Matus Telgarsky. "Representational strengths and limitations of transformers." Advances in Neural Information Processing Systems 36 (2024).
>
> **We sincerely hope that we have addressed many of your questions, in which case, we would greatly appreciate if you could increase your score as you see fit.**

---

> > ### Author Response · Authors · 2024-11-25
> > **Gentle Reminder**
> >
> > Dear Reviewer feNa,
> >
> > As we are getting closer to the end of the discussion period (Nov 26th), we were wondering if you could let us know if we have sufficiently addressed your questions (we would also greatly appreciate if you could increase your score as you see fit to reflect this).
> >
> > Please let us know if you have any additional questions and we will be happy to address them as much as we can before the deadline. Thank you and look forward to hearing from you.
> >
> > Best regards,\
> > Authors

---

> > > ### Comment · Reviewer_feNa · 2024-11-25
> > > **Response**
> > >
> > > I thank the authors for the efforts and I apologise for the fact that I do not have the appropriate background to properly assess and discuss the work.
> > >
> > > Similar to Reviewer aQCE, I am not well versed in this area -- hence presumably the somewhat open ended questions. I do not think I have the knowledge to validate the correctness / contributions of this work so I will maintain my score.

---

### Author Response · Authors · 2024-11-22
**General Response**

We thank the reviewers for their positive evaluation and constructive feedback. In this general response, we would like to highlight the growing attention from the ML and AI communities toward the problem of instrumental variable (IV) regression, as evidenced by several recent publications in top machine learning conferences (e.g., Singh et al., 2019, NeurIPS; Muandet et al., 2020, NeurIPS; Bennett et al., 2023, COLT; Harris et al., 2022, ICML; Hartford et al., 2017, ICML; Cheng et al., 2024, ICLR; Wu et al., 2022, ICML). This increasing interest is driven by the connection of IV regression to various problems in causal inference and double machine learning.

Given this context, we believe our contribution is both timely and significant. Specifically, we demonstrate that transformers not only are theoretically capable of in-context IV regression but also empirically outperform traditional IV regression baselines (e.g., 2SLS) in addressing challenges such as weak instruments, multicollinearity, and certain forms of non-linearity.

Our work bridges modern AI methods like transformers with fields such as econometrics and sociology (which are consumers of IV regression methodology), while also deepening our understanding of the capabilities of these advanced models. We are encouraged by the broader implications of this research and its potential to catalyze further investigation and advancements in this area.

We would also like to point out that, based on the reviewer feedback, we have made the following updates to the draft.

1. **Multi-collinearity**: We have added new sections, Appendix C.3 and C.4 showing theoretically and empirically that transformers are capable of handling certain levels of Multi-collinearity.
2. **Non-linear extension**: Please note that our original draft contained empirical verification of the case of quadratic model for **x** and **z**. In  Appendix C.5, we have now added another case when the relationship between **x** and **z** is given by a two-layer neural network.
3. **Varying Endogeneity**: In Appendix C.6, we have now added an experiment showing the performance of transformers on varying levels of endogeneity.
4. **Real-data experiments**: In Appendix C.7, we have now added real-data experiments on a canonical dataset for IV regression evaluation, showing the competitive performance of transformers.







Hartford, Jason, et al. "Deep IV: A flexible approach for counterfactual prediction." International Conference on Machine Learning. PMLR, 2017.

Muandet, Krikamol, et al. "Dual instrumental variable regression." Advances in Neural Information Processing Systems 33 (2020): 2710-2721.

Singh, Rahul, Maneesh Sahani, and Arthur Gretton. "Kernel instrumental variable regression." Advances in Neural Information Processing Systems 32 (2019).

Bennett, Andrew, et al. "Minimax Instrumental Variable Regression and $ L_2 $ Convergence Guarantees without Identification or Closedness." The Thirty Sixth Annual Conference on Learning Theory. PMLR, 2023.

Harris, Keegan, et al. "Strategic instrumental variable regression: Recovering causal relationships from strategic responses." International Conference on Machine Learning. PMLR, 2022.

Cheng, Debo, et al. "Conditional Instrumental Variable Regression with Representation Learning for Causal Inference." The Twelfth International Conference on Learning Representations.

Wu, Anpeng, et al. "Instrumental variable regression with confounder balancing." International Conference on Machine Learning. PMLR, 2022.

---

### Meta-Review · Area_Chair_iwN6 · 2024-12-20

**Metareview:**

The paper investigates the ability of transformers to address endogeneity in in-context linear regression, proposing that transformers can emulate the two-stage least squares (2SLS) method through gradient-based optimization. All reviewers gave positive feedback while the confidence scores are relatively low. The paper's quality is high and authors engaged with rebuttals, so I see no clear reason to reject this work. I recommended an acceptance.

**Additional Comments On Reviewer Discussion:**

The reviewers acknowledged the authors' efforts on rebuttal.

---

### Decision · Program_Chairs · 2025-01-22

Accept (Poster)